# Synth-FAR: A Synthetic Frequency-Autoregressive Driven Framework for Time Series Forecasting

**Liran Nochumsohn**[*]    *lirannoc@post.bgu.ac.il*
*The Stein Faculty of Computer and Information Science*
*Ben-Gurion University of the Negev, Israel*

**Michal Moshkovitz**[*]    *michal.moshkovitz@bosch.com*
*Bosch Center of AI*
*Haifa, Israel*

**Orly Avner**[*]    *orly.avner@gmail.com*
*New Bosmat Technological High School*
*Haifa, Israel*

**Dotan Di Castro**[*]    *dotan.dicastro@gmail.com*
*ForSight Robotics*
*Caesarea, Israel*

**Omri Azencot**    *azencot@bgu.ac.il*
*The Stein Faculty of Computer and Information Science*
*Ben-Gurion University of the Negev, Israel*

**Reviewed on OpenReview:** *https://openreview.net/forum?id=XXXX*

## Abstract

Time series forecasting is essential for predicting future values based on observed patterns. Traditional methods perform well in in-domain scenarios with ample data but struggle with scarce data, leading to the rise of zero-shot and few-shot learning. Recent advancements use large-scale models but require extensive data and resources, often learning ineffectively from the available data. This study explores factors influencing effective learning in time series forecasting using Fourier analysis. Findings show that forecasters struggle with data containing multiple frequencies and generalizing to unseen frequencies. To address this, we introduce *Synth-FAR*, a synthetic data generation framework that enhances or replaces real data by creating a mixture of autoregressive and frequency information, improving model robustness in limited data scenarios. Our method outperforms other popular synthetic data techniques, such as Kernel-Synth, in both generation time and performance, and demonstrates the potential for integration into foundation model data pipelines, thereby enhancing their effectiveness.

## 1 Introduction

Time series forecasting (TSF) plays a critical role in various areas, such as finance, healthcare, and energy, where accurate predictions of future values are essential for decision-making and planning. Traditionally,

---

[*]Authors were affiliated with *Bosch Center for AI* during the preparation of this manuscript.

in-domain learning has been the common setting for developing forecasting models, where a model is trained using data from the same domain it will later be deployed in (Salinas et al., 2020; Zhou et al., 2021). This ensures that the model captures the patterns, seasonality, and trends specific to the target domain, improving its predictive performance. However, a significant challenge arises when there is scarce or no historical information available for training, limiting the ability to apply traditional in-domain learning approaches (Sarmas et al., 2022; Fong et al., 2020). In such cases, the emergence of zero-shot (ZS) and few-shot (FS) learning settings offer potential solutions. Zero-shot learning enables models to generalize to new, unseen domains without requiring domain-specific data by leveraging knowledge transfer from other domains or tasks. Few-shot learning, on the other hand, allows fine-tuning on limited amounts of domain-specific data. In this paper, we focus mostly on the ZS TSF setting.

Zero-shot techniques for TSF are often built upon foundation models, which are pre-trained on vast amounts of diverse data and can generalize to a wide range of tasks (Das et al., 2024; Ansari et al., 2024). However, foundation models (FMs) face several challenges, such as their huge data requirements, high computational costs, difficulties in fine-tuning for specific applications, and the risk of model over-generalization, which can lead to sub-optimal performance on specialized tasks (Ekambaram et al., 2024; Liu et al., 2024b). Moreover, foundation models often struggle to fully exploit the train distribution, limiting their ability to capture domain-specific patterns crucial for accurate zero-shot time series forecasting (see Sec. 4). A potential approach to alleviate data and compute limitations is to train models, including non-foundational ones, on task-specific *synthetic* information (Dooley et al., 2024), thus eliminating real-data requirements and reducing compute time. Unfortunately, the factors that govern effective learning from synthetic data using non-foundation models remain unclear, and our work aims to advance the general understanding of this challenge.

Among the various approaches, we advocate that *Fourier analysis* is the appropriate framework for assessing the effectiveness of synthetic data in TSF (Yi et al., 2025). Fourier analysis decomposes a signal into its constituent frequencies, revealing periodic patterns, smoothing out noise, and identifying important frequency-based features (Körner, 2022). This helps understand overfitting and underfitting as forms of *frequency generalization* and *frequency adaptation*, respectively (see Sec. 4). Analyzing synthetic data through Fourier transforms clarifies how well the data captures true underlying patterns, leading to better adaptation and generalization while avoiding over-representation of irrelevant details. By harnessing Fourier analysis in time series forecasting with both non-foundational and foundational models, we identify several shortcomings. First, increasing the available frequencies in the training set while fixing those in the test set leads to inferior test results. Second, test performance improves with better alignment of train and test sets in the frequency space. Finally, foundation models overfit to certain frequencies, underperforming on general frequencies. Based on our findings, we propose that *the training set should span the predominant frequencies of the target domain*. While intuitive, this is often infeasible as the span of target frequencies is unknown. Instead, we design a method that generates lightweight, target-oriented synthetic data using a mixture of frequencies and autoregressive processes, with two configurations: 1) based on the sampling rate of the target domain, and 2) without any information on the target domain. We evaluate our approach, **Synth**etic **F**requency-**AR** (*Synth-FAR*), in zero-shot settings on recent state-of-the-art models, showing its effectiveness compared to other methods. Our main contributions include:

1. We analyze the importance of frequencies in time series models, especially in the context of generalization. We introduce the concepts frequency adaptation and frequency generalization, which facilitate the identification of potential challenges in ZS forecasting.

2. We propose a simple, easy-to-code, multi purpose and efficient time series synthetic generator that is effective for different tasks.

3. Through extensive tests, we demonstrate that our synthetic data achieves better (ZS) results than other synthetic datasets and complements foundation and non-foundation models in areas where they fall short.

## 2 Related Work

**In-domain time series forecasting.** For decades, non-deep statistical TSF models held the state-of-the-art (SOTA) status (Makridakis & Hibon, 2000; Makridakis et al., 2008), but in recent years, purely neural network-based SOTA approaches for TSF have emerged (Salinas et al., 2020; Oreshkin et al., 2020). Rapid development has led to various techniques including the usage of trend and seasonality (Zhou et al., 2022), patching time series (Nie et al., 2023), exploiting inter-channel relations (Liu et al., 2024c), and many others (Wu et al., 2021; Zhang & Yan, 2023; Xu et al., 2024; Nochumsohn et al., 2025; Zexer & Azencot, 2026). These approaches, however, have been considered in the in-domain setting, where there is an available training set that statistically aligns with the test set.

**Zero-shot.** While several attempts have been made in utilizing non-foundation models for ZS and FS TSF (Orozco & Roberts, 2020; Oreshkin et al., 2021; Jin et al., 2022; Nochumsohn et al., 2026), interest has quickly shifted to foundation models. Specifically Large Language Models (LLMs) are commonly considered, using various backbones including GPT-2 (Zhou et al., 2023; Liu et al., 2024b), LLaMA (Jin et al., 2024), and others (Gruver et al., 2024). Additional methods exploit trend-seasonality-residual decompositions (Cao et al., 2024) and Transformer blocks (Goswami et al., 2024). One of the main limitations of foundation models is their dependence on large volumes of data. Thus, recent studies have incorporated synthetic information alongside real data involving seasonal patterns and trends (Das et al., 2024) and Gaussian processes (Ansari et al., 2024). Closely related to our work is ForecastPFN (Dooley et al., 2024), where the authors perform zero-shot time series forecasting by training solely on synthetic data. Still, we argue that the factors determining effective learning in such settings remain unclear, particularly for non-foundation models.

**Frequency-domain learning models.** Fourier analysis and spectral methods provide a principled framework for decomposing signals into frequency components, and have become increasingly influential across modern machine learning (Yi et al., 2025). In deep learning, frequency-domain representations have been utilized for model compression (Rippel et al., 2015), data augmentation and invariance learning (Yang & Hong, 2022; Nochumsohn & Azencot, 2025), and operator learning through architectures such as the Fourier Neural Operator (FNO) (Li et al., 2021). Neural architectures have also incorporated both real-valued (Xu et al., 2020) and complex-valued (Cao et al., 2020) spectral representations.

In time series forecasting, spectral methods have been explored both implicitly and explicitly. Classical approaches such as ARIMA and seasonal decomposition rely on frequency-domain interpretations of periodicity (Shumway & Stoffer, 2000), while more recent neural architectures directly integrate frequency-aware modeling components. These include decomposition-based and frequency-enhanced Transformers such as Autoformer (Wu et al., 2021) and FEDformer (Zhou et al., 2022), the latter explicitly performing attention in the Fourier domain. Additional architectures such as TimesNet (Wu et al., 2023), N-BEATS (Oreshkin et al., 2020), CycleNet (Lin et al., 2024a), and spectral-temporal graph models (Cao et al., 2020) further emphasize periodic decomposition and frequency-aware representations. Recent works also introduce explicit periodic pattern modeling and frequency embedding mechanisms (Ekambaram et al., 2024; Liu et al., 2024b; Cao et al., 2024).

Closely related to our work are studies leveraging synthetic or large-scale pretraining data with diverse frequency content to improve generalization, including Chronos (Ansari et al., 2024), ForecastPFN (Dooley et al., 2024), and TimesFM (Das et al., 2024). Another related research topic is generative modeling of time series (Naiman et al., 2024b;a; Fadlon et al., 2026; Gonen et al., 2026). Other works have explored spectral augmentation and robustness, such as FrAug (Chen et al., 2023) and related frequency-mixing strategies.

**Spectral learning and analysis.** Beyond architectural design, a growing body of work studies how neural networks learn, represent, and generalize frequency information. Early studies on *spectral bias* show that deep neural networks tend to fit low-frequency components before high-frequency ones during gradient-based optimization (Rahaman et al., 2019). Related formulations of the Frequency Principle (F-Principle) further demonstrate that neural networks preferentially learn smooth and simple patterns prior to complex oscillatory structures (Xu et al., 2019a;b). Subsequent theoretical analyses connected these phenomena to the implicit regularization properties of optimization and neural tangent kernels (NTKs) (Jacot et al., 2018; Basri

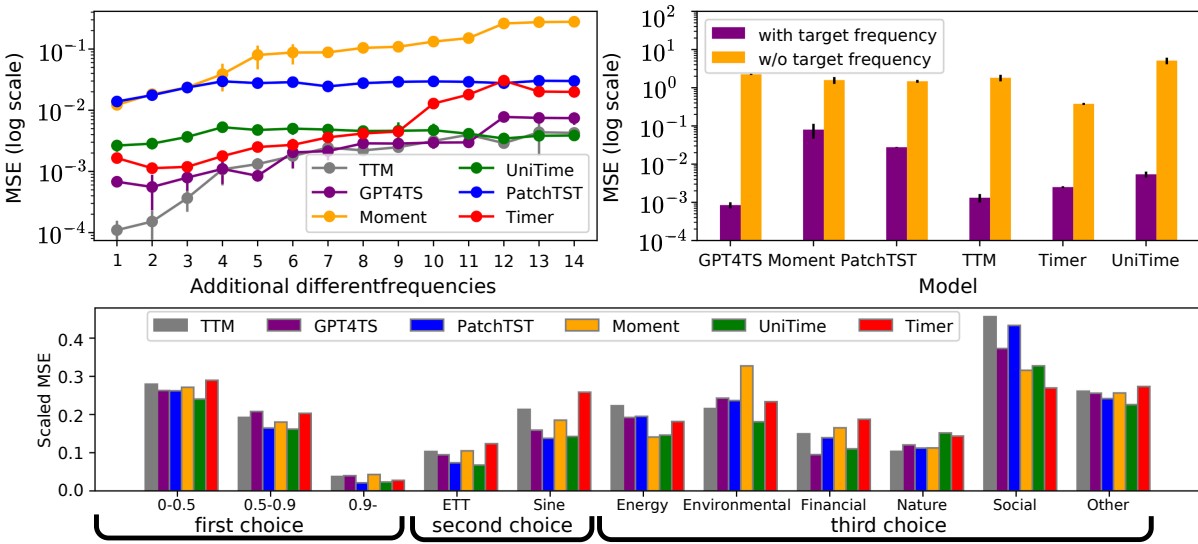

Figure 1: Top: We demonstrate poor frequency adaptation, where adding more frequencies gradually degrades performance (left). We also observe large performance differences when the fundamental target frequency exist vs. absent in the training set, implying poor frequency generalization (right). Bottom: Transfer learning performance bars for various frequency-based alignments between the train and test sets. Ranged values represent the periodogram PCC (left), ETT is a single dataset with different sampling rates (middle), and the remaining are sector based categories (right).

et al., 2020; Cao et al., 2021). Other works showed that standard coordinate-based networks struggle to represent high-frequency signals, motivating Fourier feature mappings and sinusoidal implicit representations for improving spectral expressivity and approximation fidelity (Tancik et al., 2020; Sitzmann et al., 2020; Benbarka et al., 2022).

Frequency-oriented analyses have also been explored across downstream temporal tasks including anomaly detection (Ren et al., 2019; Wu et al., 2025), classification (Wang et al., 2018; Zhang et al., 2022; Cui et al., 2016), imputation (Yang et al., 2024; Wang et al., 2025), and representation learning for temporal signals (Demirel & Holz, 2024). More broadly, classical signal-processing and statistical forecasting literature has long relied on spectral estimation, harmonic analysis, Fourier decomposition, and wavelet methods for modeling periodicity, seasonality, and non-stationary temporal dynamics (Jenkins & Priestley, 1957; Shumway & Stoffer, 2000; Percival & Walden, 1993; Mallat, 1999). Related forecasting approaches based on harmonic regression and Fourier series expansions have also been widely adopted for capturing multi-scale seasonal structure and long-range temporal dependencies (Hyndman & Athanasopoulos, 2018; De Livera et al., 2011).

In contrast to prior works that primarily incorporate spectral representations into model architectures or study spectral properties of neural optimization, our work focuses on how frequency alignment, frequency coverage, and spectral mismatch influence transfer and zero-shot forecasting performance across datasets and synthetic training distributions. In particular, we study generalization under frequency shift and analyze how synthetic data can improve robustness to unseen or underrepresented frequencies.

# 3 Background

To motivate our analysis and approach to zero-shot and few-shot forecasting as discussed in Sec. 4, we present basic results related to Fourier analysis and time series information, see also (Shumway & Stoffer, 2000). It is well-known that for any time series sample $x_1, \ldots, x_n \subset \mathbb{R}^n$ and under carefully chosen coefficients, we have

for odd $n$ that

$$x_t = a_0 + \sum_{j=1}^{(n-1)/2} \left[ a_j \cos(2\pi t\ j/n) + b_j \sin(2\pi t\ j/n) \right] \ , \tag{1}$$

for $t = 1, \ldots, n$, $a_0$ is the bias, $a_j$ and $b_j$ are the amplitude coefficients, and $t \in \mathbb{Z}$. The frequencies $\omega_j := j/n$ represent cycles per time unit, where a cycle is a complete period of the cosine or sine, e.g., for $\omega = 0.5$, the series makes two cycles per time unit. We also consider an equivalent form, obtained via a trigonometric identity of Eq. 1 and given by

$$x_t = a_0 + \sum_{j=1}^{(n-1)/2} A_j \cos(2\pi t\ \omega_j + \phi_j) \ , \tag{2}$$

where the amplitude $A_j = \sqrt{a_j^2 + b_j^2}$ and $\phi_j = \tan^{-1}(b_j/a_j)$ is the phase of the $j$th frequency, express the standard deviation and the cosine function starting point respectively. Notably, dominant periodic components in a signal are associated with larger amplitudes.

Another important concept for our work is the *periodogram* (Schuster, 1898). We define the scaled periodogram, which is closely related to the amplitude $A_j$, and it is defined via

$$P(\omega_j) = A_j^2 \ , \tag{3}$$

where large values of $P(\omega_j)$ correspond to predominant *fundamental frequencies $j/n$*, whereas small values of $P(\omega_j)$ can be viewed as noise. In practice, the scaled periodogram can be estimated via the discrete Fourier transform (DFT), which represents a weighted average of the data $d(\omega_j) = n^{-1/2} \sum_{t=1}^{n} x_t \exp(-2\pi i t\ j/n)$, with $i$ the imaginary number. It follows that $P(\omega_j) = \frac{4}{n}|d(\omega_j)|^2$. Finally, *Harmonics* represent frequencies of the form $k\bar{\omega}_j$ for a dominant fundamental frequency $\bar{\omega}_j$, $k \in \mathbb{N}$. They appear in time series when non-sinusoidal components contribute to the structure of the signal. Below, we will show that harmonics, as depicted in the periodogram, are crucial in understanding the effect of data on zero-shot and few-shot learning and information transfer in large time series models.

## 4 Fourier Analysis for Zero-Shot TSF

Many existing approaches for zero-shot TSF are based on large foundation models (Ansari et al., 2024). These neural networks are computationally demanding and need large volumes of data for training. In this work, we aim to maximize the learning efficiency from data, with the goal of reducing data and compute requirements, especially in the ZS settings, where data is scarce or unavailable. Particularly, we are interested in answering the following overarching question:

*What factors govern effective learning in zero-shot time series forecasting?*

Understanding such factors better may lead to reducing data requirements by generating compact task-specific synthetic data. Similarly, compute reduction can be achieved by using non-foundation models on that data. Ultimately, if we succeed in answering the above question, we could potentially employ *non-foundation* models for solving zero-shot TSF, training solely on *synthetic data*.

### 4.1 Frequency adaptation and generalization

Toward uncovering the factors that determine effective learning, we examine time series forecasting through the lens of Fourier analysis. Specifically, to quantify the differences between the train and test sets and their corresponding forecasting errors, we will use the periodogram (see Sec. 3) and the following two new frequency-based concepts that were adapted from the *Transfer learning* literature (Wang et al., 2022).

**Definition 4.1** (Frequency Adaptation). The model's ability to perform well in the case where the training set consists of the target frequencies along with other, unrelated, frequencies.

**Definition 4.2** (Frequency Generalization). The model's ability to perform well during inference on data with frequencies that were unavailable during training.

In other words, Def. 4.1 describes the model's difficulty in learning from multiple frequencies, where some may be unnecessary. The term originates from *domain adaptation* (Wang et al., 2022; Wang & Deng, 2018; Singhal et al., 2023), which addresses entire dataset domains rather than frequency-associated data. It is also closely related to *capacity*, introduced in (Han et al., 2024) to assess data fit, and *domain confusion* (Liu et al., 2024b) related to datasets from different domains. Def. 4.2, on the other hand, deals with the ability of a trained model to obtain consistent performance across learned as well as unseen frequencies. This definition is closely related to *domain generalization* (Wang et al., 2022), where there, the generalization is in the context of performing well on datasets of different domains.

Equipped with these definitions, we consider experiments, aiming to identify whether frequency adaptation and frequency generalization assist in understanding model behavior. For these experiments, we use recent non-foundation and foundation state-of-the-art (SOTA) TSF models including GPT4TS (Zhou et al., 2023), Moment (Goswami et al., 2024), PatchTST (Nie et al., 2023), TTM (Ekambaram et al., 2024), Timer (Liu et al., 2024e), and UniTime (Liu et al., 2024b). The first experiment trains and infers the above models on a simple sine wave dataset with $\omega = 1/24$, representing an hour to daily based sampling rate. From here and throughout our discussion, we interchangeably use the terms sampling rate and frequency. Then, we incrementally add additional sine waves with various frequencies to the training set, re-train, and measure the prediction error of the $\omega = 1/24$ sine wave only. We plot in Fig. 1 (left) the mean squared error (MSE) of the prediction in log scale vs. the number of additional train frequencies. Notably, all models present an increase in test MSE, even if mild, as more frequencies are added, suggesting that they struggle with frequency adaptation.

In the second experiment, we used the same models. However, now every model is trained twice: on a mixed frequency sine wave training set including the target sine wave, and without it. We plot the test MSE errors in log scale and present the results in Fig. 1 (right). The bar chart shows that in all cases, having access to the target frequency sine wave (purple) leads to significant performance gains in comparison to training without that frequency (orange). This experiment suggests that deep TSF models may struggle to generalize to unseen frequencies, even on simple toy examples.

The third experiment explores the effect of similar and dissimilar frequency spaces in the context of transfer learning. Using the above models, we utilize a pool of datasets, see Fig. 14. We train each model separately on every dataset in the pool, and we use the learned model to infer over all the remaining datasets. To make the test MSE of different datasets comparable, we perform min-max normalization between train datasets per training setup as shown in Fig. 14. We organize the results into clusters based on the following choices: 1) The train and test sets share similar or dissimilar frequencies, as measured by the periodogram. 2) The train and test are sampled from the same dataset (ETT, Sine), but with different sampling rates (frequencies). Specifically, ETTh1 and ETTm1 are sampled at an hourly rate and per 15 minutes, respectively. 3) The train and test are from similar domains or sectors, e.g., energy-related information. We present in Fig. 1 the scaled MSE measures of this experiment for the abovementioned choices. Particularly, the three leftmost bar groups correspond to the Pearson correlation coefficient (PCC) of the periodogram between datasets (Choice 1). Then, ETT is the electricity transformer temperature dataset, and 'Sine' are sines waves of different rates (Choice 2). Finally, the six rightmost bar groups are various sector domains (Choice 3). The results show that while same-sector training may help (e.g., TTM on nature), the general performance is inconsistent. Similarly, using the same data (ETT) lowers the scaled MSE, however, it is still higher than training on datasets whose PCC is correlated in the frequency space (PCC $\geq 0.9$).

## 4.2 AR($p$) for complex frequencies

In certain datasets, unless the domain's dynamics are well understood, models may prefer to rely on the most recent values. For example, in datasets with complex frequencies or low frequencies, a limited input size may not be sufficient to capture the underlying frequency. Therefore, focusing on the most recent values of the input may be a better overall choice, which brings us to present the following concept.

An autoregressive process is based on the idea that the current value of the series, $x_t$ can be explained as a function of $p$ past values, where $p$ determines the number of steps into the past needed to forecast the current

value. Formally:

$$x_t = w_t + x_{t-1}\gamma_1 + x_{t-2}\gamma_2, ..., x_{t-p}\gamma_p \; , \tag{4}$$
$$\text{where,} \quad w_t \sim \mathcal{N}(0, \sigma_w^2) \; ,$$

By selecting an appropriate $p$, the model can better capture the underlying patterns in the data, leading to improved forecasting performance and most importantly, assist with frequency adaptation. This approach can be particularly beneficial for datasets with complex, low-frequency, and mixed frequency patterns, as it allows the model to adapt to the most relevant recent information, thereby enhancing its predictive accuracy, thus shifting its reliance on fixed frequency patterns.

To summarize, the above analysis reinforces the emergence of Fourier analysis as a useful perspective for studying factors that affect effective learning. Moreover, it leads to the following straightforward observation: *training on datasets that share a similar periodogram with the target data improves the results of deep neural networks for TSF, in zero-shot scenarios and more generally.* Unfortunately, this observation is impractical to implement, as the full target frequency distribution is unknown, and even if the frequency is known, datasets with complex frequencies pose a challenge. To address this, we propose a synthetic dataset with two configurations: 1) mixture of frequencies and AR processes. 2) target frequency enhanced, with a simple heuristic given the sampling rate mixed with AR processes, which we found to be highly effective.

### 4.3 Synth-FAR: Synthetic time series based on fundamental frequencies and AR($p$)

Following our analysis, we propose a simple yet effective approach to zero-shot TSF, which we refer to as *Synth-FAR*. Namely, we generate task-specific synthetic data and use it to train non-foundation and foundation models. Our approach has the potential to replace standard multi-dataset training of large time series models or serve as a complementary process. To generate data, we assume that the target distribution is mostly dominated by the fundamental target frequency and its harmonics augmented by varying trend and noise components. To create our synthetic dataset, we first generate a pool of sines and AR processes, followed by sampling from that pool and constructing various time series signals. We illustrate this approach in Fig. 2 in comparison to Synth-TimesFM and Kernel-Synth (Das et al., 2024; Ansari et al., 2024).

**A pool of sines and AR($p$).** Given the fundamental target frequency $\bar{\omega}$, we create a pool $P$ consisting of $m_s$ sine waves whose frequencies are the harmonics of $\bar{\omega}$, i.e., where the amplitude is sampled from an exponential distribution $A_k \sim \text{Exp}(A')$, and the phase is drawn from a uniform distribution $\phi_k \sim \mathcal{U}([0, 2\pi])$. The frequencies $\omega_k$ are sampled from a uniform distribution over the harmonics. Namely, we have that $\omega_k \sim \mathcal{U}(\Omega)$, where $\Omega := \{\bar{\omega}, 2\bar{\omega}, \dots, h\bar{\omega}\}$. Next we add to $P$ a total of $m_{ar}$ AR($p$) processes, as defined in Eq. 4, where $\sum_{j=1}^{p} \gamma_j = 1$ and $\gamma_j = 1/p$. This formulation offers some interesting properties. If $p = 1$, it trains the model to become *naive*. Conversely, if $p = p_{max}$, which takes the maximal lookback, we can expect a *mean* model that returns the mean of the input. The variables $m_s, h, A'$ and $m_{ar}, p$ are hyper-parameters whose values are detailed and ablated in App. B. The formulation of $P$ is defined as follows:

$$P := \{s_s^1, s_s^2, \dots, s_s^{m_s}, s_{ar}^1, s_{ar}^2, \dots, s_{ar}^{m_{ar}}\} \; , \tag{5}$$
$$\text{where,} \quad (s_s^k)_t := A_k \sin(2\pi t \, \omega_k + \phi_k) \; ,$$
$$\text{and,} \; (s_{ar}^k)_t = x_t^k := w_t + x_{t-1}^k \gamma_1 + x_{t-2}^k \gamma_2, ..., x_{t-p}^k \gamma_{t-p} \; .$$

**Dataset construction.** We generate a synthetic multivariate time series $x \in \mathbb{R}^{d \times n}$ of $d$ variates and length $n$, i.e., $x = (x_t^j)$, for $t = 1, \dots, n$ and $j = 1, \dots, d$, by sampling uniformly from $P$. In particular, we draw $l$ series per variate $j$ and sum them together. Formally,

$$x^j = \sum_{i=1}^{l} s^i \; , \text{ where, } \quad s^i \sim \mathcal{U}(P) \; , \; j = 1, \dots, d \; . \tag{6}$$

The hyper-parameter $l$ controls the number of series sampled from $P$, and it directly influences the correlation factor in that if $l$ is larger, then more variates in the time series are correlated. To create a full dataset, we

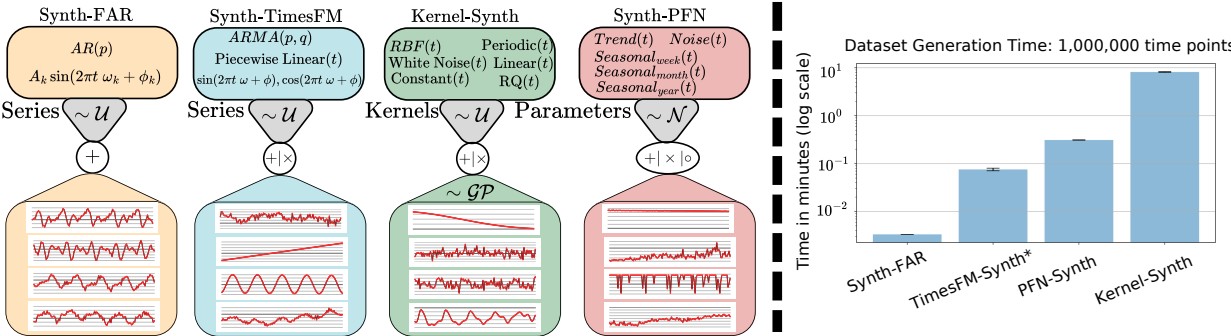

Figure 2: Left: Illustration of the generation frameworks for Synth-FAR, Synth-TimesFM, Kernel-Synth and Synth-PFN (left to right). Synth-FAR produces more distilled, frequency-focused patterns, while the remaining methods generate a broader and potentially overly diverse range of patterns, and requires more operations. Right: Dataset generation time.

Table 1: Comparison between different synthetic data methods with the known target sampling rate (left block) and without it (right block). The mean over the datasets is given in the last row. Red bold, marks lowest in the entire row, black marks lowest in each block.

| | Known Sampling Rate | | | | | | | | Unknown Sampling Rate (mix) | | | | | | | | | | | |
| | Synth-FAR | | Ker-Synth | | FM | | PFN | | Synth-FAR | | Ker-Synth | | FM | | PFN | | Naive | | Mean | |
| | MSE | MAE | MSE | MAE | MSE | MAE | MSE | MAE | MSE | MAE | MSE | MAE | MSE | MAE | MSE | MAE | MSE | MAE | MSE | MAE |
|---|---|---|---|---|---|---|---|---|---|---|---|---|---|---|---|---|---|---|---|---|
| ETTh1 | **0.452** | **0.439** | 0.506 | 0.460 | 0.619 | 0.529 | 0.783 | 0.578 | **0.529** | **0.484** | 0.555 | 0.494 | 1.045 | 0.664 | 0.710 | 0.557 | 1.294 | 0.713 | 0.701 | 0.558 |
| ETTh2 | **0.327** | **0.358** | 0.345 | 0.375 | 0.330 | 0.361 | 0.841 | 0.543 | **0.338** | **0.374** | 0.348 | 0.380 | 0.436 | 0.425 | 0.691 | 0.495 | 0.432 | 0.422 | 0.353 | 0.387 |
| ETTm1 | **0.397** | **0.393** | 0.456 | 0.403 | 0.750 | 0.539 | 2.392 | 0.962 | **0.576** | **0.480** | 0.688 | 0.523 | 2.705 | 0.810 | 2.378 | 0.959 | 1.214 | 0.665 | 0.693 | 0.548 |
| ETTm2 | 0.209 | 0.280 | 0.213 | **0.276** | **0.206** | 0.287 | 0.482 | 0.452 | 0.235 | 0.306 | **0.226** | **0.302** | 0.338 | 0.361 | 0.476 | 0.450 | 0.266 | 0.328 | 0.229 | 0.307 |
| Electricity | **0.290** | **0.357** | 0.318 | 0.381 | 0.440 | 0.478 | 0.704 | 0.497 | **0.330** | **0.383** | 0.471 | 0.492 | 1.154 | 0.840 | 0.580 | 0.469 | 1.588 | 0.945 | 0.846 | 0.762 |
| Traffic | **0.941** | **0.549** | 1.022 | 0.579 | 1.203 | 0.667 | 1.767 | 0.765 | **1.190** | **0.650** | 1.262 | 0.709 | 1.971 | 0.971 | 1.634 | 0.765 | 2.714 | 1.077 | 1.410 | 0.805 |
| Weather | 0.272 | 0.265 | **0.223** | **0.248** | 0.233 | 0.257 | 0.573 | 0.414 | 0.243 | **0.269** | 0.243 | 0.275 | 0.404 | 0.332 | 0.558 | 0.409 | 0.259 | 0.254 | **0.215** | 0.271 |
| Average | **0.413** | **0.377** | 0.440 | 0.389 | 0.540 | 0.445 | 1.077 | 0.602 | **0.492** | **0.421** | 0.542 | 0.454 | 1.150 | 0.629 | 1.004 | 0.586 | 1.110 | 0.629 | 0.635 | 0.520 |

simply repeat the above process to generate multiple time series. For additional diversity, one can create several datasets with different parameters such as the number of harmonics.

We propose to construct several datasets each associated with a fundamental frequency $\bar{\omega}$, given a mix of natural fundamental frequencies or the fundamental frequency (a scalar) of a target distribution. In the latter case, We derive the fundamental frequency using the sampling rate of the target dataset, which is typically given as a co-variate and exploited by several models (Cao et al., 2024; Liu et al., 2024b; Ekambaram et al., 2024; Das et al., 2024; Liu et al., 2024d). See also App. C.1 for details on the relation between the sampling rate and the fundamental frequency, and ways to estimate it.

## 5 Experiments and Analysis

In this section, we evaluate Synth-FAR in comparison to recent state-of-the-art (SOTA) forecasting approaches in three settings: Synthetic data comparison (Sec. 5.1), real and synthetic data pre-training evaluations (Sec. 5.2), and lastly a comparison to prominent pre-trained foundation models (Sec. 5.3). We consider the popular long-term time series forecasting (LTSF) benchmark (Zhou et al., 2021; Wu et al., 2021) for the first two settings, and the recently published Time-MMD repository (Liu et al., 2024a) for the last. We observed that foundation models exhibit a tendency to overfit specific frequencies and their associated datasets, particularly in the LTSF benchmark. Frequency over-fit along with other technical limitations, may hinder their performance on other target datasets. More details regarding foundation models' limitations which motivate our approach are described in App. C.4. For all experiments in this work, a lookback of 96 is employed. Additional details including dataset information and Synth-FAR implementations are given in App. 6 and App. C, respectively.

The implementation details of Synth-FAR are as follows: For each sampling rate, we generated 3 Synth-FAR datasets, each corresponding to a different maximum harmonics parameter $h$ as described in 4.3, specifically $h \in \{1, 2, 3\}$. From each generated dataset, we sample a total maximum of 10,000 sequences of size lookback+horizon, which are used for training or validation. For the known sampling rate setting, we create only 3 Synth-FAR datasets as described above. However, for the mixed setting, we employ 7 common frequencies: 1/12, 1/52, 1/7, 1/30, 1/24, 1/96, and 1/144, corresponding to monthly, weekly, daily-weekly, daily-monthly, hourly, 15 minutes, and 10 minutes, respectively, repeating the process above for each frequency. For $\text{AR}(p)$, we use $p = 8$ for all experiments, with all weights set to $\frac{1}{p}$. We provide ablation studies in App. B and additional parameters and details in App. C.

## 5.1 Comparing synthetic dataset approaches

We evaluate forecasting results using synthetic data, comparing recent approaches such as TimesFM synthetic data (Das et al., 2024), ForecastPFN (Dooley et al., 2024), and Chronos Kernel-Synth (Ansari et al., 2024). TimesFM and Chronos use synthetic data to diversify real data for pre-trained models, while ForecastPFN trains solely on synthetic data. We generate 500 channels, each 1024 in length, for ForecastPFN, TimesFM, and Kernel-Synth to ensure diversity. We compare two setups: 1) Known target sampling rate, and 2) Unknown target sampling rate, assuming no prior knowledge of the target domain. For the latter, we use a mixed variant of Synth-FAR, including datasets with common natural frequencies (e.g., 15 minutes, hourly, daily, and weekly) and their harmonics. For the synthetic datasets, we use their original implementation in Setup 2, and in Setup 1, we narrow their frequency space to the relevant fundamental frequency based on our heuristic given the sampling rate.

We detail in Tab. 1 the results, with the left and right blocks corresponding to known and unknown sampling rates, respectively. The MSE and MAE measures are averaged on a forecasting horizon of 96 across seven models, namely, TTM (Ekambaram et al., 2024), UniTime (Liu et al., 2024b), Moment (Goswami et al., 2024), Timer (Liu et al., 2024e), GPT4TS (Zhou et al., 2023), PatchTST (Nie et al., 2023), and TimeMixer Wang et al. (2024). As in Tab. 2. We also use the naive and mean forecasters as baselines for comparison. Notably, a large performance difference is observed between the left and right blocks, in favor of the known sampling rate case. Thus, utilizing the target sampling rate or the fundamental frequency facilitates the alleviation of poor frequency adaptation. When analyzing each block separately, we find Synth-FAR to be superior to the other baselines, presenting an average MSE reduction of **6.1%** vs. Kernel-Synth, and **23.5%** vs. (Times)FM in the left block, and an **9.2%** reduction vs. Kernel-Synth in the right block. Remarkably, while Synth-FAR trains on a approximately 50% of the data ForecastPFN, TimesFM, and Kernel-Synth use, it consistently obtains better error measures.

## 5.2 Real and synth data zero-shot forecasting

In this evaluation, we compare forecasting performances obtained by four settings. 1) **R**: Real data, 2) **R+S(M)**: Real data with Synth-FAR with mixed frequencies, 3) **S(M)**: Synth-FAR with mixed frequencies, and lastly 4) **S**: Synth-FAR with the sampling rate. In this experiment, we include the same models from the previous subsection also considering DLinear (Zeng et al., 2023). These models are either foundation models, designed for the zero-shot forecasting setting, or non-foundation models that marked significant milestones in developing such approaches. Inspired by TTM, we employ a slightly similar setup and train these baselines in the real data case on a subset of datasets from PEMS (Liu et al., 2022), and a subset of Monash (Godahewa et al., 2021), considering only datasets with a minimum length of approximately 1,000 timesteps. These datasets span a large range of sampling rates, including 4 seconds, 10 minutes, 1 hour, and more. Importantly, all the sampling rates of the evaluation data are also contained in this large training set, except for 15 minutes, which is the sampling rate of the ETTm datasets. We show in Tab. 2 the zero-shot forecasting results on several datasets (rows) as obtained by various methods (columns). We report the mean squared error (MSE), where each result represents the average MSE over the forecast horizons $96, 192, 336, 720$ and three random seeds. The bottom row lists the average errors across all datasets. In App. 11, we provide the full table.

Notably, in most cases, performance gradually improves from **R** to **S**. This improvement is due to poor frequency generalization in **R**, as seen with ETTm1, which shows significant enhancements when Synth-FAR

Table 2: A comparison of zero-shot forecasting when training with real data vs. synthetic data. Including Synth-FAR outperforms real data in the all cases, with a notable advantage to Synth-FAR with the sampling rate denoted (**S**). The Lowest MSE for each model is marked with bold, and second lowest with an underline. The full table is given in Tab. 11.

| Model | TTM | | | | Timer | | | | Moment | | | | UniTime | | | |
|---|---|---|---|---|---|---|---|---|---|---|---|---|---|---|---|---|
| Data Type | R | R+S(M) | S(M) | S | R | R+S(M) | S(M) | S | R | R+S(M) | S(M) | S | R | R+S(M) | S(M) | S |
| ETTh1 | 0.636 | **0.497** | 0.569 | 0.521 | 0.722 | 0.603 | 0.639 | **0.575** | 0.68 | 0.714 | 0.553 | **0.536** | 0.95 | 0.574 | 0.537 | **0.505** |
| ETTh2 | 0.438 | **0.408** | 0.43 | 0.418 | 0.601 | 0.604 | 0.455 | **0.452** | **0.408** | 0.422 | 0.436 | 0.424 | 0.562 | 0.526 | **0.407** | 0.422 |
| ETTm1 | 1.066 | **0.468** | 0.472 | 0.477 | 1.268 | 1.17 | 0.681 | **0.486** | 0.842 | 0.718 | 0.676 | **0.487** | 1.133 | 1.114 | 0.601 | **0.46** |
| ETTm2 | 0.337 | **0.307** | 0.313 | 0.313 | 0.414 | 0.414 | 0.354 | **0.33** | 0.326 | **0.321** | 0.336 | 0.33 | 0.407 | 0.382 | 0.317 | **0.308** |
| Exchange | 0.382 | 0.376 | **0.358** | 0.359 | **0.35** | 0.361 | 0.423 | 0.371 | 0.412 | 0.422 | 0.419 | **0.382** | 0.366 | **0.359** | 0.416 | 0.36 |
| Electricity | 0.499 | **0.283** | 0.321 | 0.304 | 0.516 | **0.305** | 0.344 | 0.32 | 0.768 | 0.862 | 0.38 | **0.321** | 0.573 | **0.298** | 0.31 | 0.303 |
| Traffic | 1.105 | **0.845** | 1.035 | 0.919 | 0.961 | **0.913** | 1.284 | 1.03 | 1.331 | 1.426 | 1.035 | **0.931** | 0.985 | 0.905 | 1.003 | **0.872** |
| Weather | 0.342 | **0.31** | 0.322 | 0.333 | 0.339 | **0.337** | 0.337 | 0.374 | **0.284** | 0.293 | 0.316 | 0.348 | 0.304 | **0.296** | 0.3 | 0.332 |
| Average | 0.601 | **0.437** | 0.478 | 0.456 | 0.646 | 0.588 | 0.565 | **0.492** | 0.631 | 0.647 | 0.519 | **0.47** | 0.66 | 0.557 | 0.486 | **0.445** |

| Model | TimeMixer | | | | GPT4TS | | | | PatchTST | | | | DLinear | | | |
|---|---|---|---|---|---|---|---|---|---|---|---|---|---|---|---|---|
| Data Type | R | R+S(M) | S(M) | S | R | R+S(M) | S(M) | S | R | R+S(M) | S(M) | S | R | R+S(M) | S(M) | S |
| ETTh1 | 0.763 | 0.573 | 0.59 | **0.53** | 0.615 | 0.592 | 0.635 | **0.462** | 1.958 | 1.119 | 0.538 | **0.493** | 0.621 | 0.549 | 0.704 | **0.529** |
| ETTh2 | 0.624 | 0.57 | 0.428 | **0.424** | 0.525 | 0.488 | 0.44 | **0.397** | 0.617 | 0.586 | 0.416 | **0.407** | 1.01 | 0.609 | 0.493 | **0.457** |
| ETTm1 | 1.335 | 1.069 | 0.659 | **0.505** | 1.188 | 0.79 | 0.74 | **0.426** | 1.243 | 1.114 | 0.572 | **0.445** | 0.943 | 0.782 | 0.682 | **0.518** |
| ETTm2 | 0.417 | 0.401 | 0.341 | **0.324** | 0.393 | 0.356 | 0.342 | **0.302** | 0.448 | 0.443 | 0.32 | **0.3** | 0.964 | 0.54 | 0.386 | **0.36** |
| Exchange | 0.367 | 0.37 | 0.422 | **0.361** | 0.392 | 0.4 | 0.423 | **0.367** | 0.362 | **0.359** | 0.413 | 0.361 | 0.786 | 0.375 | 0.276 | **0.236** |
| Electricity | 0.396 | **0.295** | 0.329 | 0.308 | 0.442 | 0.316 | 0.366 | **0.29** | 0.649 | **0.302** | 0.323 | 0.304 | 0.413 | 0.34 | 0.388 | **0.309** |
| Traffic | 0.897 | 0.884 | 1.153 | 0.937 | 0.986 | 0.92 | 1.24 | **0.824** | 0.954 | 0.887 | 1.072 | **0.87** | 0.982 | 1.004 | 1.53 | **0.93** |
| Weather | 0.295 | **0.294** | 0.322 | 0.369 | 0.311 | **0.309** | 0.32 | 0.334 | 0.313 | 0.324 | **0.305** | 0.321 | 0.343 | **0.294** | 0.319 | 0.444 |
| Average | 0.637 | 0.557 | 0.53 | **0.47** | 0.606 | 0.521 | 0.563 | **0.425** | 0.818 | 0.642 | 0.495 | **0.438** | 0.758 | 0.562 | 0.597 | **0.473** |

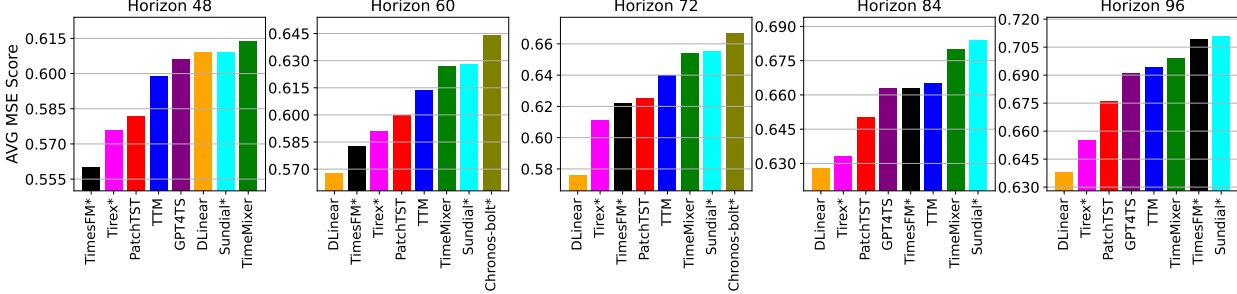

Figure 3: The average MSE performance across 10 Time-MMD datasets. The plots from left to right represent the performance with respect to the horizons 48, 60, 72, 84, and 96. Only the eight best-performing models are given in each plot.

is included (e.g., 1.188 vs. 0.79 when selecting GPT4TS **R+S(M)**). Secondly, poor frequency adaptation is prevalent in different synthetic and real data settings, as more frequencies require attention, with an exception for the TTM model, which demonstrates superior performance in the **R+S(M)** setting. This can be attributed to the model's effectiveness in handling various data types and frequencies, highlighting the importance of using Synth-FAR to complement real data and improve performance. The majority of the lowest MSE scores and the lowest average MSE are observed in the **S** column, which suffers less from poor generalization and adaptation. This experiment is valuable as it underscores the pitfalls of using arbitrary datasets for training a large and general model for TSF. Specifically, we use a lookback of 96, which is the original standard lookback for the LTSF benchmarks (Zhou et al., 2021), and is not commonly used by pre-trained TSF foundation models for training and evaluation, where a lookback of 512 and above is standard but not always available. In the next subsection, however, we extend the topic of pre-trained foundation models.

## 5.3 Pre-trained foundation models

To demonstrate the effectiveness of Synth-FAR in areas where pre-trained foundation models may fall short, we employ a new benchmark comprised of the recently introduced Time-MMD dataset repository (Liu et al.,

2024a). In this experiment, we use Synth-FAR with the sampling rate and the in-domain forecasting models DLinear, GPT4TS, TimeMixer, and PatchTST. The model TTM is also included as we find it notably effective. All mentioned models are trained solely on Synth-FAR. Additionally, we include the performance of prominent pre-trained models available online, namely Chronos (Ansari et al., 2024), MOIRAI (Woo et al., 2024), Timer (XL) (Liu et al., 2025a; 2024e), and TimesFM (Das et al., 2024). Unfortunately, the pre-trained version of TTM is not suitable for a lookback of 96, highlighting some limitations of foundation models.

The results, shown in Fig. 3, exhibit the average MSE score across the 10 datasets for horizons 48, 60, 72, 84, and 96, for the 8 best-performing models. Detailed results are provided in App. 10. To differentiate between pre-trained foundation models and non-foundation models, we add the "*" symbol to mark pre-trained models. The results suggest that, in most cases, all synthetic-trained models perform consistently better on average than Timer*, Chronos*, and MOIRAI*, which are not shown as they are not in the best-performing group. Notably, DLinear outperforms TimesFM* for all horizons except for 48. This experiment suggests that although foundation models in TSF have made significant improvements in recent years, there is still room for improvement, and their performance in other domains remains unclear.

## 5.4 Data Generation time

Generating synthetic data introduces overhead to the computation pipeline. In what follows, we compare the generation time of different synthetic approaches. We test this by generating a single dataset with one million time points comprised of 1000 variates each of length 1000 for each of the methods, Synth-FAR, TimesFM, ForecastPFN, and Kernel-Synth. The time required by each method is 0.2 seconds for Synth-FAR, 4.5 seconds for TimesFM, 18.6 seconds for ForecastPFN, and 8 minutes for Kernel-Synth, presenting a significant advantage to Synth-FAR (see inset). For each reported result, we calculated the average creation time of three datasets. In addition to the generation time, we also note that Synth-FAR is easy-to-code, requiring only a few lines of code as presented in the code snippet in App. C.3.

## 6 Datasets and Models

In this section, we provide additional details regarding the experimental settings, models, and datasets. Each experiment was carried out three times with three different random seeds to ensure robustness and reliability of the results. Our objective was to maintain fidelity to the original parameters of each model, while establishing a unified framework for consistent comparison. Therefore, for each model we employ the original implementation with slight modifications to allow a fair comparison in a unified framework. Throughout the experiments, a lookback of 96 and a train,test and validation fraction split of 0.6,0.2,0.2 respectively for ETT datasets and 0.7,0.2,0.1 for the remaining datasets was employed for training in accordance with the original protocol for the LTSF (Informer) datasets (Zhou et al., 2021; Wu et al., 2021; Liu et al., 2024c). The reported results represent the test fraction of the data. All experiments were conducted with NVIDIA RTX3090 32GB GPU, and each experiment was trained end to end on a single GPU.

### 6.1 Models

In this work we selected the following models for evaluation:

- **PatchTST** (Nie et al., 2023). An in-domain TSF transformer-based model, introducing instance normalization, patching, a simple vanilla transformer and linear projection. PatchTST is a notable model, as many later released large-scale TSF models employ similar components including instance normalization, linear projection, patching, patch masking and reconstruction.

- **GPT4TS** (Zhou et al., 2023). A unified time-series model designed for a range of tasks including forecasting. GPT4TS uses a pre-trained frozen GPT-2 model, under the assumption that language domain data could be adapted to time-series data. GPT4TS is an important milestone towards foundation models as it showed success with employing a unified pre-trained language transformer for a range of downstream tasks with fine-tuning.

- **TTM** (Ekambaram et al., 2024) is a pre-trained model with a light-weight architecture which utilizes diverse resolution sampling with the implementation of patches of different lengths and resolution prefix tuning, allowing the model to encode sampling rate specific information.

- **Timer** (Liu et al., 2024e) employs a GPT-style architecture, originally designed for a range of tasks such as imputation, anomaly detection, and forecasting. Although originally designed for auto-regressive next token prediction, we employ a non-autoregressive setup in this work, aligning our evaluation with other models.

- **Moment** (Goswami et al., 2024). A transformer-based foundation model for time series, designed for various downstream tasks such as forecasting, classification, imputation, and anomaly detection. Moment utilizes transformers, patching, and learnable mask embedding.

- **TimeMixer** (Wang et al., 2024). A fully MLP-based architecture with Past-Decomposable-Mixing (PDM) and Future-Multipredictor-Mixing (FMM) blocks. PDM decomposes and mixes seasonal and trend components, while FMM combines multiple predictors for enhanced forecasting.

- **UniTime** (Liu et al., 2024b). A cross-domain large forecasting model empowered by a trainable GPT-2 backbone. UniTime also employs masking for generalization and to increase convergence speed. Language prompts are also utilized for identification information for training. However, we find in our implementation that this contribution hurts performance in ZS, therefore we do not provide "domain-instructions" as suggested in the original paper.

- **Chronos** (Ansari et al., 2024). Chronos is a framework for pretrained probabilistic time series models that tokenize time series values and trains transformer-based language models. It uses the T5 family of models, pre-trained on a mix of public and synthetic datasets, namely Kernel-Synth, to enhance generalization.

- **DLinear** (Zeng et al., 2023). DLinear decomposes raw data into trend and seasonal components using a moving average kernel. It applies one-layer linear models to each component and sums the results for the final prediction, enhancing performance when a clear trend is present.

- **TimesFM** Das et al. (2024). TimesFM for time-series forecasting uses a decoder-only architecture with input patching, breaking time-series into patches to improve performance and inference speed. It employs a large-scale corpus of real-world and synthetic data for pretraining. The model predicts longer output patches than input patches, enhancing accuracy for long-horizon forecasting. Patch masking is used during training to handle various context lengths.

- **MOIRAI** Woo et al. (2024) uses a patch-based approach with a masked encoder architecture for time series modeling, flattening multivariate series into a single sequence. It employs a multi-patch size input projection layer and a multi-patch size output projection layer to decode tokens into mixture distribution parameters. The core Transformer module is an encoder-only architecture with various improvements, including pre-normalization, RMSNorm, query-key normalization, and SwiGLU non-linearity.

- **TiReX** (Auer et al., 2026) is a decoder-only pre-trained time series forecasting model built on xLSTM blocks that enable efficient recurrence and strong state tracking. The model converts normalized time series into non-overlapping patches, embeds them into tokens processed by stacked xLSTM layers, and outputs quantile forecasts for probabilistic prediction. During training, it uses Contiguous Patch Masking (CPM) and data augmentations to improve long-horizon forecasting and robustness, enabling strong zero-shot generalization across unseen datasets.

- **Sundial** (Liu et al., 2025b) a decoder-only Transformer-based time series foundation model that operates directly on continuous-valued patch tokens, avoiding discrete tokenization. It models the full predictive distribution of future patches using a generative objective called TimeFlow Loss, which applies flow-matching to transform Gaussian noise into future forecasts conditioned on Transformer representations of the historical context.

**Model Selection.** Although the given models offer a limited scope with respect to the available models, we selected these baselines for several reasons: 1) code availability for training: an easy access to trainable, original implementations through Hugging Face or github. 2) Performance and time efficiency: these works offer a thorough comparison to other comparable methods and showed better overall performance including faster inference or training time. For example, TTM's superior inference speed compared to other models (Ekambaram et al., 2024). 3) Prominence: for example PatchTST and GPT4TS are important milestones towards foundation models for TSF, due to their popularity and architectural contributions. These reasons eventually guided our decision making toward model selection for evaluation.

**Comparison protocol.** Our goal is not to exactly reproduce the original training procedures of each baseline, but rather to evaluate all methods under a unified experimental protocol. Whenever possible, we use the original implementations and recommended hyperparameters while adapting only those components necessary to ensure consistent data splits, lookback lengths, optimization settings, and evaluation metrics across models. Some foundation models impose architectural or pretraining constraints that prevent perfectly identical training configurations; these limitations are inherent to the models themselves rather than to the proposed method. We therefore report results obtained under a common evaluation framework to facilitate a controlled comparison.

## 6.2 Datasets

In this work, we evaluate the proposed Synth-FAR on the common LTSF (Zhou et al., 2021) benchmark datasets and the recently introduced Time-MMD repository (Liu et al., 2024a). We train the baseline models in Tab. 2 and on a subset of datasets from the Monash repository (Godahewa et al., 2021). Specifically, we select the datasets that have a minimum length of 1,000 timesteps, in order to enable a training configuration of horizon length 720, which requires each example to be 846 timeseps long. The PEMS repository (Liu et al., 2022) is also included for training. This training setup is slightly similar to the one employed in (Ekambaram et al., 2024). In Tab. 3, we provide details regarding the selected datasets for training and testing. To ensure that certain large datasets do no dominate training, we limit the maximum number of examples per dataset to 500,000 for training and validation. Selecting a subset of the entire training set is also a common practice in large unified training frameworks (Ekambaram et al., 2024; Liu et al., 2024e). In our case, it can prevent large datasets with many examples to engulf the effect of smaller and medium size datasets during training. The given train datasets cover a range of sectors such as nature, energy, traffic and financial and various sampling rates. Most sectors and sampling rates of the evaluation datasets are included in the train data for the experiments in 5.2, except for the sampling rate for ETTm1 and ETTm2. For the Time-MMD datasets used in this work for ZS only, each variate is normalized with standard scaler separately.

In this work we focus on datasets with a variety of seasonal prominence. Not all datasets in our evaluations exhibit purely seasonal patterns; each dataset possesses a different degree of seasonality. This is illustrated in Figure 4, where we have included a *measure of seasonality* (Wang et al., 2006) and sample series from every evaluated dataset used in this work. It is evident that while datasets like Traffic and electricity display strong seasonal patterns, others such as ETTm2 and Climate have weaker seasonal components.

## 6.3 Synthetic Datasets

In what follows, we provide additional details regarding the synthetic datasets discussed in Sec. 5.1.

- **TimesFM** (Das et al., 2024): Synthetic generated data, where each channel selects up to three possible components that are eventually added together, or multiplied (trend only) among ARMA process, mixture of cosines and sines, and piece-wise linear trends. In this work, we provided results based on our implementation as the original implementation is not available.

- **ForecastPFN** (Dooley et al., 2024): They assume that there exists a shared distribution among real time-series datasets, which can be derived from natural periodic data, trend, global trends and noise. ForecastPFN synthetic data applies multiplication and addition to create signals that meet their prior distributions criteria. To handle extreme scales in their generated data, they introduce robust scaling and outlier removal, which is also employed for ForecastPFN in Tab. 1.

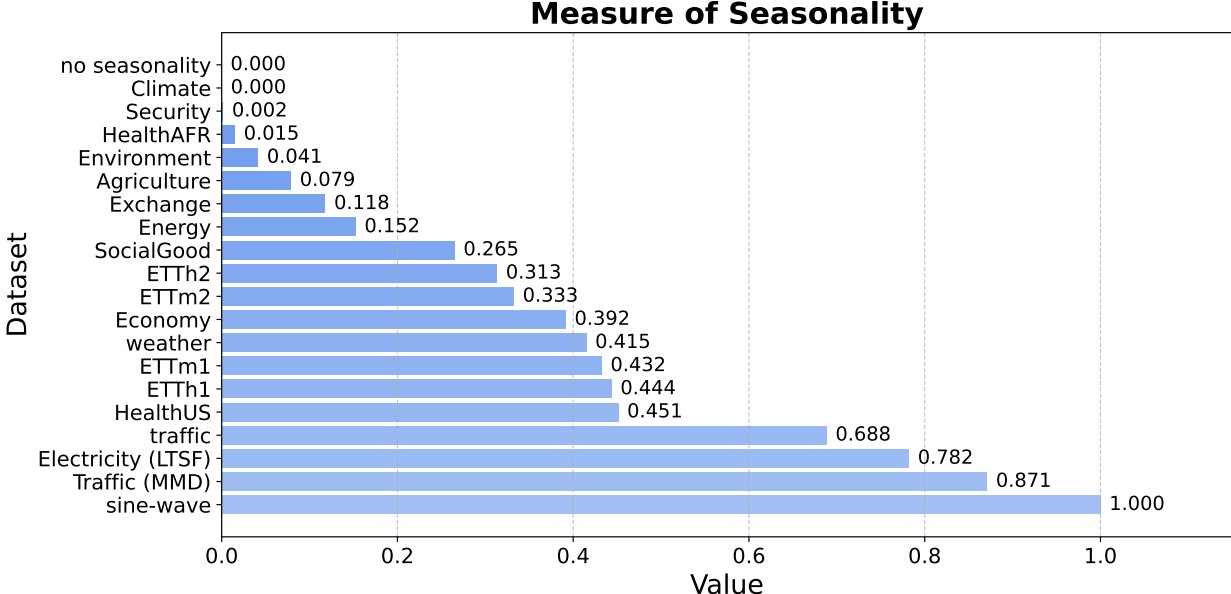

Figure 4: The *Measure of Seasonality* metric of each dataset used in this work. The score 0 represents no seasonality, whereas 1 represents a sine-wave without noise. Formulations: $1 - \frac{VAR(noise)}{VAR(noise+seasonal)}$, where *noise*, *seasonal*, are extracted with STL decomposition, and $VAR$ returns the series variance.

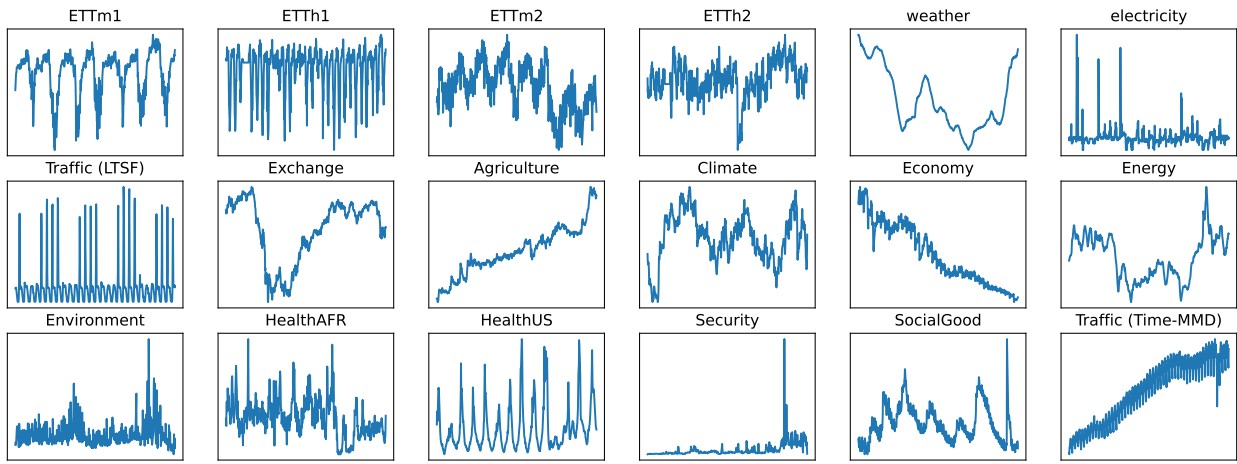

Figure 5: Examples of the different datasets used for evaluation in this work.

- **Kernel-Synth** (Ansari et al., 2024): a Gaussian process (GP)-based synthetic time series generation method. Kernels are sampled from a kernel bank and then randomly combined using a binary operator ($\times$ or $+$). The resultant kernel is used in a GP prior to the generation of a synthetic time series.

In each of these methods, the generated channels are independent of the other channels, yet they attain cross-channel relations such as correlations and causality due to the underlying generation process. Synth-FAR on the other hand, supports multivariate channels with a controllable degree of linear similarly (Pearson

Table 3: Details on the considered datasets.

| Dataset | Repository | Channels | Min/max channel length | Sampling rate | Sector | Usage |
|---|---|---|---|---|---|---|
| ETTh1 | LTSF | 7 | 17,420 | hourly | Energy | Evaluation |
| ETTh2 | LTSF | 7 | 17,420 | hourly | Energy | Evaluation |
| ETTm1 | LTSF | 7 | 69,680 | 15 minutes | Energy | Evaluation |
| ETTm2 | LTSF | 7 | 69,680 | 15 minutes | Energy | Evaluation |
| Electricity | LTSF | 321 | 26,304 | hourly | Energy | Evaluation |
| Traffic | LTSF | 862 | 17,544 | hourly | Transport, Environmental | Evaluation |
| Weather | LTSF | 21 | 52,696 | 10 minutes | Nature | Evaluation |
| Exchange | LTSF | 8 | 7,588 | daily | Financial | Evaluation |
| London Smart Meters | Monash | 5,560 | 288/39,648 | 30 minutes | Energy | Training |
| Aus. Electricity Demand | Monash | 5 | 230,736/232,272 | 30 minutes | Energy, Environmental | Training |
| Wind Farms | Monash | 339 | 6,345/527,040 | minutely | Energy | Training |
| Bitcoin | Monash | 18 | 2,659/4,581 | daily | Financial | Training |
| KDD Cup 2018 | Monash | 270 | 9,504/10,920 | hourly | Nature, Environmental | Training |
| Weather (Monash) | Monash | 3,010 | 1,332/65,981 | daily | Nature | Training |
| Solar | Monash | 137 | 52,560 | 10 minutes | Nature | Training |
| Sunspot | Monash | 1 | 23,741 | daily | Nature | Training |
| Us Births | Monash | 1 | 7,305 | daily | Nature | Training |
| Saugeen River Flow | Monash | 1 | 23,741 | daily | Nature | Training |
| Solar Power | Monash | 1 | 7,397,222 | 4 seconds | Energy | Training |
| Wind Power | Monash | 1 | 7,397,147 | 4 seconds | Energy | Training |
| PEMS03 | PEMS | 358 | 25,887 | 5 minutes | Transport | Training |
| PEMS04 | PEMS | 307 | 16,992 | 5 minutes | Transport | Training |
| PEMS07 | PEMS | 883 | 28,224 | 5 minutes | Transport | Training |
| PEMS08 | PEMS | 170 | 17,856 | 5 minutes | Transport | Training |
| Agriculture | Time-MMD | 1 | 496 | Monthly | Agriculture | Evaluation |
| Climate | Time-MMD | 5 | 496 | Monthly | Nature | Evaluation |
| Economy | Time-MMD | 3 | 423 | Monthly | Financial | Evaluation |
| Energy | Time-MMD | 9 | 1479 | Weekly | Energy | Evaluation |
| Environment | Time-MMD | 4(1) | 11102 | Daily | Environmental | Evaluation |
| Health US | Time-MMD | 11(7) | 1389 | Weekly | Social, Healthcare | Evaluation |
| Health AFR | Time-MMD | 1 | 1459 | Weekly | Social, Healthcare | Evaluation |
| Security | Time-MMD | 1 | 297 | Monthly | Social | Evaluation |
| Social Good | Time-MMD | 1 | 900 | Monthly | Social | Evaluation |
| Traffic | Time-MMD | 1 | 531 | Monthly | Transport | Evaluation |

* The brackets (x) mark the number of variates used for forecasting in practice.

correlation) through the parameter $l$. In Tab. 1, each synthetic dataset was standardized per channel except for ForecastPFN which was scaled with a robust scaler in accordance with the original implementation.

# 7 Conclusion

Deep foundation models are increasingly employed for zero-shot and few-shot time series forecasting. However, our analysis reveals key challenges in learning from complex frequency patterns and limited data. Using Fourier analysis, we show that both foundation and non-foundation models struggle with learning from multiple frequencies (poor frequency adaptation) and with generalizing to unseen frequencies (poor frequency generalization). To address these challenges, we propose *Synth-FAR*, a lightweight synthetic data generation framework that enriches or replaces real data through frequency- and autoregressive-driven synthetic signals. Extensive experiments demonstrate that Synth-FAR consistently improves forecasting performance and robustness in zero-shot settings across both foundation and non-foundation models while requiring substantially less synthetic data generation time than existing approaches.

Beyond the proposed generator, this work provides a frequency-oriented perspective for understanding effective learning in zero-shot time series forecasting, highlighting the importance of frequency alignment and coverage when constructing synthetic training data. Although Synth-FAR consistently improves performance across the evaluated benchmarks, its design is primarily motivated by frequency-oriented analysis and is therefore expected to provide the greatest benefit on datasets exhibiting meaningful periodic structure. While our experiments demonstrate competitive performance even on datasets with weaker seasonality, extending the framework to better capture non-periodic and highly irregular temporal dynamics remains an important direction for future research. More broadly, we hope that the proposed analysis and principles will inspire future synthetic data generation strategies and contribute to the development of more robust and data-efficient forecasting models.

**Acknowledgments**

This research was partially supported by the Lynn and William Frankel Center of The Stein Faculty of Computer and Information Science, Ben-Gurion University of the Negev, ISF grants 668/21 and 1299/25, an ISF equipment grant, and by the Israeli Council for Higher Education (CHE) via the Data Science Research Center, Ben-Gurion University of the Negev, Israel.

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

# A  Appendix

In this appendix, we provide additional information and details to supplement the main body of the paper. This includes ablation studies in App. B, supplementary method details in App. C, prominent foundation models limitations in C.4, datasets and models details in App. 6, and extended tables and results in App. E that were not included in the main text. The purpose of this appendix is to provide readers with a more comprehensive understanding of the research methodology and results, as well as to offer further insights into the experimental procedures and analysis.

# B  Ablation and Additional Experiments

## B.1  Ablation: AR($p$) processes

To evaluate the effectiveness of the AR($p$) model, we conducted an ablation study to compare the impact of different values of $m_{ar}$ on the results. Specifically, we tested $m_{ar} = 0$, $m_{ar} = 10$, and $m_{ar} = 100$ on TTM, UniTime, Timer, and Moment with a forecast horizon of 96. The results are presented in Fig. 6. In the mixed frequency setting, for all datasets, the AR($p$) processes demonstrate a positive effect, with a noticeable gradual decrease in MSE. This indicates that AR($p$) acts as a conflicting frequency mitigator that enhances frequency adaptation. In the fixed sampling rate setup, where the sampling rate is predetermined, it is observed that AR($p$) processes may sometimes degrade performance, as seen in the Electricity, ETTh1, and Traffic datasets. However, for the Weather dataset, increasing AR($p$) improves performance in both setups. This improvement may be attributed to the limitations of the sampling rate heuristic or the presence of complex frequencies in the Weather dataset.

All datasets and configurations include AR($p$) using the same $m_{ar}$ parameters. Unfortunately, since we are working with zero-shot cases, we cannot alter these parameters for specific cases. However, if permitted, it is recommended to do so, as datasets with very strong seasonal components may suffer slightly due to the added random process.

We find that AR($p$) is essential, based on the observations made in Section 4.2, if the frequency cannot be easily extracted by the model due to lacking frequencies, complex frequencies, or low frequencies that are not easily captured, then the other dominant factors are the last values of the input, which are generally correlated with the forecast output. Based on this observation, we found that AR($p$) supports models with:

1. **Better Adaptation**: In a mixed frequency setting, AR($p$) improves performance (see Fig 6).

2. **Forecast Stability**: In a mixed frequency setting, AR($p$) lowers the output variance (see Fig 6 error bars).

3. **Regularization**: If the period estimation is not accurate, or the dataset has only low or complex frequencies, AR($p$) aids in minimizing the error (see Fig 5. 6 Weather dataset). Additionally, AR($p$) avoids over-fitting to specific frequencies.

## B.2  Ablation: Harmonics

We conducted experiments to explore the impact of different harmonics on performance, as shown in Fig. 7. In this experiment, we created five configurations. Each configuration, except for the last, contains a training set consisting of three Synth-FAR generated datasets. The first four configurations consider only $h = 1$, $h = 2$, $h = 3$, and $h = 4$, respectively. The last configuration contains a mix of harmonics $h \in 1, 2, 3$, one for each generated dataset $X$. This configuration resembles the one used in this work.

If the sampling rate is known, considering $h > 1$ improves performance in all datasets except for the Weather dataset, most notably in ETTh1, Traffic, and Electricity. This suggests that Synth-FAR better assists the model in learning the structure of finer periodic signals, as shown in App. 13. The Weather dataset, on the other hand, sees performance degradation, which could be explained by the presence of complex signals,

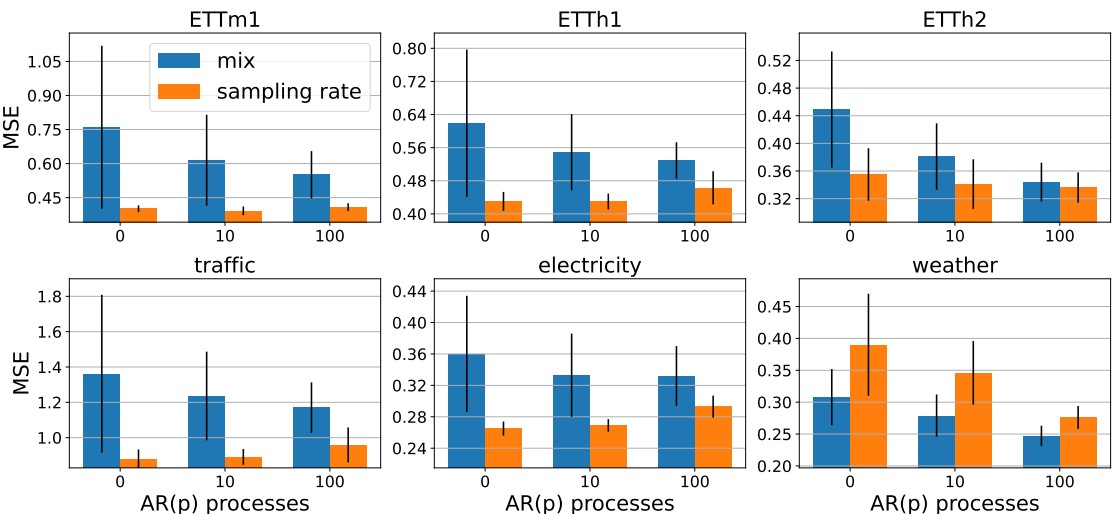

Figure 6: The influence of the number of series with AR($p$) on the ZS performance per dataset, where each result reports the average MSE for TTM, UniTime, Timer, and Momentt for a forecast horizon 96.

low frequencies, or failure to approximate the fundamental frequency. In such cases, a mixed approach is better, as it may include the relevant frequency of interest and its harmonics, avoiding overfitting to a specific frequency.

In the mixed Synth-FAR setting, a gradual improvement with larger $h$ is observed only for ETTm1, ETTh1, and Weather. The mixed frequency setting introduces more frequencies, leading to poorer frequency adaptation. In the original setting used in this paper, we mix harmonics parameters $h \in 1, 2, 3$ to allow different representations. This approach is shown to be better than the worst case for each dataset and relatively consistent in performance across different configurations.

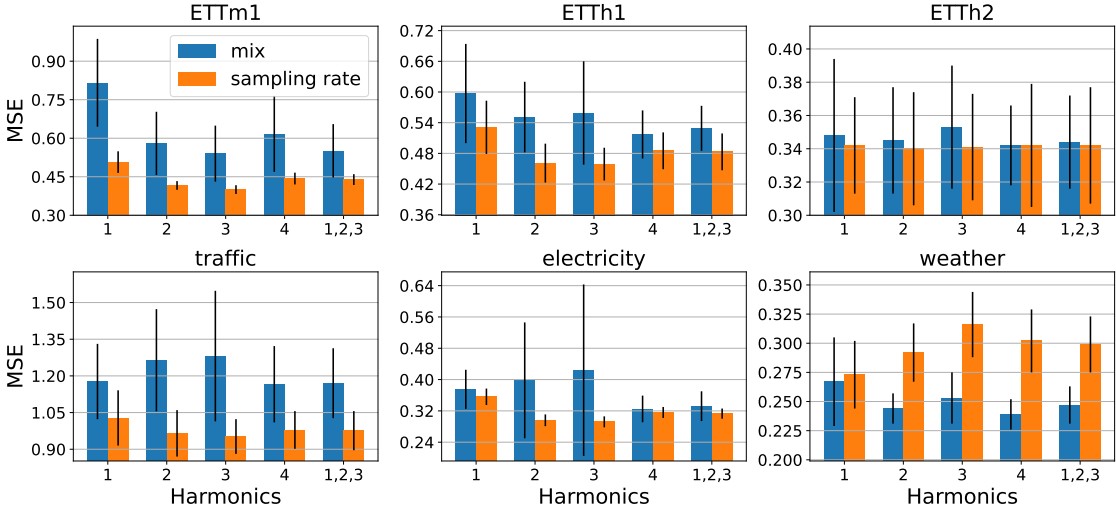

Figure 7: The influence of the number of harmonics on the ZS performance per dataset, where each result reports the average MSE for TTM, Timer, Moment, UniTime, and Moment for a forecast horizon 96.

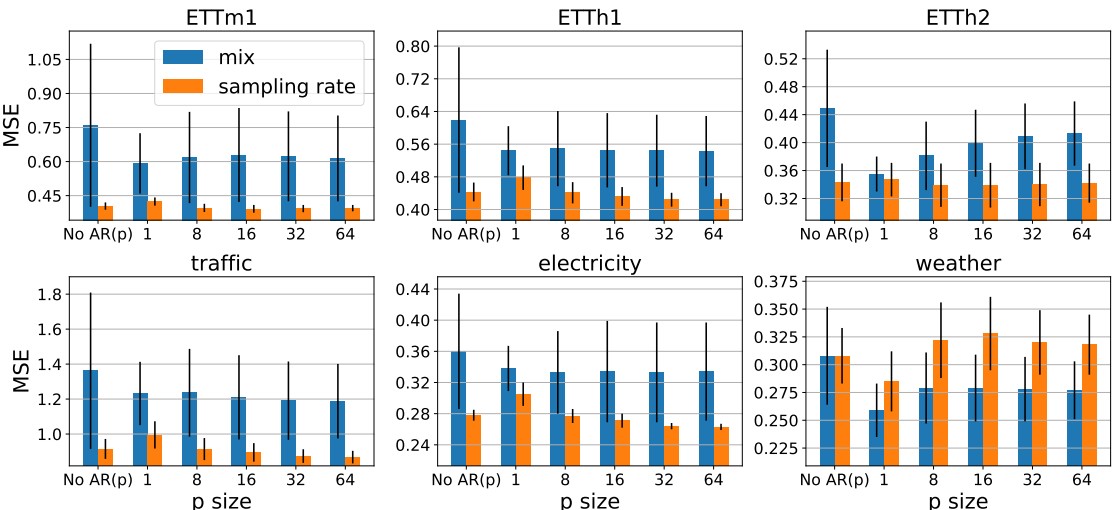

Figure 8: The influence of the $p$ in AR($p$) per dataset, where each result reports the average MSE for TTM, Timer, Moment, UniTime, and Moment for a forecast horizon 96.

### B.3   Ablation: AR $p$ size

In this subsection, we conduct experiments to test different values for $p$, which determines the size of the lookback to consider for the autoregressive process, as defined in 4. In this work, we select $p = 8$. In the sampling rate setting, not including AR at all exhibits inferior performance in most cases, with the exception of the Weather dataset. This inferiority becomes more apparent with the mixed frequency setting, where any form of $p > 1$ shows improvement. Similar to the other ablations, we do not find an optimal parameter $p$. For instance, $p = 1$ is preferable for Weather in both the mix and sampling rate settings. However, for Electricity and ETTh1, $p = 1$ is the worst option, even worse than not having AR($p$) in the sampling rate setting.

### B.4   Varying Lookbacks

We conducted additional experiments for the models TTM, Moment, and Timer using commonly employed lookbacks for evaluations: 192, 336, and 512, in addition to 96, as shown in Table 2 of the main text. For the models: TTM and Moment, their capacity increases as follows: 0.15M, 0.37M, 0.87M, and 7.18M for TTM, and 39.76M, 44.18M, 50.82M, and 58.92M for Moment, corresponding to lookbacks of 96, 192, 336, and 512, respectively. The results strongly suggest that injecting Synth-FAR with real data (**R+S(M)**) can significantly enhance performance for Timer and TTM, regardless of the lookback size or capacity. For Moment, the model appears to utilize the synthetic data with the sampling rate noted as **S** more effectively than other settings. In conclusion, Synth-FAR is shown to be effective for both larger capacities and lookbacks.

### B.5   Embedding Frequency Knowledge in Time Series Models

To further evaluate the effectiveness of Synth-FAR, additional experiments were conducted comparing models trained exclusively on Synth-FAR—specifically DLinear (S) and TTM (S)—against SparseTSF, CycleNet, and TTM (Lin et al., 2024b;a; Ekambaram et al., 2024). These models incorporate architectural components designed to explicitly capture seasonal patterns, leveraging input frequency or sampling rate to enhance zero-shot performance. DLinear was selected as a baseline due to its structural similarity to SparseTSF and CycleNet, which also utilize linear transformations as a core modeling element.

The results, summarized in Table. 5, demonstrate that TTM (S) and DLinear (S) achieve lower mean squared error (MSE) in 6 out of 7 and 5 out of 7 cases, respectively. These findings suggest that Synth-FAR is competitive, and in many cases superior, even when compared to models that explicitly encode periodic structure.

Table 4: The MSE performance of utilizing S(**M**) for mixed Synth-FAR and **S** for frequency-specific Synth-FAR with varying lookback cases: 96, 192, 336, and 512. All settings employ a 96 steps horizon forecast. The model capacity of TTM and Moment increases as the lookback grows.

| Model | TTM | | | | Timer | | | | Moment | | | | Lookback |
|---|---|---|---|---|---|---|---|---|---|---|---|---|---|
| Data Type | R | R+S(M) | S(M) | S | R | R+S(M) | S(M) | S | R | R+S(M) | S(M) | S | |
| ETTh1 | 0.592 | **0.438** | 0.522 | _0.459_ | 0.66 | **0.521** | 0.576 | **0.521** | 0.64 | 0.701 | _0.547_ | **0.434** | |
| ETTh2 | 0.357 | _0.324_ | 0.337 | **0.322** | 0.52 | 0.531 | **0.344** | _0.368_ | **0.32** | 0.353 | 0.382 | _0.323_ | |
| ETTm1 | 1.105 | _0.404_ | 0.408 | **0.398** | 1.239 | 1.165 | _0.62_ | **0.422** | 0.922 | 0.693 | _0.637_ | **0.422** | |
| ETTm2 | 0.236 | **0.2** | 0.207 | _0.203_ | 0.314 | 0.316 | _0.25_ | **0.223** | _0.232_ | **0.229** | 0.261 | _0.232_ | 96 |
| Electricity | 0.484 | **0.269** | 0.31 | _0.289_ | 0.438 | **0.283** | 0.336 | _0.313_ | 0.687 | 0.846 | _0.384_ | **0.279** | |
| Traffic | 1.098 | **0.88** | 1.087 | _0.961_ | _0.963_ | **0.95** | 1.378 | 1.097 | 1.251 | 1.41 | _1.149_ | **0.869** | |
| Weather | 0.258 | **0.242** | _0.252_ | 0.257 | 0.263 | _0.259_ | **0.254** | 0.291 | **0.197** | _0.216_ | 0.26 | 0.292 | |
| Average | 0.59 | **0.394** | 0.446 | _0.413_ | 0.628 | 0.575 | _0.537_ | **0.462** | 0.607 | 0.635 | _0.517_ | **0.407** | |

| Model | TTM | | | | Timer | | | | Moment | | | | Lookback |
|---|---|---|---|---|---|---|---|---|---|---|---|---|---|
| Data Type | R | R+S(M) | S(M) | S | R | R+S(M) | S(M) | S | R | R+S(M) | S(M) | S | |
| ETTh1 | 0.499 | **0.436** | 0.537 | _0.474_ | 0.539 | **0.464** | 0.554 | _0.53_ | 0.607 | _0.601_ | 0.61 | **0.459** | |
| ETTh2 | 0.349 | **0.32** | 0.339 | _0.321_ | 0.379 | 0.375 | _0.369_ | **0.367** | 0.359 | _0.335_ | 0.508 | **0.331** | |
| ETTm1 | 1.026 | _0.404_ | 0.411 | **0.4** | 1.369 | 0.652 | _0.484_ | **0.417** | 1.327 | 0.79 | _0.53_ | **0.41** | |
| ETTm2 | 0.24 | **0.184** | 0.192 | **0.184** | 0.298 | 0.26 | _0.219_ | **0.212** | 0.297 | _0.238_ | 0.34 | **0.213** | 192 |
| Electricity | 0.31 | **0.24** | 0.304 | _0.28_ | 0.46 | **0.261** | _0.307_ | 0.318 | 0.595 | 0.635 | _0.371_ | **0.305** | |
| Traffic | 0.823 | **0.763** | 0.95 | _0.808_ | _0.813_ | **0.78** | 1.05 | 0.978 | 1.141 | 1.189 | _1.071_ | **0.793** | |
| Weather | 0.279 | **0.207** | _0.215_ | _0.215_ | 0.228 | **0.203** | _0.205_ | 0.224 | _0.219_ | **0.214** | 0.311 | 0.227 | |
| Average | 0.504 | **0.365** | 0.421 | _0.383_ | 0.584 | **0.428** | 0.455 | _0.435_ | 0.649 | 0.572 | _0.534_ | **0.391** | |

| Model | TTM | | | | Timer | | | | Moment | | | | Lookback |
|---|---|---|---|---|---|---|---|---|---|---|---|---|---|
| Data Type | R | R+S(M) | S(M) | S | R | R+S(M) | S(M) | S | R | R+S(M) | S(M) | S | |
| ETTh1 | _0.435_ | **0.415** | 0.53 | 0.497 | _0.504_ | **0.479** | 0.525 | 0.512 | 0.59 | _0.553_ | 0.763 | **0.439** | |
| ETTh2 | _0.312_ | **0.299** | 0.322 | 0.318 | _0.391_ | 0.396 | 0.393 | **0.389** | 0.382 | _0.364_ | 0.546 | **0.326** | |
| ETTm1 | 0.578 | **0.399** | 0.418 | 0.412 | 0.764 | _0.492_ | 0.493 | **0.434** | 0.546 | _0.464_ | 0.655 | **0.381** | |
| ETTm2 | 0.208 | **0.175** | 0.185 | _0.182_ | 0.26 | 0.234 | _0.219_ | **0.217** | 0.265 | _0.233_ | 0.331 | **0.206** | 336 |
| Electricity | _0.251_ | **0.234** | 0.306 | 0.289 | _0.293_ | **0.266** | 0.317 | 0.334 | 0.553 | _0.509_ | 0.576 | **0.281** | |
| Traffic | **0.707** | _0.714_ | 0.952 | 0.88 | **0.69** | _0.743_ | 0.897 | 0.901 | 1.083 | _1.051_ | 1.418 | **0.744** | |
| Weather | 0.235 | 0.187 | **0.183** | _0.184_ | 0.216 | **0.2** | 0.226 | _0.208_ | 0.231 | _0.211_ | 0.276 | **0.203** | |
| Average | _0.389_ | **0.346** | 0.414 | 0.395 | 0.445 | **0.401** | 0.439 | _0.428_ | 0.521 | _0.484_ | 0.652 | **0.369** | |

| Model | TTM | | | | Timer | | | | Moment | | | | Lookback |
|---|---|---|---|---|---|---|---|---|---|---|---|---|---|
| Data Type | R | R+S(M) | S(M) | S | R | R+S(M) | S(M) | S | R | R+S(M) | S(M) | S | |
| ETTh1 | _0.445_ | **0.424** | 0.52 | 0.491 | _0.481_ | **0.465** | 0.524 | _0.481_ | 0.556 | _0.499_ | 0.715 | **0.412** | |
| ETTh2 | _0.293_ | **0.281** | 0.324 | 0.317 | 0.389 | **0.378** | 0.386 | **0.378** | _0.353_ | _0.353_ | 0.507 | **0.328** | |
| ETTm1 | 0.703 | **0.413** | 0.44 | _0.434_ | 0.734 | _0.476_ | 0.516 | **0.456** | 0.565 | _0.443_ | 0.668 | **0.398** | |
| ETTm2 | 0.248 | **0.174** | 0.184 | _0.181_ | 0.252 | **0.219** | 0.22 | 0.222 | 0.256 | _0.228_ | 0.327 | **0.204** | 512 |
| Electricity | 0.3 | **0.253** | 0.306 | _0.293_ | _0.289_ | **0.251** | 0.307 | 0.317 | 0.491 | _0.419_ | 0.64 | **0.265** | |
| Traffic | _0.789_ | **0.762** | 0.916 | 0.906 | **0.716** | _0.743_ | 0.9 | 0.872 | 1.01 | _0.927_ | 1.34 | **0.726** | |
| Weather | 0.203 | **0.163** | _0.176_ | _0.176_ | 0.219 | **0.196** | 0.233 | _0.218_ | 0.218 | _0.201_ | 0.272 | **0.19** | |
| Average | 0.426 | **0.353** | 0.409 | _0.4_ | 0.44 | **0.39** | 0.441 | _0.421_ | 0.493 | _0.439_ | 0.638 | **0.36** | |

Table 5: The MSE performance of comparison of setups with sampling rate information. The models CycleNet (R), SparseTSF (R), and TTM (R) are trained on real data only, and offer architectural components that support periodic forecasting given the sampling rate. The models PatchTST (S) and TTM (S) are trained on synthetic data only. A lookback of 96 and a forecast horizon of 96 was used in this experiment

| Dataset | CycleNet (R) | SparseTSF (R) | TTM (R) | PatchTST (S) | TTM (S) |
|---|---|---|---|---|---|
| ETTh1 | 0.571 | 0.506 | 0.592 | **0.433** | _0.459_ |
| ETTh2 | _0.320_ | 0.329 | 0.357 | **0.312** | 0.322 |
| ETTm1 | 1.112 | 1.101 | 1.105 | **0.373** | _0.398_ |
| ETTm2 | 0.253 | 0.253 | 0.236 | **0.193** | _0.203_ |
| electricity | 0.422 | 0.405 | 0.484 | **0.287** | _0.289_ |
| traffic | 1.124 | _0.955_ | 1.098 | **0.901** | 0.961 |
| weather | _0.242_ | **0.209** | 0.258 | 0.394 | 0.257 |

It is important to highlight that TTM is used extensively in this study, always with its resolution prefix tuning (RPT) mechanism enabled to incorporate frequency-specific information. Similarly, in subsection 5.3 and Figure 3 of the main text, the TimesFM model (Das et al., 2024) also adopts a comparable frequency-aware strategy. These results underscore the robustness of Synth-FAR, showing that its benefits persist even when evaluated alongside architectures designed for seasonal pattern recognition.

# C   Supplementary method details

## C.1   Fundamental Frequency Estimation

The proposed Synth-FAR relies on prior information to mitigate potential frequency confusion and improve frequency generalization. In this section, we outline several practical approaches for obtaining the frequency information required by our method.

**Sampling rate.** The sampling interval $\Delta t$ provides direct information about the temporal spacing between consecutive observations and can be leveraged to infer plausible underlying periodicities. Specifically, if the signal is sampled every $\Delta t$ units, then a candidate period $T$ expressed in the same units corresponds to a normalized frequency

$f = \frac{1}{T}$.

Many real-world signals exhibit recurring natural or behavioral cycles such as daily, weekly, monthly, or yearly patterns. These common periods act as anchors for estimating the fundamental frequency. For example, for intraday signals sampled every $5, 10, 15, 30$, or $60$ minutes, a natural assumption is a daily periodicity. Since a day contains $24 \times 60 = 1440$ minutes, the corresponding periods in number of samples are

$T = \frac{1440}{\Delta t}$,

yielding normalized frequencies

$\frac{1}{288}, \frac{1}{144}, \frac{1}{96}, \frac{1}{48}, \frac{1}{24}$,

respectively. Likewise, for daily sampled data, common candidate periodicities include weekly or monthly cycles, corresponding to $f = 1/7$ or $f = 1/30$. In our experiments, we primarily use the weekly prior. This hierarchical reasoning can be extended whenever the sampling resolution naturally aligns with stronger macro-periodic structures. While this approach is intuitive and often effective, it may fail when the true periodicity is irregular or domain-specific (e.g., $f = 1/100$). Nevertheless, sampling rate remains a useful prior for coarse frequency estimation.

**Periodogram.** The periodogram is a classical tool for estimating the spectral density of a time series (Schuster, 1898). Given a sequence $x_0, x_1, \ldots, x_{N-1}$, its discrete Fourier transform (DFT) is

$X_k = \sum_{n=0}^{N-1} x_n e^{-i2\pi kn/N}, \quad k = 0, \ldots, N-1,$

and the periodogram is defined as

$P(f_k) = \frac{1}{N}|X_k|^2$.

The dominant peaks of $P(f_k)$ indicate strong periodic components. In many cases, after detrending the signal to remove low-frequency drift, the fundamental frequency can be estimated as the lowest significant peak, while harmonics appear at integer multiples $2f_0, 3f_0, \ldots$. Hence, the periodogram can provide not only an estimate of the fundamental frequency $f_0$, but also structural harmonic information that may further guide synthetic generation.

**Prior frequency information.** Another source of frequency estimation is direct prior knowledge. In many domains, periodicities are known a priori: electricity demand often follows daily and weekly cycles, traffic patterns may exhibit rush-hour periodicity, and retail demand may reflect weekly or seasonal trends. Formally, one may specify a candidate set of frequencies

$\mathcal{F} = \{f_1, f_2, \ldots, f_m\},$

based on domain expertise or prior spectral analysis, and select the most plausible candidate for Synth-FAR. Although such priors are not always immediately observable, combining domain knowledge with exploratory spectral analysis often leads to reliable frequency identification.

### C.2 Synth-FAR hyper-parameters and implementation details

In what follows, we complement the description in Sec. 4 and provide details and descriptions of the parameters and their selected values presented in Tab. 6.

To implement Synth-FAR, we follow the next steps: 1) Create three datasets, each corresponding to $h = 1, 2, 3$ with the given parameters in Tab. 6. To create each dataset, using the code in App. C.3 with the relevant parameters is recommended. 2) Each channel is standardized, according to the LTSF protocol (Nie et al., 2023; Wu et al., 2021; Zhou et al., 2021), however, this may not be required depending on the use case or the model, since many models include instance normalization (Nie et al., 2023) as part of their architectural pipeline. 3)

Table 6: Synth-FAR hyper-parameters.

| Notation | Selected Value | Description |
|---|---|---|
| $A'$ | 5 | The expected amplitude for $A_k$ generation. |
| $\bar{\omega}$ | Depends on dataset, see C.1 | The estimated fundamental frequency |
| $m_s$, $m_{ar}$ | 100 | Number of sine waves and AR($p$) processes for the pool $P$ |
| $h$ | $1, 2, 3$ | Determines the maximum number of harmonics |
| $l$ | $0.1 * |P|$ | The number of sine waves from $P$ used for the creation of $x^j$ |
| $n$ | $10,000, 1024^1$ | A fixed length for the all sine waves, AR($p$) processes, and $x$ |
| $d$ | 50 | the number of variates for $x$ |
| | $10,000$ | number of examples sampled from $X$ for training/validating |

$^1$ For Time-MMD we use $n = 1024$.

From each of the three comprised datasets, we draw $10,000$ samples only each of the lookback and horizon of interest, instead of using the entire dataset ($n*d$). Therefore, in practice we use only $10,000$ series from a single Synth-FAR datase, hence the final timestep size for training per-dataset is ($10,000 * (lookback + horizon)$). The reason we employ sampling is due to the same pattern repetition along the entire signal, concluding that the entire signal's length is not necessary.

### C.3 Synth-FAR implementation

Here, we provide the python implementation for a single Synth-FAR dataset generation in C.3, covering the two steps described in Sec. 4.3.

```python
import numpy as np

# step 1, create the pool of sine waves
def create_sine_pool(m, n, A_avg, harmonics, w_fund):
    set_fund = [w_fund*i for i in range(1, harmonics+1) if w_fund*i<0.5]
    P_s = []
    t = np.arange(n)
    for k in range(m):
        A = np.random.exponential(scale=A_avg-0.01) + 0.01  # to avoid 0
        w = np.random.choice(set_fund)
        phi = np.random.uniform(0, 2*np.pi)
        s_k = A * np.sin(2*np.pi*t*w + phi)
        P_s.append(s_k)
    return np.array(P_s)

# step 1, create the pool of AR($p$)
def create_ar_pool(m, n, std=1, p=8):
    p_vec = np.ones(p) / p  # AR coefficients 1/p
    P_ar = []
    for k in range(m):
        s_ar = np.zeros(n)
        for j in range(1, n):
            lookback = min(j, p)
            s_ar[j] = np.dot(s_ar[j-lookback:j], p_vec[:lookback])
            s_ar[j] += np.random.normal(0, std)
        P_ar.append(s_ar)
    return np.array(P_ar)

# create the signal step 2
def create_synth(P, var, p_frac):
    m,n = P.shape  # number of series
    l = int(m * p_frac)
    X = []  # Synth-FAR sample dataset
    for i in range(var):
        idx = np.random.choice(m, l)
        s_i = np.sum(P[idx], axis=0)
        X.append(s_i)
    return np.array(X)

A_avg = 5 # average amplitude
```

```
w_fund = 1/24 # fundamental frequency
harmonics = 3  # harmonics
m_s = 100 # sine wave pool size

p = 8 # AR order
std = 1 # standard deviation of the noise
m_ar = 100 # AR pool size

var = 50 # number of variates
n = 250 # signal length
p_frac = 0.1 # determines the size of l, as a fraction of the pool size

P_sine = create_sine_pool(m_s, n, A_avg, harmonics, w_fund)
P_ar = create_ar_pool(m_ar, n, std, p)
P = np.vstack((P_sine, P_ar))

X = create_synth(P, var, p_frac)
```

Table 7: Failure cases of Synth-FAR and corresponding mismatch types. The table reports forecasting performance where Synth-FAR underperforms a simple benchmark. *Low Seasonality Mismatch* refers to datasets with weak or diffuse periodic structure, where persistence-based forecasts are more effective. *Multivariate Mismatch* refers to datasets whose variates exhibit heterogeneous spectral patterns, making a single shared estimated frequency unsuitable across channels. Lower is better.

| Dataset | Synth-FAR | | | Benchmark | | Mismatch Type |
|---|---|---|---|---|---|---|
| | GPT4TS | PatchTST | TTM | Mean | Naive | |
| Agriculture | 0.198 | 0.179 | 0.183 | 0.353 | **0.158** | Low Seasonality Mismatch |
| Exchange | 0.094 | 0.092 | 0.089 | 0.139 | **0.081** | Low Seasonality Mismatch |
| Weather | 0.262 | 0.246 | 0.257 | **0.215** | 0.259 | Multivariate Mismatch |

### C.4 Synth-FAR limitations

Unfortunately, Synth-FAR is not uniformly effective across all zero-shot forecasting scenarios. Its main assumption is that the training signal contains a dominant and transferable periodic structure that can be approximated from the estimated fundamental frequency. When this assumption is violated, the generated synthetic series may not match the statistical behavior of the target data, which can reduce downstream forecasting performance.

In particular, datasets with a broad spectrum of relevant frequencies often exhibit weaker regular seasonality and a higher degree of stochasticity (Demirel & Holz, 2024). In such settings, the energy of the signal is distributed across multiple frequencies rather than concentrated around a single dominant mode. As a result, estimating one fundamental frequency—or even a small set of harmonics—is insufficient to faithfully represent the temporal dynamics. This suggests that richer variants such as mixed Synth-FAR, which combine multiple sampled frequencies, are more appropriate in these cases, as also indicated in Tab. 1.

Table 7 summarizes representative failure cases and reveals two recurring mismatch types:

**Low Seasonality Mismatch.** In datasets such as *Agriculture* and *Exchange*, Synth-FAR improves over simple baselines such as the mean forecast, but fails to outperform the naive baseline. For example, on Agriculture, Synth-FAR obtains 0.179–0.198 versus 0.353 for the mean predictor, yet the naive baseline achieves the best result with 0.158. Similarly, on Exchange, Synth-FAR reaches 0.089–0.094, while the naive forecast obtains 0.081. These results indicate that although some weak periodicity exists, it is not strong enough to justify a frequency-driven synthetic signal. Instead, local persistence and short-term continuity dominate, making naive carry-forward forecasts more competitive.

**Multivariate Mismatch.** The *Weather* dataset highlights a different limitation. Here, different variates exhibit substantially different spectral profiles, meaning that a single dataset-level estimated frequency does not transfer well across channels. Consequently, Synth-FAR underperforms the mean baseline (0.246–0.262 vs. 0.215). This suggests that when multivariate channels contain heterogeneous seasonalities, a shared synthetic generator can introduce frequency mismatch for many variables. In such cases, channel-specific or clustered frequency estimation may be required.

Figure 9 illustrates the different mismatch types through the periodograms of Agriculture, Exchange, and Weather. The red dashed line denotes the estimated fundamental frequency. In many cases, this estimated frequency does not align well with the dominant spectral characteristics of the data. For Agriculture and Exchange, the mismatch stems from weak or diffuse seasonality, where no clear dominant frequency exists (*low seasonality mismatch*). For Weather, the mismatch arises from substantial variability across different variates, whose spectral peaks differ significantly from one another (*multivariate mismatch*).

Overall, these failure cases emphasize that Synth-FAR is most effective when the target data contains coherent and sufficiently strong seasonal structure. When seasonality is weak, highly dispersed, or inconsistent across variates, simpler baselines may be preferable.

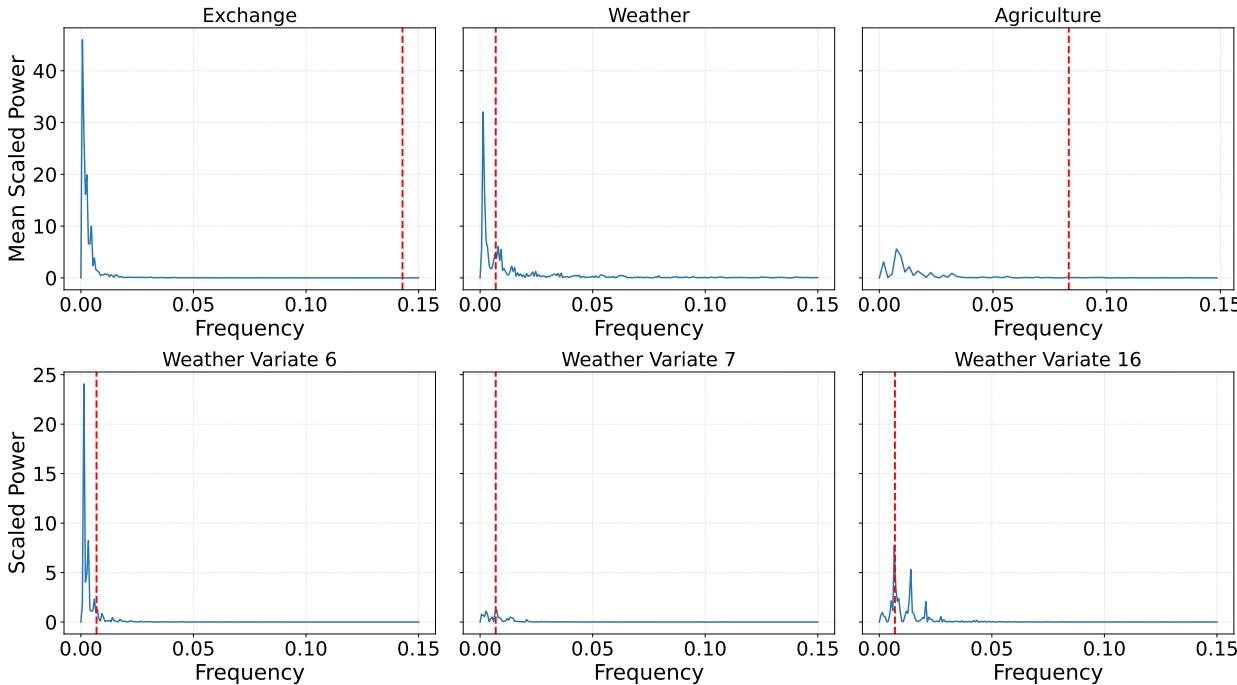

Figure 9: The scaled periodogram of failure cases of SynthFAR. The red line marks the estimated fundamental frequency. Top: the average periodogram frequency across variates of Exchange, Weather, and Agriculture. Bottom: the periodogram of different variates, showing that the variates have significantly different periodograms.

## D   Pre-trained foundation models limitations

In this section, we further expand the discussion about frequency generalization and frequency adaptation discussed in Sec. 4. Here, we consider pre-trained models, obtained from the original repositories of TimesFM (Das et al., 2024), Timer (Liu et al., 2024e), TTM (Ekambaram et al., 2024), and MOIRAI (Woo et al., 2024) with lookback values of 512 for TTM, TimesFM, MOIRAI, and 672 for Timer were utilized, to align with the specifications of the original trained models available online. To test whether the given models can generalize well, we evaluated their performance on simple periodic signals, with one to four harmonics of different frequencies. The results are depicted in Fig. 10, where the left plots detail the test MSE in log scale as a function of the frequency, and the right plots show examples of the evaluated signals. We find that models achieve reasonable errors on the $1/24$ frequency and its 2-harmonic frequency $1/12$, where the red dashed lines are positioned. This could be explained by the amount of pre-training data associated with the $1/24$ frequency, which often relates to an hourly sampling rate. For example, hourly sampled data accounts for $> 62\%$ of the pre-training datasets of TimesFM (Das et al., 2024). On the other hand, when evaluated on less common frequencies, we observe a significant performance degradation with MSE values getting closer to 1 (left, top) and even higher. This behavior becomes more apparent when the number of harmonics is greater than 1 (left, bottom). We also show the forecast predictions of individual signals for the over-fitted $1/24$ frequency in Fig. 11, and for the under-fitted $1/25$ frequency in Fig. 12. This analysis complements our frequency-based analysis above, suggesting that large time series forecasting models suffer from poor frequency adaptation and attain poor frequency generalization.

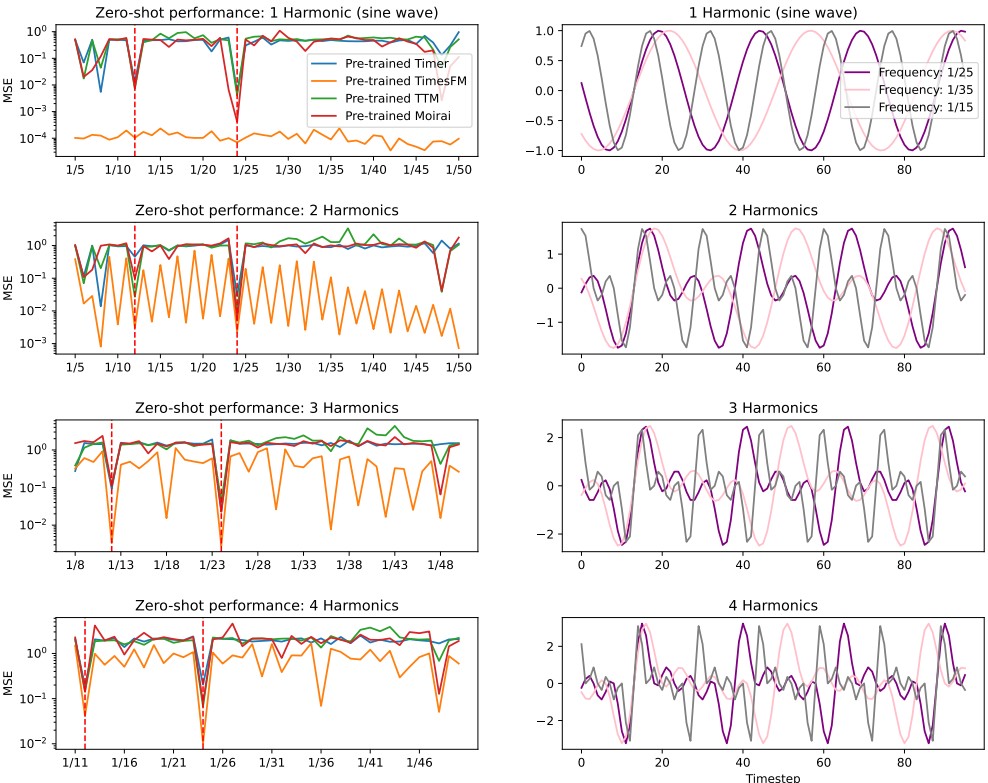

Figure 10: Zero-shot performance of pre-trained models on signals with 1 to 4 harmonics. The models perform well on the 1/24 and 1/12 frequencies, but for the remaining frequencies, the performance decreases significantly.

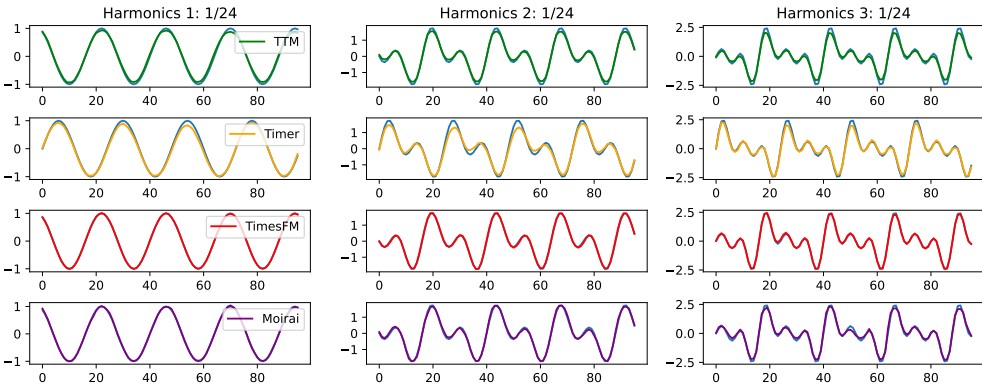

Figure 11: Forecast depiction of a 1/24 periodic series with different harmonics

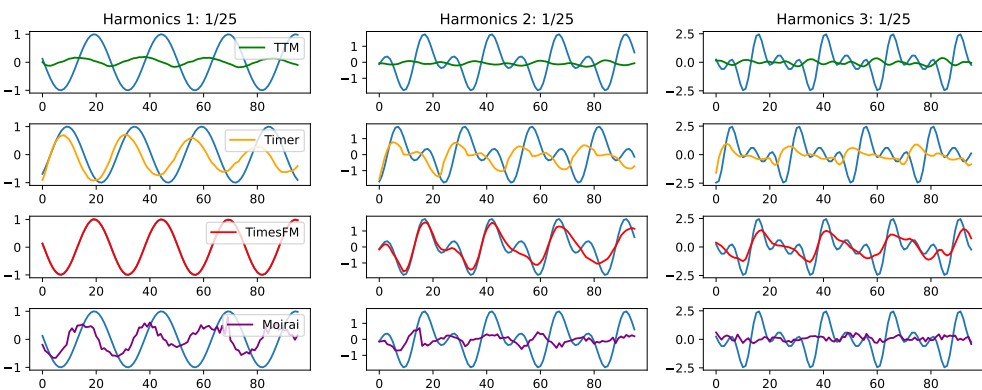

Figure 12: Forecast depiction of a 1/25 periodic series with different harmonics

# E  Extended experiments and results

In this section, we provide additional depictions and tables that expand the experiments in the main body. The Fig. 14 depicts the Pearson correlation coefficient (PCC) between every pair of datasets included in Fig. 1. Values closer to 1 represent a higher Periodogram similarity.

## E.1  Periodogram

The periodogram is often mention in this work as an effective tool for the analysis of a signal to extract the fundamental frequency. In Fig. 13, we provide examples of the periodograms of Exchange, Traffic and Electricity, where it is shown that the periodogram of Traffic and Electricity are very similar, with an identical fundamental frequency $1/24$, with apparent harmonics, suggesting that they are potential candidates for successful transfer learning. Traffic and Electricity are both mentioned in the high periodogram correlation category in Fig. 1, and also shown in the periodogram correlation matrix Fig. 14 with PCC $> 0.9$. On the other hand, Exchange has a wider spread of significant frequencies, e.g., $(0, 0.025]$, hence, being characterized with more randomness without clear dominant fundamental frequencies. Consequently, Synth-FAR fails to perform well on Exchange, since the estimated fundamental frequency does not represent the true spectral span of the data.

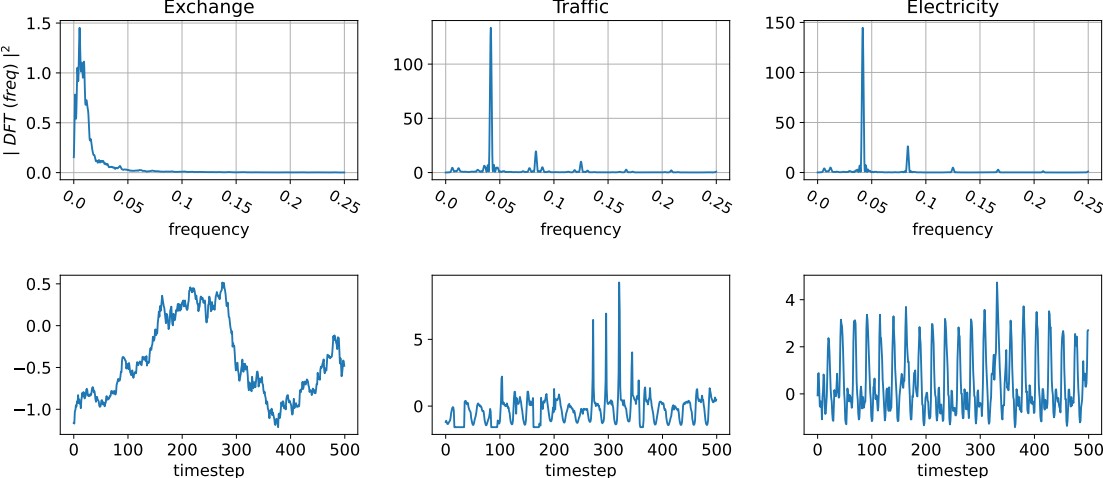

Figure 13: Top: Periodogram of the frequency range $(0, 0.25]$ for the datasets Exchange, Traffic, and Electricity visualized from left to right. Bottom: A random example from each dataset.

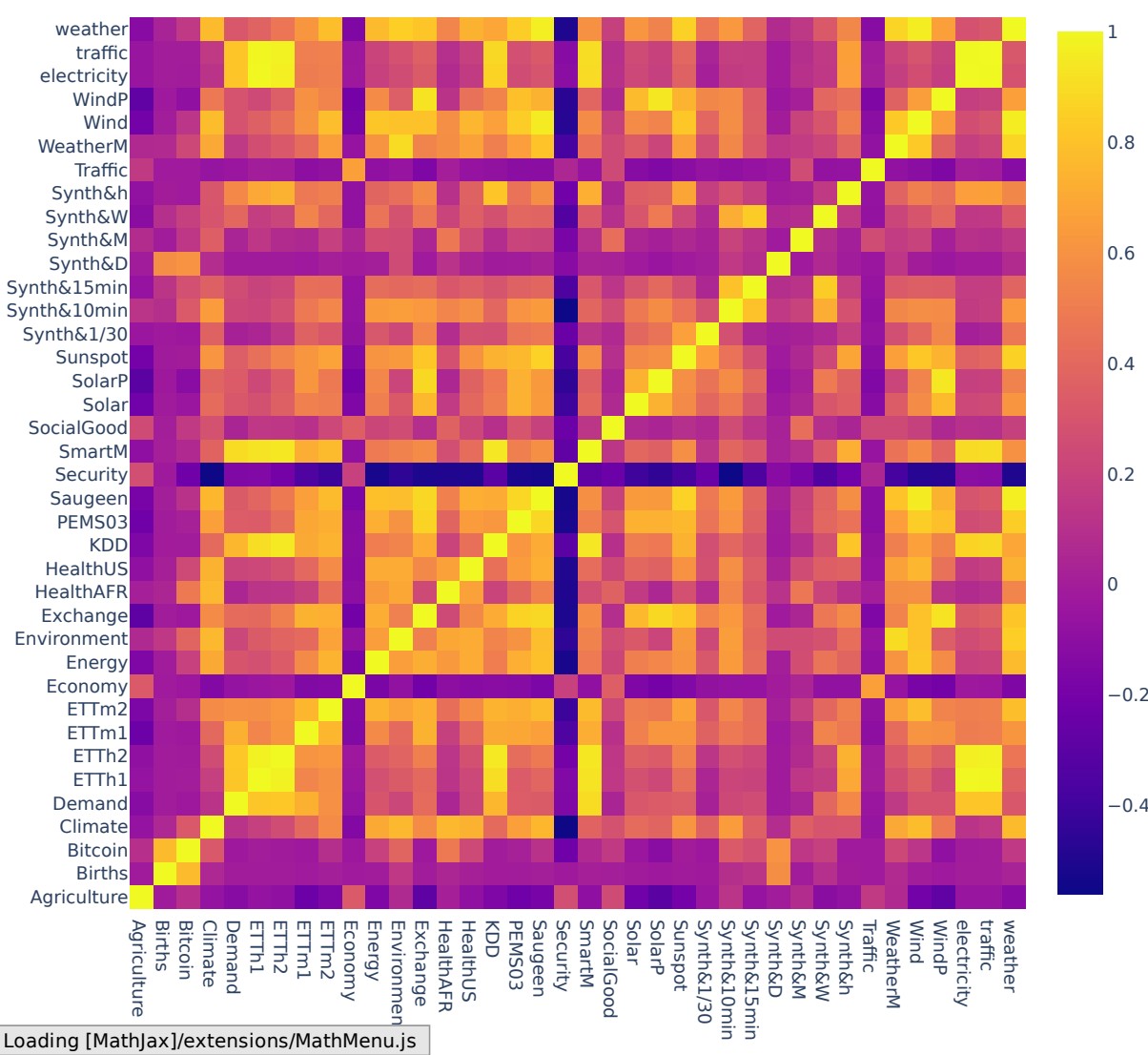

Figure 14: Pearson correlation score between the periodograms of each pair. Before computing the periodogram, a moving average is extracted and removed from the signal. Synth&X denotes a sinewave of a specific frequency X.

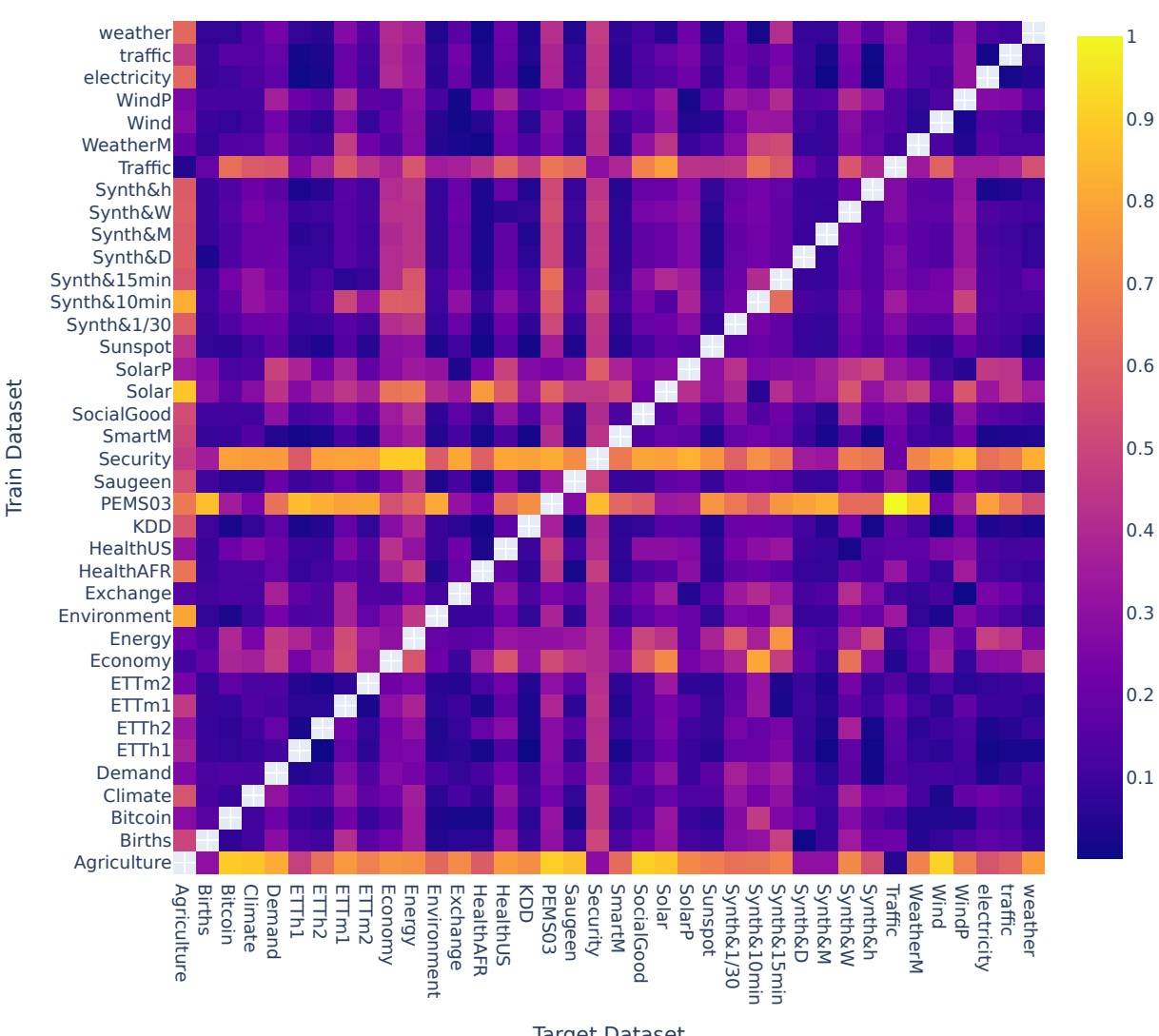

Figure 15: Scaled min-max MSE score between each train dataset and target datasets, the min-max is applied on each column s.t. each train dataset receives a score from 0 to 1. A score of 1 represents the best performer, which is in almost all cases given when the train and target dataset are the same, if the train and target are the same the MSE is removed (grey).

### E.2 Full Experimental Results

In this section, we provide the comprehensive results from the experiments discussed in the main text, including standard deviations for all metrics.

**Synthetic Comparisons**   Table 8 presents the expanded version of Table 1 with standard deviations included. For further clarity, Table 9 provides these results aggregated by model.

**Zero-Shot Evaluations**   In Table 11, we extend the results of Table 2 to represent all four horizons, reporting both the mean and standard deviation for each.

**MMD Benchmark**   Complementing Figure 3, Table 10 contains the full performance metrics and standard deviations for every model, dataset, and horizon evaluated in the MMD benchmark.

Table 8: Comparison between synthetic data methods with known (top) and unknown (bottom) sampling rates. Mean ± std over 3 runs.

### Known Sampling Rate

|  | Synth-FAR | | Ker-Synth | | FM | | PFN | |
|---|---|---|---|---|---|---|---|---|
|  | MSE | MAE | MSE | MAE | MSE | MAE | MSE | MAE |
| ETTh1 | 0.452 ± 0.007 | 0.439 ± 0.004 | 0.506 ± 0.016 | 0.460 ± 0.008 | 0.619 ± 0.018 | 0.529 ± 0.007 | 0.783 ± 0.020 | 0.578 ± 0.006 |
| ETTh2 | 0.327 ± 0.005 | 0.358 ± 0.003 | 0.345 ± 0.008 | 0.375 ± 0.005 | 0.330 ± 0.006 | 0.361 ± 0.003 | 0.841 ± 0.042 | 0.543 ± 0.010 |
| ETTm1 | 0.397 ± 0.005 | 0.393 ± 0.003 | 0.456 ± 0.035 | 0.403 ± 0.009 | 0.750 ± 0.041 | 0.539 ± 0.014 | 2.392 ± 0.176 | 0.962 ± 0.030 |
| ETTm2 | 0.209 ± 0.002 | 0.280 ± 0.002 | 0.213 ± 0.006 | 0.276 ± 0.003 | 0.206 ± 0.003 | 0.287 ± 0.004 | 0.482 ± 0.026 | 0.452 ± 0.010 |
| Electricity | 0.290 ± 0.006 | 0.357 ± 0.004 | 0.318 ± 0.014 | 0.381 ± 0.010 | 0.440 ± 0.010 | 0.478 ± 0.005 | 0.704 ± 0.020 | 0.497 ± 0.004 |
| Traffic | 0.941 ± 0.013 | 0.549 ± 0.007 | 1.022 ± 0.047 | 0.579 ± 0.017 | 1.203 ± 0.046 | 0.667 ± 0.016 | 1.767 ± 0.066 | 0.765 ± 0.009 |
| Weather | 0.272 ± 0.010 | 0.265 ± 0.003 | 0.223 ± 0.007 | 0.248 ± 0.004 | 0.233 ± 0.018 | 0.257 ± 0.010 | 0.573 ± 0.061 | 0.414 ± 0.020 |
| Average | 0.413 ± 0.007 | 0.377 ± 0.004 | 0.440 ± 0.019 | 0.389 ± 0.008 | 0.540 ± 0.020 | 0.445 ± 0.008 | 1.077 ± 0.059 | 0.602 ± 0.013 |

### Unknown Sampling Rate

|  | Synth-FAR | | Ker-Synth | | FM | | PFN | | Naive | | Mean | |
|---|---|---|---|---|---|---|---|---|---|---|---|---|
|  | MSE | MAE | MSE | MAE | MSE | MAE | MSE | MAE | MSE | MAE | MSE | MAE |
| ETTh1 | 0.529 ± 0.042 | 0.484 ± 0.023 | 0.555 ± 0.075 | 0.494 ± 0.038 | 1.045 ± 0.354 | 0.664 ± 0.113 | 0.710 ± 0.164 | 0.557 ± 0.072 | 1.294 | 0.713 | 0.701 | 0.558 |
| ETTh2 | 0.338 ± 0.023 | 0.374 ± 0.014 | 0.348 ± 0.019 | 0.380 ± 0.012 | 0.436 ± 0.095 | 0.425 ± 0.048 | 0.691 ± 0.196 | 0.495 ± 0.068 | 0.432 | 0.422 | 0.353 | 0.387 |
| ETTm1 | 0.576 ± 0.096 | 0.480 ± 0.045 | 0.688 ± 0.147 | 0.523 ± 0.053 | 2.705 ± 1.537 | 0.810 ± 0.170 | 2.378 ± 0.939 | 0.959 ± 0.179 | 1.214 | 0.665 | 0.693 | 0.548 |
| ETTm2 | 0.235 ± 0.020 | 0.306 ± 0.015 | 0.226 ± 0.016 | 0.302 ± 0.015 | 0.338 ± 0.071 | 0.361 ± 0.038 | 0.476 ± 0.160 | 0.450 ± 0.080 | 0.266 | 0.328 | 0.229 | 0.307 |
| Electricity | 0.330 ± 0.031 | 0.383 ± 0.032 | 0.471 ± 0.152 | 0.492 ± 0.104 | 1.154 ± 0.227 | 0.840 ± 0.077 | 0.580 ± 0.159 | 0.469 ± 0.067 | 1.588 | 0.945 | 0.846 | 0.762 |
| Traffic | 1.190 ± 0.119 | 0.650 ± 0.037 | 1.262 ± 0.133 | 0.709 ± 0.068 | 1.971 ± 0.493 | 0.971 ± 0.135 | 1.634 ± 0.244 | 0.765 ± 0.081 | 2.714 | 1.077 | 1.410 | 0.805 |
| Weather | 0.243 ± 0.014 | 0.269 ± 0.010 | 0.243 ± 0.021 | 0.275 ± 0.011 | 0.404 ± 0.170 | 0.332 ± 0.053 | 0.558 ± 0.172 | 0.409 ± 0.066 | 0.259 | 0.254 | 0.215 | 0.271 |
| Average | 0.492 ± 0.049 | 0.421 ± 0.025 | 0.542 ± 0.080 | 0.454 ± 0.043 | 1.150 ± 0.421 | 0.629 ± 0.091 | 1.004 ± 0.291 | 0.586 ± 0.088 | 1.110 | 0.629 | 0.635 | 0.520 |

Table 9: Full table of Tab. 1, the comparison between different synthetic data methods with the known target sampling rate (left block) and without it (right block). The model DLinear is also added in this table for reference, it was not included in Tab. 1

| | | Known Sampling Rate | | | | | | | | Unknown Sampling Rate | | | | | | | | | | | |
| | | Synth-FAR | | Ker-Synth | | FM | | PFN | | Synth-FAR | | Ker-Synth | | FM | | PFN | | Naive | | Mean | |
| | | MSE | MAE | MSE | MAE | MSE | MAE | MSE | MAE | MSE | MAE | MSE | MAE | MSE | MAE | MSE | MAE | MSE | MAE | MSE | MAE |
|---|---|---|---|---|---|---|---|---|---|---|---|---|---|---|---|---|---|---|---|---|---|
| Moment | ETTh1 | 0.434 | 0.423 | 0.714 | 0.567 | 0.691 | 0.570 | 1.117 | 0.728 | 0.547 | 0.501 | 0.708 | 0.572 | 1.797 | 0.913 | 1.014 | 0.697 | 1.294 | 0.713 | 0.701 | 0.558 |
| | ETTh2 | 0.323 | 0.361 | 0.396 | 0.412 | 0.359 | 0.380 | 1.033 | 0.627 | 0.382 | 0.400 | 0.380 | 0.402 | 0.626 | 0.527 | 1.025 | 0.626 | 0.432 | 0.422 | 0.353 | 0.387 |
| | ETTm1 | 0.422 | 0.404 | 0.461 | 0.437 | 0.578 | 0.501 | 3.778 | 1.280 | 0.637 | 0.524 | 0.977 | 0.631 | 2.781 | 1.044 | 3.778 | 1.280 | 1.214 | 0.665 | 0.693 | 0.548 |
| | ETTm2 | 0.232 | 0.290 | 0.201 | 0.276 | 0.212 | 0.287 | 0.780 | 0.605 | 0.261 | 0.324 | 0.257 | 0.334 | 0.372 | 0.419 | 0.780 | 0.605 | 0.266 | 0.328 | 0.229 | 0.307 |
| | Electricity | 0.279 | 0.357 | 0.482 | 0.507 | 0.541 | 0.541 | 1.039 | 0.716 | 0.384 | 0.448 | 0.753 | 0.675 | 1.599 | 0.992 | 0.895 | 0.610 | 1.588 | 0.945 | 0.846 | 0.762 |
| | Traffic | 0.869 | 0.508 | 1.421 | 0.767 | 1.318 | 0.752 | 2.443 | 1.060 | 1.149 | 0.653 | 1.531 | 0.840 | 3.073 | 1.272 | 2. | 0.924 | 2.714 | 1.077 | 1.410 | 0.805 |
| | Weather | 0.292 | 0.284 | 0.195 | 0.240 | 0.269 | 0.288 | 0.798 | 0.526 | 0.260 | 0.288 | 0.255 | 0.287 | 0.491 | 0.387 | 0.798 | 0.526 | 0.259 | 0.254 | 0.215 | 0.271 |
| | Average | 0.407 | 0.375 | 0.553 | 0.458 | 0.567 | 0.474 | 1.570 | 0.792 | 0.517 | 0.448 | 0.694 | 0.534 | 1.534 | 0.793 | 1.470 | 0.753 | 1.110 | 0.629 | 0.635 | 0.520 |
| GPT4TS | ETTh1 | 0.406 | 0.412 | 0.451 | 0.431 | 0.598 | 0.520 | 0.813 | 0.595 | 0.580 | 0.511 | 0.582 | 0.511 | 0.945 | 0.645 | 0.852 | 0.613 | 1.294 | 0.713 | 0.701 | 0.558 |
| | ETTh2 | 0.307 | 0.345 | 0.325 | 0.363 | 0.327 | 0.358 | 0.876 | 0.552 | 0.344 | 0.383 | 0.347 | 0.381 | 0.491 | 0.441 | 0.900 | 0.550 | 0.432 | 0.422 | 0.353 | 0.387 |
| | ETTm1 | 0.354 | 0.366 | 0.464 | 0.403 | 0.589 | 0.494 | 3.706 | 1.147 | 0.708 | 0.541 | 0.727 | 0.541 | 5.111 | 0.999 | 3.706 | 1.147 | 1.214 | 0.665 | 0.693 | 0.548 |
| | ETTm2 | 0.200 | 0.268 | 0.208 | 0.275 | 0.192 | 0.274 | 0.602 | 0.509 | 0.248 | 0.317 | 0.226 | 0.304 | 0.426 | 0.397 | 0.602 | 0.509 | 0.266 | 0.328 | 0.229 | 0.307 |
| | Electricity | 0.278 | 0.355 | 0.277 | 0.347 | 0.418 | 0.462 | 0.706 | 0.476 | 0.356 | 0.404 | 0.596 | 0.587 | 1.067 | 0.839 | 0.678 | 0.495 | 1.588 | 0.945 | 0.846 | 0.762 |
| | Traffic | 0.863 | 0.511 | 0.926 | 0.530 | 1.143 | 0.641 | 1.869 | 0.767 | 1.315 | 0.704 | 1.311 | 0.761 | 1.952 | 0.952 | 1.860 | 0.816 | 2.714 | 1.077 | 1.410 | 0.805 |
| | Weather | 0.262 | 0.258 | 0.225 | 0.250 | 0.244 | 0.266 | 0.796 | 0.469 | 0.242 | 0.270 | 0.248 | 0.279 | 0.752 | 0.426 | 0.796 | 0.469 | 0.259 | 0.254 | 0.215 | 0.271 |
| | Average | 0.381 | 0.359 | 0.411 | 0.371 | 0.502 | 0.431 | 1.338 | 0.645 | 0.542 | 0.447 | 0.577 | 0.481 | 1.535 | 0.671 | 1.342 | 0.657 | 1.110 | 0.629 | 0.635 | 0.520 |
| PatchTST | ETTh1 | 0.433 | 0.430 | 0.450 | 0.434 | 0.569 | 0.505 | 0.718 | 0.546 | 0.480 | 0.460 | 0.507 | 0.474 | 0.847 | 0.605 | 0.640 | 0.526 | 1.294 | 0.713 | 0.701 | 0.558 |
| | ETTh2 | 0.312 | 0.346 | 0.332 | 0.367 | 0.310 | 0.351 | 0.799 | 0.521 | 0.320 | 0.363 | 0.330 | 0.370 | 0.365 | 0.398 | 0.570 | 0.455 | 0.432 | 0.422 | 0.353 | 0.387 |
| | ETTm1 | 0.373 | 0.384 | 0.411 | 0.379 | 0.666 | 0.515 | 1.793 | 0.862 | 0.521 | 0.465 | 0.571 | 0.493 | 1.168 | 0.625 | 1.793 | 0.862 | 1.214 | 0.665 | 0.693 | 0.548 |
| | ETTm2 | 0.193 | 0.268 | 0.209 | 0.269 | 0.194 | 0.277 | 0.432 | 0.428 | 0.220 | 0.297 | 0.212 | 0.291 | 0.264 | 0.324 | 0.432 | 0.428 | 0.266 | 0.328 | 0.229 | 0.307 |
| | Electricity | 0.287 | 0.360 | 0.288 | 0.364 | 0.410 | 0.468 | 0.631 | 0.461 | 0.311 | 0.371 | 0.476 | 0.506 | 1.222 | 0.846 | 0.488 | 0.437 | 1.588 | 0.945 | 0.846 | 0.762 |
| | Traffic | 0.901 | 0.537 | 0.923 | 0.539 | 1.115 | 0.628 | 1.750 | 0.723 | 1.116 | 0.629 | 1.181 | 0.693 | 1.746 | 0.915 | 1.612 | 0.745 | 2.714 | 1.077 | 1.410 | 0.805 |
| | Weather | 0.246 | 0.255 | 0.221 | 0.247 | 0.216 | 0.247 | 0.508 | 0.389 | 0.228 | 0.264 | 0.219 | 0.264 | 0.315 | 0.302 | 0.508 | 0.389 | 0.259 | 0.254 | 0.215 | 0.271 |
| | Average | 0.392 | 0.369 | 0.405 | 0.371 | 0.497 | 0.427 | 0.947 | 0.561 | 0.457 | 0.407 | 0.499 | 0.442 | 0.847 | 0.574 | 0.863 | 0.549 | 1.110 | 0.629 | 0.635 | 0.520 |
| TimeMixer | ETTh1 | 0.473 | 0.450 | 0.505 | 0.457 | 0.661 | 0.541 | 0.704 | 0.537 | 0.527 | 0.476 | 0.534 | 0.477 | 0.825 | 0.598 | 0.638 | 0.513 | 1.294 | 0.713 | 0.701 | 0.558 |
| | ETTh2 | 0.326 | 0.356 | 0.341 | 0.373 | 0.348 | 0.368 | 0.819 | 0.530 | 0.324 | 0.365 | 0.345 | 0.376 | 0.372 | 0.394 | 0.544 | 0.446 | 0.432 | 0.422 | 0.353 | 0.387 |
| | ETTm1 | 0.422 | 0.406 | 0.428 | 0.393 | 1.146 | 0.638 | 1.832 | 0.841 | 0.595 | 0.473 | 0.565 | 0.475 | 3.091 | 0.804 | 1.842 | 0.843 | 1.214 | 0.665 | 0.693 | 0.548 |
| | ETTm2 | 0.211 | 0.283 | 0.211 | 0.273 | 0.232 | 0.310 | 0.344 | 0.384 | 0.242 | 0.311 | 0.225 | 0.297 | 0.363 | 0.356 | 0.345 | 0.384 | 0.266 | 0.328 | 0.229 | 0.307 |
| | Electricity | 0.294 | 0.356 | 0.309 | 0.372 | 0.443 | 0.474 | 0.644 | 0.454 | 0.313 | 0.359 | 0.373 | 0.417 | 0.930 | 0.755 | 0.519 | 0.436 | 1.588 | 0.945 | 0.846 | 0.762 |
| | Traffic | 0.986 | 0.572 | 1.016 | 0.569 | 1.314 | 0.686 | 1.575 | 0.690 | 1.223 | 0.643 | 1.247 | 0.675 | 1.703 | 0.886 | 1.528 | 0.707 | 2.714 | 1.077 | 1.410 | 0.805 |
| | Weather | 0.295 | 0.267 | 0.245 | 0.258 | 0.266 | 0.268 | 0.421 | 0.353 | 0.244 | 0.266 | 0.277 | 0.290 | 0.338 | 0.311 | 0.421 | 0.353 | 0.259 | 0.254 | 0.215 | 0.271 |
| | Average | 0.430 | 0.384 | 0.436 | 0.385 | 0.630 | 0.469 | 0.906 | 0.541 | 0.495 | 0.413 | 0.509 | 0.430 | 1.089 | 0.586 | 0.834 | 0.526 | 1.110 | 0.629 | 0.635 | 0.520 |
| UniTime | ETTh1 | 0.439 | 0.434 | 0.455 | 0.435 | 0.569 | 0.508 | 0.688 | 0.549 | 0.472 | 0.452 | 0.515 | 0.478 | 0.856 | 0.606 | 0.560 | 0.499 | 1.294 | 0.713 | 0.701 | 0.558 |
| | ETTh2 | 0.330 | 0.357 | 0.331 | 0.366 | 0.315 | 0.353 | 0.808 | 0.530 | 0.314 | 0.360 | 0.340 | 0.374 | 0.375 | 0.398 | 0.654 | 0.474 | 0.432 | 0.422 | 0.353 | 0.387 |
| | ETTm1 | 0.389 | 0.389 | 0.502 | 0.403 | 0.693 | 0.525 | 2.073 | 0.905 | 0.540 | 0.469 | 0.728 | 0.537 | 1.306 | 0.647 | 2.073 | 0.905 | 1.214 | 0.665 | 0.693 | 0.548 |
| | ETTm2 | 0.204 | 0.277 | 0.225 | 0.278 | 0.197 | 0.280 | 0.435 | 0.430 | 0.217 | 0.294 | 0.225 | 0.303 | 0.250 | 0.320 | 0.435 | 0.430 | 0.266 | 0.328 | 0.229 | 0.307 |
| | Electricity | 0.291 | 0.355 | 0.281 | 0.356 | 0.397 | 0.453 | 0.597 | 0.448 | 0.298 | 0.363 | 0.397 | 0.448 | 1.188 | 0.831 | 0.468 | 0.421 | 1.588 | 0.945 | 0.846 | 0.762 |
| | Traffic | 0.909 | 0.534 | 0.926 | 0.536 | 1.087 | 0.624 | 1.381 | 0.678 | 1.065 | 0.607 | 1.198 | 0.687 | 1.811 | 0.940 | 1.437 | 0.722 | 2.714 | 1.077 | 1.410 | 0.805 |
| | Weather | 0.263 | 0.264 | 0.226 | 0.251 | 0.199 | 0.236 | 0.532 | 0.401 | 0.223 | 0.262 | 0.236 | 0.272 | 0.264 | 0.286 | 0.532 | 0.401 | 0.259 | 0.254 | 0.215 | 0.271 |
| | Average | 0.404 | 0.373 | 0.421 | 0.375 | 0.494 | 0.426 | 0.931 | 0.563 | 0.447 | 0.401 | 0.520 | 0.443 | 0.864 | 0.575 | 0.880 | 0.550 | 1.110 | 0.629 | 0.635 | 0.520 |
| TTM | ETTh1 | 0.459 | 0.443 | 0.454 | 0.437 | 0.587 | 0.517 | 0.634 | 0.526 | 0.522 | 0.483 | 0.483 | 0.459 | 1.184 | 0.667 | 0.588 | 0.512 | 1.294 | 0.713 | 0.701 | 0.558 |
| | ETTh2 | 0.322 | 0.352 | 0.325 | 0.365 | 0.325 | 0.358 | 0.740 | 0.512 | 0.337 | 0.367 | 0.328 | 0.369 | 0.428 | 0.415 | 0.500 | 0.438 | 0.432 | 0.422 | 0.353 | 0.387 |
| | ETTm1 | 0.398 | 0.393 | 0.436 | 0.393 | 0.810 | 0.555 | 1.834 | 0.862 | 0.408 | 0.404 | 0.567 | 0.489 | 4.137 | 0.872 | 1.730 | 0.840 | 1.214 | 0.665 | 0.693 | 0.548 |
| | ETTm2 | 0.203 | 0.276 | 0.203 | 0.271 | 0.207 | 0.291 | 0.383 | 0.405 | 0.207 | 0.282 | 0.209 | 0.288 | 0.405 | 0.372 | 0.383 | 0.405 | 0.266 | 0.328 | 0.229 | 0.307 |
| | Electricity | 0.289 | 0.351 | 0.285 | 0.360 | 0.419 | 0.463 | 0.615 | 0.448 | 0.310 | 0.366 | 0.353 | 0.418 | 1.133 | 0.848 | 0.449 | 0.426 | 1.588 | 0.945 | 0.846 | 0.762 |
| | Traffic | 0.961 | 0.564 | 0.918 | 0.539 | 1.158 | 0.650 | 1.535 | 0.705 | 1.087 | 0.646 | 1.120 | 0.641 | 1.798 | 0.926 | 1.295 | 0.693 | 2.714 | 1.077 | 1.410 | 0.805 |
| | Weather | 0.257 | 0.257 | 0.212 | 0.241 | 0.224 | 0.250 | 0.474 | 0.379 | 0.252 | 0.257 | 0.216 | 0.261 | 0.372 | 0.312 | 0.368 | 0.341 | 0.259 | 0.254 | 0.215 | 0.271 |
| | Average | 0.413 | 0.377 | 0.405 | 0.372 | 0.533 | 0.441 | 0.888 | 0.548 | 0.446 | 0.396 | 0.468 | 0.418 | 1.351 | 0.630 | 0.753 | 0.520 | 1.110 | 0.629 | 0.635 | 0.520 |
| Timer | ETTh1 | 0.521 | 0.480 | 0.515 | 0.462 | 0.658 | 0.543 | 0.810 | 0.568 | 0.576 | 0.503 | 0.554 | 0.485 | 0.862 | 0.614 | 0.680 | 0.542 | 1.294 | 0.713 | 0.701 | 0.558 |
| | ETTh2 | 0.368 | 0.390 | 0.362 | 0.381 | 0.329 | 0.362 | 0.813 | 0.527 | 0.344 | 0.382 | 0.363 | 0.386 | 0.393 | 0.403 | 0.645 | 0.478 | 0.432 | 0.422 | 0.353 | 0.387 |
| | ETTm1 | 0.422 | 0.408 | 0.489 | 0.411 | 0.771 | 0.547 | 1.725 | 0.837 | 0.620 | 0.486 | 0.684 | 0.499 | 1.342 | 0.677 | 1.725 | 0.837 | 1.214 | 0.665 | 0.693 | 0.548 |
| | ETTm2 | 0.223 | 0.295 | 0.235 | 0.287 | 0.206 | 0.288 | 0.398 | 0.406 | 0.250 | 0.317 | 0.228 | 0.300 | 0.283 | 0.337 | 0.398 | 0.406 | 0.266 | 0.328 | 0.229 | 0.307 |
| | Electricity | 0.313 | 0.363 | 0.303 | 0.361 | 0.454 | 0.482 | 0.694 | 0.477 | 0.336 | 0.370 | 0.351 | 0.395 | 0.940 | 0.770 | 0.565 | 0.457 | 1.588 | 0.945 | 0.846 | 0.762 |
| | Traffic | 1.097 | 0.619 | 1.023 | 0.571 | 1.284 | 0.690 | 1.816 | 0.733 | 1.378 | 0.694 | 1.247 | 0.669 | 1.713 | 0.906 | 1.706 | 0.748 | 2.714 | 1.077 | 1.410 | 0.805 |
| | Weather | 0.291 | 0.269 | 0.234 | 0.250 | 0.214 | 0.243 | 0.484 | 0.382 | 0.254 | 0.271 | 0.248 | 0.276 | 0.299 | 0.298 | 0.484 | 0.382 | 0.259 | 0.254 | 0.215 | 0.271 |
| | Average | 0.462 | 0.403 | 0.452 | 0.389 | 0.559 | 0.451 | 0.963 | 0.561 | 0.537 | 0.432 | 0.525 | 0.430 | 0.833 | 0.572 | 0.886 | 0.550 | 1.110 | 0.629 | 0.635 | 0.520 |
| DLinear | ETTh1 | 0.470 | 0.449 | 0.531 | 0.516 | 0.674 | 0.575 | 0.739 | 0.555 | 0.661 | 0.546 | 0.673 | 0.598 | 1.234 | 0.803 | 0.717 | 0.586 | 1.294 | 0.713 | 0.701 | 0.558 |
| | ETTh2 | 0.329 | 0.360 | 0.871 | 0.664 | 0.588 | 0.536 | 0.764 | 0.531 | 0.370 | 0.401 | 1.130 | 0.774 | 2.265 | 1.076 | 0.719 | 0.529 | 0.432 | 0.422 | 0.353 | 0.387 |
| | ETTm1 | 0.433 | 0.414 | 0.445 | 0.449 | 0.611 | 0.547 | 3.136 | 1.141 | 0.616 | 0.519 | 0.796 | 0.632 | 1.154 | 0.768 | 3.136 | 1.141 | 1.214 | 0.665 | 0.693 | 0.548 |
| | ETTm2 | 0.216 | 0.294 | 0.355 | 0.416 | 0.524 | 0.512 | 0.584 | 0.524 | 0.254 | 0.329 | 1.062 | 0.758 | 2.215 | 1.054 | 0.584 | 0.524 | 0.266 | 0.328 | 0.229 | 0.307 |
| | Electricity | 0.301 | 0.363 | 0.304 | 0.390 | 0.462 | 0.490 | 0.628 | 0.461 | 0.381 | 0.431 | 0.638 | 0.639 | 1.357 | 0.929 | 0.594 | 0.481 | 1.588 | 0.945 | 0.846 | 0.762 |
| | Traffic | 0.981 | 0.572 | 0.929 | 0.536 | 1.290 | 0.684 | 1.801 | 0.750 | 1.640 | 0.783 | 1.260 | 0.752 | 1.976 | 0.971 | 1.523 | 0.775 | 2.714 | 1.077 | 1.410 | 0.805 |
| | Weather | 0.394 | 0.291 | 0.203 | 0.270 | 0.293 | 0.320 | 0.588 | 0.450 | 0.253 | 0.275 | 0.322 | 0.396 | 0.481 | 0.483 | 0.588 | 0.450 | 0.259 | 0.254 | 0.215 | 0.271 |
| | Average | 0.446 | 0.392 | 0.520 | 0.463 | 0.635 | 0.523 | 1.177 | 0.630 | 0.596 | 0.469 | 0.840 | 0.650 | 1.526 | 0.869 | 1.123 | 0.641 | 1.110 | 0.629 | 0.635 | 0.520 |

Table 10: The MSE performance across 10 Time-MMD datasets comparing pre-trained foundation models with in-domain models trained solely with Synth-FAR. The horizons 48, 60, 72, 84, and 96 were selected for this comparison.

| Horizon | Dataset | Synth-FAR Pretrain | | | | | Pretrained Foundation Models | | | | | | | | Benchmark | |
|---|---|---|---|---|---|---|---|---|---|---|---|---|---|---|---|---|
| | | GPT4TS | PatchTST | DLinear | TimeMixer | TTM | Tirex* | TimesFM-2.0* | Sundial* | Chronos-bolt* | Moirai* | Chronos* | TimesFM* | Timer* | Mean | Naive |
| | Agriculture | 0.155 ± 0.001 | 0.147 ± 0.007 | 0.156 ± 0.002 | 0.147 ± 0.002 | 0.146 ± 0.004 | 0.139 ± 0.000 | 0.140 ± 0.000 | 0.174 ± 0.000 | 0.147 ± 0.000 | 0.165 ± 0.000 | 0.145 ± 0.000 | 0.145 ± 0.000 | 0.176 ± 0.000 | 0.249 ± 0.000 | **0.136 ± 0.000** |
| | Climate | 0.932 ± 0.016 | 0.874 ± 0.013 | **0.763 ± 0.007** | 0.888 ± 0.011 | 0.892 ± 0.005 | 0.896 ± 0.000 | 1.232 ± 0.000 | 1.015 ± 0.000 | 1.015 ± 0.000 | 2.282 ± 0.000 | 0.791 ± 0.000 | 1.153 ± 0.000 | 1.003 ± 0.000 | 1.302 ± 0.000 | 0.855 ± 0.000 |
| | Economy | 0.190 ± 0.004 | 0.170 ± 0.009 | 0.149 ± 0.011 | 0.167 ± 0.003 | 0.158 ± 0.005 | 0.162 ± 0.000 | 0.158 ± 0.000 | 0.222 ± 0.000 | 0.150 ± 0.000 | 0.180 ± 0.000 | 0.186 ± 0.000 | 0.172 ± 0.000 | 0.296 ± 0.000 | 0.396 ± 0.000 | 0.159 ± 0.000 |
| | Energy | 0.227 ± 0.002 | 0.246 ± 0.003 | 0.230 ± 0.006 | 0.243 ± 0.004 | 0.254 ± 0.002 | **0.214 ± 0.000** | 0.342 ± 0.000 | 0.233 ± 0.000 | 0.278 ± 0.000 | 0.450 ± 0.000 | 0.217 ± 0.000 | 0.290 ± 0.000 | 0.238 ± 0.000 | 0.335 ± 0.000 | 0.223 ± 0.000 |
| | Environment | 0.786 ± 0.001 | 0.802 ± 0.010 | **0.782 ± 0.006** | 0.816 ± 0.008 | 0.826 ± 0.004 | 0.804 ± 0.000 | 0.852 ± 0.000 | 0.818 ± 0.000 | 0.845 ± 0.000 | 1.386 ± 0.000 | 0.942 ± 0.000 | 0.870 ± 0.000 | 0.900 ± 0.000 | 0.928 ± 0.000 | 1.214 ± 0.000 |
| 48 | HealthAFR | 0.759 ± 0.016 | 0.670 ± 0.061 | 1.080 ± 0.062 | 0.913 ± 0.018 | 0.735 ± 0.037 | **0.395 ± 0.000** | 0.418 ± 0.000 | 0.431 ± 0.000 | 0.401 ± 0.000 | 1.287 ± 0.000 | 0.602 ± 0.000 | 0.409 ± 0.000 | 0.463 ± 0.000 | 0.467 ± 0.000 | 1.425 ± 0.000 |
| | HealthUS | 0.464 ± 0.008 | **0.433 ± 0.009** | 0.529 ± 0.009 | 0.482 ± 0.001 | 0.472 ± 0.006 | 0.499 ± 0.000 | 0.478 ± 0.000 | 0.464 ± 0.000 | 0.688 ± 0.000 | 1.287 ± 0.000 | 0.601 ± 0.000 | 0.505 ± 0.000 | 0.587 ± 0.000 | 0.614 ± 0.000 | 1.031 ± 0.000 |
| | Security | 1.474 ± 0.001 | 1.463 ± 0.002 | 1.432 ± 0.008 | 1.464 ± 0.003 | 1.459 ± 0.001 | 1.574 ± 0.000 | 1.573 ± 0.000 | 1.517 ± 0.000 | 1.578 ± 0.000 | **1.425 ± 0.000** | 1.619 ± 0.000 | 1.542 ± 0.000 | 1.519 ± 0.000 | 1.524 ± 0.000 | 1.483 ± 0.000 |
| | SocialGood | 0.996 ± 0.016 | 0.937 ± 0.034 | 0.896 ± 0.028 | 0.959 ± 0.020 | 0.982 ± 0.020 | 1.037 ± 0.000 | **0.279 ± 0.000** | 1.160 ± 0.000 | 1.106 ± 0.000 | 1.147 ± 0.000 | 1.195 ± 0.000 | 0.483 ± 0.000 | 1.014 ± 0.000 | 1.242 ± 0.000 | 1.081 ± 0.000 |
| | Traffic | 0.082 ± 0.004 | 0.073 ± 0.001 | 0.069 ± 0.005 | 0.062 ± 0.005 | 0.066 ± 0.001 | 0.040 ± 0.000 | **0.031 ± 0.000** | 0.061 ± 0.000 | 0.040 ± 0.000 | 0.034 ± 0.000 | 0.039 ± 0.000 | 0.031 ± 0.000 | 0.107 ± 0.000 | 0.225 ± 0.000 | 0.152 ± 0.000 |
| | Average | 0.606 ± 0.007 | 0.582 ± 0.015 | 0.609 ± 0.014 | 0.614 ± 0.008 | 0.599 ± 0.008 | 0.576 ± 0.000 | **0.550 ± 0.000** | 0.609 ± 0.000 | 0.625 ± 0.000 | 0.964 ± 0.000 | 0.634 ± 0.000 | 0.560 ± 0.000 | 0.630 ± 0.000 | 0.728 ± 0.000 | 0.776 ± 0.000 |
| | Agriculture | 0.158 ± 0.001 | 0.151 ± 0.010 | 0.155 ± 0.003 | 0.150 ± 0.006 | 0.149 ± 0.004 | 0.143 ± 0.000 | **0.130 ± 0.000** | 0.186 ± 0.000 | 0.156 ± 0.000 | 0.192 ± 0.000 | 0.160 ± 0.000 | 0.148 ± 0.000 | 0.190 ± 0.000 | 0.265 ± 0.000 | 0.139 ± 0.000 |
| | Climate | 1.043 ± 0.012 | 0.996 ± 0.013 | **0.832 ± 0.009** | 0.999 ± 0.008 | 1.010 ± 0.009 | 1.032 ± 0.000 | 1.460 ± 0.000 | 1.168 ± 0.000 | 1.157 ± 0.000 | 2.315 ± 0.000 | 0.904 ± 0.000 | 1.418 ± 0.000 | 1.153 ± 0.000 | 1.386 ± 0.000 | 0.988 ± 0.000 |
| | Economy | 0.243 ± 0.001 | 0.222 ± 0.012 | **0.178 ± 0.006** | 0.218 ± 0.007 | 0.206 ± 0.006 | 0.210 ± 0.000 | 0.199 ± 0.000 | 0.299 ± 0.000 | 0.194 ± 0.000 | 0.272 ± 0.000 | 0.240 ± 0.000 | 0.233 ± 0.000 | 0.386 ± 0.000 | 0.465 ± 0.000 | 0.198 ± 0.000 |
| | Energy | 0.264 ± 0.002 | 0.271 ± 0.003 | 0.254 ± 0.007 | 0.271 ± 0.009 | 0.284 ± 0.002 | **0.247 ± 0.000** | 0.418 ± 0.000 | 0.273 ± 0.000 | 0.315 ± 0.000 | 0.916 ± 0.000 | 0.247 ± 0.000 | 0.349 ± 0.000 | 0.274 ± 0.000 | 0.363 ± 0.000 | 0.251 ± 0.000 |
| | Environment | 0.831 ± 0.001 | 0.847 ± 0.004 | **0.812 ± 0.006** | 0.872 ± 0.009 | 0.874 ± 0.006 | 0.840 ± 0.000 | 0.891 ± 0.000 | 0.866 ± 0.000 | 0.888 ± 0.000 | 1.470 ± 0.000 | 0.980 ± 0.000 | 0.909 ± 0.000 | 0.938 ± 0.000 | 0.962 ± 0.000 | 1.257 ± 0.000 |
| 60 | HealthAFR | 1.011 ± 0.003 | 0.651 ± 0.053 | 0.680 ± 0.030 | 0.863 ± 0.090 | 0.681 ± 0.027 | **0.324 ± 0.000** | 0.351 ± 0.000 | 0.352 ± 0.000 | 0.329 ± 0.000 | 1.394 ± 0.000 | 0.497 ± 0.000 | 0.340 ± 0.000 | 0.399 ± 0.000 | 0.395 ± 0.000 | 1.372 ± 0.000 |
| | HealthUS | 0.538 ± 0.005 | **0.441 ± 0.009** | 0.461 ± 0.004 | 0.484 ± 0.005 | 0.478 ± 0.003 | 0.507 ± 0.000 | 0.501 ± 0.000 | 0.471 ± 0.000 | 0.727 ± 0.000 | 1.461 ± 0.000 | 0.582 ± 0.000 | 0.520 ± 0.000 | 0.595 ± 0.000 | 0.623 ± 0.000 | 0.946 ± 0.000 |
| | Security | 1.305 ± 0.002 | 1.299 ± 0.001 | **1.260 ± 0.006** | 1.296 ± 0.006 | 1.294 ± 0.002 | 1.399 ± 0.000 | 1.403 ± 0.000 | 1.346 ± 0.000 | 1.398 ± 0.000 | 1.265 ± 0.000 | 1.419 ± 0.000 | 1.385 ± 0.000 | 1.350 ± 0.000 | 1.344 ± 0.000 | 1.348 ± 0.000 |
| | SocialGood | 1.077 ± 0.015 | 1.027 ± 0.028 | 0.967 ± 0.018 | 1.039 ± 0.015 | 1.078 ± 0.023 | 1.158 ± 0.000 | **0.268 ± 0.000** | 1.226 ± 0.000 | 1.233 ± 0.000 | 1.385 ± 0.000 | 1.251 ± 0.000 | 0.497 ± 0.000 | 1.134 ± 0.000 | 1.277 ± 0.000 | 1.168 ± 0.000 |
| | Traffic | 0.101 ± 0.003 | 0.091 ± 0.004 | 0.082 ± 0.002 | 0.080 ± 0.006 | 0.081 ± 0.002 | 0.053 ± 0.000 | **0.033 ± 0.000** | 0.089 ± 0.000 | 0.044 ± 0.000 | 0.043 ± 0.000 | 0.050 ± 0.000 | 0.033 ± 0.000 | 0.147 ± 0.000 | 0.256 ± 0.000 | 0.163 ± 0.000 |
| | Average | 0.657 ± 0.004 | 0.600 ± 0.014 | 0.568 ± 0.009 | 0.627 ± 0.016 | 0.614 ± 0.008 | 0.591 ± 0.000 | **0.565 ± 0.000** | 0.628 ± 0.000 | 0.644 ± 0.000 | 1.073 ± 0.000 | 0.633 ± 0.000 | 0.583 ± 0.000 | 0.657 ± 0.000 | 0.734 ± 0.000 | 0.783 ± 0.000 |
| | Agriculture | 0.168 ± 0.002 | 0.159 ± 0.015 | 0.160 ± 0.005 | 0.157 ± 0.010 | 0.155 ± 0.006 | 0.150 ± 0.000 | **0.130 ± 0.000** | 0.206 ± 0.000 | 0.164 ± 0.000 | 0.164 ± 0.000 | 0.189 ± 0.000 | 0.157 ± 0.000 | 0.208 ± 0.000 | 0.288 ± 0.000 | 0.144 ± 0.000 |
| | Climate | 1.145 ± 0.013 | 1.105 ± 0.019 | **0.887 ± 0.008** | 1.113 ± 0.008 | 1.118 ± 0.010 | 1.155 ± 0.000 | 1.698 ± 0.000 | 1.290 ± 0.000 | 1.286 ± 0.000 | 5.115 ± 0.000 | 0.984 ± 0.000 | 1.701 ± 0.000 | 1.268 ± 0.000 | 1.456 ± 0.000 | 1.111 ± 0.000 |
| | Economy | 0.304 ± 0.001 | 0.284 ± 0.022 | **0.216 ± 0.008** | 0.271 ± 0.003 | 0.262 ± 0.006 | 0.265 ± 0.000 | 0.244 ± 0.000 | 0.387 ± 0.000 | 0.243 ± 0.000 | 0.284 ± 0.000 | 0.297 ± 0.000 | 0.299 ± 0.000 | 0.474 ± 0.000 | 0.542 ± 0.000 | 0.242 ± 0.000 |
| | Energy | 0.302 ± 0.004 | 0.303 ± 0.009 | 0.285 ± 0.007 | 0.300 ± 0.005 | 0.317 ± 0.002 | **0.281 ± 0.000** | 0.494 ± 0.000 | 0.310 ± 0.000 | 0.350 ± 0.000 | 0.547 ± 0.000 | 0.281 ± 0.000 | 0.407 ± 0.000 | 0.308 ± 0.000 | 0.390 ± 0.000 | 0.285 ± 0.000 |
| | Environment | 0.876 ± 0.001 | 0.893 ± 0.001 | **0.842 ± 0.003** | 0.919 ± 0.012 | 0.937 ± 0.003 | 0.872 ± 0.000 | 0.925 ± 0.000 | 0.908 ± 0.000 | 0.927 ± 0.000 | 3.077 ± 0.000 | 1.020 ± 0.000 | 0.943 ± 0.000 | 0.972 ± 0.000 | 0.992 ± 0.000 | 1.307 ± 0.000 |
| 72 | HealthAFR | 0.961 ± 0.025 | 0.616 ± 0.103 | 0.629 ± 0.050 | 0.837 ± 0.020 | 0.635 ± 0.051 | **0.278 ± 0.000** | 0.306 ± 0.000 | 0.307 ± 0.000 | 0.282 ± 0.000 | 1.911 ± 0.000 | 0.452 ± 0.000 | 0.293 ± 0.000 | 0.358 ± 0.000 | 0.347 ± 0.000 | 1.345 ± 0.000 |
| | HealthUS | 0.560 ± 0.008 | **0.461 ± 0.009** | 0.473 ± 0.005 | 0.500 ± 0.007 | 0.495 ± 0.008 | 0.528 ± 0.000 | 0.538 ± 0.000 | 0.483 ± 0.000 | 0.758 ± 0.000 | 1.459 ± 0.000 | 0.592 ± 0.000 | 0.543 ± 0.000 | 0.604 ± 0.000 | 0.632 ± 0.000 | 0.970 ± 0.000 |
| | Security | 1.213 ± 0.003 | 1.212 ± 0.003 | **1.166 ± 0.002** | 1.209 ± 0.004 | 1.213 ± 0.004 | 1.279 ± 0.000 | 1.300 ± 0.000 | 1.242 ± 0.000 | 1.281 ± 0.000 | 1.173 ± 0.000 | 1.290 ± 0.000 | 1.293 ± 0.000 | 1.244 ± 0.000 | 1.238 ± 0.000 | 1.260 ± 0.000 |
| | SocialGood | 1.147 ± 0.014 | 1.106 ± 0.039 | 0.998 ± 0.017 | 1.135 ± 0.018 | 1.168 ± 0.022 | 1.235 ± 0.000 | **0.257 ± 0.000** | 1.283 ± 0.000 | 1.328 ± 0.000 | 3.188 ± 0.000 | 1.381 ± 0.000 | 0.545 ± 0.000 | 1.247 ± 0.000 | 1.306 ± 0.000 | 1.273 ± 0.000 |
| | Traffic | 0.124 ± 0.003 | 0.111 ± 0.006 | 0.100 ± 0.007 | 0.097 ± 0.005 | 0.101 ± 0.003 | 0.071 ± 0.000 | **0.034 ± 0.000** | 0.130 ± 0.000 | 0.048 ± 0.000 | 0.054 ± 0.000 | 0.071 ± 0.000 | 0.036 ± 0.000 | 0.183 ± 0.000 | 0.290 ± 0.000 | 0.178 ± 0.000 |
| | Average | 0.680 ± 0.007 | 0.625 ± 0.023 | **0.576 ± 0.011** | 0.654 ± 0.009 | 0.640 ± 0.012 | 0.611 ± 0.000 | 0.593 ± 0.000 | 0.655 ± 0.000 | 0.667 ± 0.000 | 1.697 ± 0.000 | 0.656 ± 0.000 | 0.622 ± 0.000 | 0.687 ± 0.000 | 0.748 ± 0.000 | 0.812 ± 0.000 |
| | Agriculture | 0.177 ± 0.002 | 0.168 ± 0.014 | 0.164 ± 0.004 | 0.166 ± 0.005 | 0.163 ± 0.007 | 0.154 ± 0.000 | **0.132 ± 0.000** | 0.235 ± 0.000 | 0.168 ± 0.000 | 0.199 ± 0.000 | 0.199 ± 0.000 | 0.168 ± 0.000 | 0.235 ± 0.000 | 0.317 ± 0.000 | 0.150 ± 0.000 |
| | Climate | 1.245 ± 0.011 | 1.205 ± 0.020 | **0.931 ± 0.001** | 1.216 ± 0.009 | 1.219 ± 0.014 | 1.267 ± 0.000 | 1.954 ± 0.000 | 1.383 ± 0.000 | 1.404 ± 0.000 | 57.706 ± 0.000 | 1.046 ± 0.000 | 2.000 ± 0.000 | 1.354 ± 0.000 | 1.514 ± 0.000 | 1.229 ± 0.000 |
| | Economy | 0.369 ± 0.005 | 0.349 ± 0.022 | **0.256 ± 0.007** | 0.334 ± 0.010 | 0.324 ± 0.009 | 0.330 ± 0.000 | 0.291 ± 0.000 | 0.483 ± 0.000 | 0.303 ± 0.000 | 0.360 ± 0.000 | 0.366 ± 0.000 | 0.370 ± 0.000 | 0.566 ± 0.000 | 0.621 ± 0.000 | 0.292 ± 0.000 |
| | Energy | 0.315 ± 0.004 | 0.334 ± 0.003 | 0.339 ± 0.005 | 0.325 ± 0.004 | 0.348 ± 0.009 | 0.316 ± 0.000 | 0.581 ± 0.000 | 0.344 ± 0.000 | 0.387 ± 0.000 | 0.661 ± 0.000 | **0.314 ± 0.000** | 0.452 ± 0.000 | 0.340 ± 0.000 | 0.415 ± 0.000 | 0.323 ± 0.000 |
| | Environment | 0.920 ± 0.001 | 0.940 ± 0.017 | **0.866 ± 0.006** | 0.947 ± 0.005 | 0.989 ± 0.005 | 0.899 ± 0.000 | 0.955 ± 0.000 | 0.946 ± 0.000 | 0.961 ± 0.000 | 3.539 ± 0.000 | 1.055 ± 0.000 | 0.971 ± 0.000 | 1.001 ± 0.000 | 1.018 ± 0.000 | 1.354 ± 0.000 |
| 84 | HealthAFR | 0.619 ± 0.021 | 0.565 ± 0.064 | 0.916 ± 0.035 | 0.813 ± 0.058 | 0.600 ± 0.105 | **0.245 ± 0.000** | 0.274 ± 0.000 | 0.286 ± 0.000 | 0.248 ± 0.000 | 0.670 ± 0.000 | 0.480 ± 0.000 | 0.260 ± 0.000 | 0.351 ± 0.000 | 0.313 ± 0.000 | 1.332 ± 0.000 |
| | HealthUS | 0.486 ± 0.002 | **0.472 ± 0.007** | 0.573 ± 0.003 | 0.515 ± 0.008 | 0.501 ± 0.013 | 0.542 ± 0.000 | 0.572 ± 0.000 | 0.484 ± 0.000 | 0.772 ± 0.000 | 1.570 ± 0.000 | 0.600 ± 0.000 | 0.557 ± 0.000 | 0.607 ± 0.000 | 0.622 ± 0.000 | 0.968 ± 0.000 |
| | Security | 1.142 ± 0.003 | 1.149 ± 0.004 | 1.098 ± 0.004 | 1.149 ± 0.006 | 1.147 ± 0.003 | 1.199 ± 0.000 | 1.238 ± 0.000 | 1.175 ± 0.000 | 1.206 ± 0.000 | **1.088 ± 0.000** | 1.212 ± 0.000 | 1.228 ± 0.000 | 1.174 ± 0.000 | 1.171 ± 0.000 | 1.222 ± 0.000 |
| | SocialGood | 1.208 ± 0.014 | 1.183 ± 0.040 | 1.018 ± 0.017 | 1.208 ± 0.013 | 1.235 ± 0.021 | 1.285 ± 0.000 | **0.255 ± 0.000** | 1.326 ± 0.000 | 1.408 ± 0.000 | 2.026 ± 0.000 | 1.473 ± 0.000 | 0.584 ± 0.000 | 1.325 ± 0.000 | 1.322 ± 0.000 | 1.362 ± 0.000 |
| | Traffic | 0.149 ± 0.004 | 0.138 ± 0.008 | 0.121 ± 0.006 | 0.123 ± 0.009 | 0.126 ± 0.008 | 0.093 ± 0.000 | **0.036 ± 0.000** | 0.181 ± 0.000 | 0.054 ± 0.000 | 0.077 ± 0.000 | 0.089 ± 0.000 | 0.041 ± 0.000 | 0.228 ± 0.000 | 0.327 ± 0.000 | 0.197 ± 0.000 |
| | Average | 0.663 ± 0.007 | 0.650 ± 0.020 | **0.628 ± 0.009** | 0.680 ± 0.013 | 0.665 ± 0.019 | 0.633 ± 0.000 | 0.629 ± 0.000 | 0.684 ± 0.000 | 0.691 ± 0.000 | 6.790 ± 0.000 | 0.683 ± 0.000 | 0.663 ± 0.000 | 0.718 ± 0.000 | 0.764 ± 0.000 | 0.843 ± 0.000 |
| | Agriculture | 0.198 ± 0.004 | 0.179 ± 0.010 | 0.170 ± 0.008 | 0.180 ± 0.009 | 0.183 ± 0.009 | 0.160 ± 0.000 | **0.132 ± 0.000** | 0.268 ± 0.000 | 0.171 ± 0.000 | 0.252 ± 0.000 | 0.212 ± 0.000 | 0.180 ± 0.000 | 0.268 ± 0.000 | 0.353 ± 0.000 | 0.158 ± 0.000 |
| | Climate | 1.338 ± 0.011 | 1.314 ± 0.008 | **0.978 ± 0.010** | 1.317 ± 0.013 | 1.313 ± 0.018 | 1.361 ± 0.000 | 2.240 ± 0.000 | 1.447 ± 0.000 | 1.497 ± 0.000 | 3.969 ± 0.000 | 1.103 ± 0.000 | 2.335 ± 0.000 | 1.418 ± 0.000 | 1.564 ± 0.000 | 1.348 ± 0.000 |
| | Economy | 0.442 ± 0.007 | 0.414 ± 0.030 | **0.293 ± 0.011** | 0.406 ± 0.018 | 0.400 ± 0.009 | 0.407 ± 0.000 | 0.340 ± 0.000 | 0.584 ± 0.000 | 0.369 ± 0.000 | 0.414 ± 0.000 | 0.447 ± 0.000 | 0.454 ± 0.000 | 0.658 ± 0.000 | 0.705 ± 0.000 | 0.349 ± 0.000 |
| | Energy | **0.342 ± 0.005** | 0.361 ± 0.004 | 0.384 ± 0.008 | 0.359 ± 0.005 | 0.378 ± 0.005 | 0.348 ± 0.000 | 0.647 ± 0.000 | 0.374 ± 0.000 | 0.423 ± 0.000 | 0.749 ± 0.000 | 0.345 ± 0.000 | 0.489 ± 0.000 | 0.370 ± 0.000 | 0.439 ± 0.000 | 0.354 ± 0.000 |
| | Environment | 0.958 ± 0.005 | 0.971 ± 0.020 | **0.891 ± 0.010** | 0.992 ± 0.033 | 1.043 ± 0.013 | 0.921 ± 0.000 | 0.979 ± 0.000 | 0.978 ± 0.000 | 0.989 ± 0.000 | 2.060 ± 0.000 | 1.083 ± 0.000 | 0.994 ± 0.000 | 1.025 ± 0.000 | 1.041 ± 0.000 | 1.399 ± 0.000 |
| 96 | HealthAFR | 0.610 ± 0.008 | 0.527 ± 0.028 | 0.857 ± 0.031 | 0.726 ± 0.026 | 0.576 ± 0.059 | **0.219 ± 0.000** | 0.250 ± 0.000 | 0.280 ± 0.000 | 0.223 ± 0.000 | 2.795 ± 0.000 | 0.377 ± 0.000 | 0.236 ± 0.000 | 0.327 ± 0.000 | 0.288 ± 0.000 | 1.325 ± 0.000 |
| | HealthUS | 0.485 ± 0.008 | **0.467 ± 0.007** | 0.573 ± 0.002 | 0.496 ± 0.010 | 0.510 ± 0.011 | 0.534 ± 0.000 | 0.586 ± 0.000 | 0.473 ± 0.000 | 0.760 ± 0.000 | 2.345 ± 0.000 | 0.590 ± 0.000 | 0.553 ± 0.000 | 0.597 ± 0.000 | 0.608 ± 0.000 | 0.965 ± 0.000 |
| | Security | 1.103 ± 0.001 | 1.101 ± 0.003 | 1.047 ± 0.006 | 1.100 ± 0.004 | 1.101 ± 0.004 | 1.159 ± 0.000 | 1.206 ± 0.000 | 1.141 ± 0.000 | 1.168 ± 0.000 | **0.999 ± 0.000** | 1.173 ± 0.000 | 1.192 ± 0.000 | 1.136 ± 0.000 | 1.137 ± 0.000 | 1.160 ± 0.000 |
| | SocialGood | 1.256 ± 0.014 | 1.260 ± 0.029 | 1.049 ± 0.008 | 1.264 ± 0.013 | 1.280 ± 0.026 | 1.326 ± 0.000 | **0.266 ± 0.000** | 1.331 ± 0.000 | 1.466 ± 0.000 | 1.844 ± 0.000 | 1.555 ± 0.000 | 0.610 ± 0.000 | 1.353 ± 0.000 | 1.324 ± 0.000 | 1.439 ± 0.000 |
| | Traffic | 0.179 ± 0.006 | 0.164 ± 0.007 | 0.140 ± 0.012 | 0.148 ± 0.006 | 0.159 ± 0.009 | 0.119 ± 0.000 | **0.035 ± 0.000** | 0.235 ± 0.000 | 0.061 ± 0.000 | 0.059 ± 0.000 | 0.122 ± 0.000 | 0.046 ± 0.000 | 0.265 ± 0.000 | 0.368 ± 0.000 | 0.219 ± 0.000 |
| | Average | 0.691 ± 0.007 | 0.676 ± 0.015 | **0.638 ± 0.011** | 0.699 ± 0.014 | 0.694 ± 0.016 | 0.655 ± 0.000 | 0.668 ± 0.000 | 0.711 ± 0.000 | 0.713 ± 0.000 | 1.549 ± 0.000 | 0.701 ± 0.000 | 0.709 ± 0.000 | 0.742 ± 0.000 | 0.783 ± 0.000 | 0.872 ± 0.000 |

Table 11: A full comparison of zero-shot forecasting with standard deviation when training with real data vs. synthetic data. Including Synth-FAR outperforms real data in most cases, with a notable advantage to Synth-FAR with the sampling rate denoted (**S**). The Lowest MSE is each block is marked with bold, and second lowest with an underline.

| Model Data Type dataset horizon | TTM R | R+S(M) | S(M) | S | Timer R | R+S(M) | S(M) | S | Moment R | R+S(M) | S(M) | S |
|---|---|---|---|---|---|---|---|---|---|---|---|---|
| ETTh1 96 | 0.592±0.080 | **0.438±0.003** | 0.522±0.013 | 0.459±0.001 | 0.660±0.025 | **0.521±0.011** | 0.576±0.008 | 0.521±0.011 | 0.640±0.001 | 0.701±0.000 | 0.547±0.006 | **0.434±0.016** |
| 192 | 0.644±0.047 | **0.488±0.004** | 0.559±0.008 | 0.513±0.002 | 0.744±0.029 | 0.583±0.006 | 0.634±0.008 | **0.572±0.004** | 0.675±0.000 | 0.718±0.000 | **0.554±0.006** | 0.582±0.004 |
| 336 | 0.631±0.056 | **0.529±0.010** | 0.605±0.006 | 0.562±0.001 | 0.750±0.016 | 0.644±0.019 | 0.682±0.006 | **0.616±0.005** | 0.692±0.000 | 0.723±0.000 | 0.581±0.008 | 0.595±0.007 |
| 720 | 0.675±0.099 | **0.532±0.011** | 0.589±0.010 | 0.548±0.007 | 0.736±0.052 | 0.664±0.048 | 0.662±0.009 | **0.591±0.015** | 0.712±0.001 | 0.712±0.000 | **0.531±0.006** | 0.534±0.003 |
| ETTh2 96 | 0.357±0.028 | 0.324±0.006 | 0.337±0.005 | **0.322±0.001** | 0.520±0.006 | 0.531±0.008 | **0.344±0.004** | 0.368±0.020 | 0.320±0.000 | 0.353±0.000 | 0.382±0.003 | 0.323±0.002 |
| 192 | 0.449±0.007 | **0.407±0.005** | 0.427±0.004 | 0.414±0.003 | 0.648±0.015 | 0.642±0.026 | **0.451±0.001** | 0.457±0.006 | 0.413±0.000 | 0.428±0.000 | 0.437±0.003 | 0.442±0.002 |
| 336 | 0.474±0.015 | **0.452±0.006** | 0.482±0.004 | 0.471±0.003 | 0.602±0.006 | 0.600±0.011 | 0.513±0.003 | **0.503±0.002** | 0.450±0.000 | 0.454±0.000 | 0.471±0.003 | 0.480±0.001 |
| 720 | 0.473±0.019 | **0.451±0.002** | 0.475±0.001 | 0.465±0.004 | 0.635±0.005 | 0.644±0.007 | 0.512±0.001 | **0.480±0.003** | 0.451±0.000 | 0.451±0.000 | 0.455±0.002 | 0.451±0.003 |
| ETTm1 96 | 1.105±0.119 | 0.404±0.014 | 0.408±0.004 | **0.398±0.001** | 1.239±0.019 | 1.165±0.043 | 0.620±0.078 | **0.422±0.004** | 0.922±0.001 | 0.693±0.000 | 0.637±0.060 | **0.422±0.018** |
| 192 | 1.014±0.053 | 0.455±0.010 | **0.450±0.005** | 0.456±0.004 | 1.301±0.013 | 1.199±0.115 | 0.686±0.072 | **0.475±0.002** | 0.907±0.001 | 0.710±0.000 | 0.685±0.031 | **0.444±0.015** |
| 336 | 0.993±0.048 | **0.477±0.005** | 0.484±0.003 | 0.491±0.008 | 1.240±0.013 | 1.153±0.016 | 0.686±0.059 | **0.495±0.009** | 0.793±0.002 | 0.722±0.000 | 0.685±0.017 | **0.514±0.009** |
| 720 | 1.152±0.199 | **0.537±0.004** | 0.546±0.003 | 0.563±0.005 | 1.291±0.037 | 1.162±0.020 | 0.733±0.058 | **0.551±0.008** | 0.746±0.000 | 0.746±0.000 | 0.698±0.028 | **0.569±0.002** |
| ETTm2 96 | 0.236±0.015 | **0.209±0.004** | 0.207±0.000 | 0.203±0.000 | 0.314±0.002 | 0.316±0.004 | 0.250±0.012 | **0.223±0.004** | 0.232±0.000 | **0.229±0.000** | 0.261±0.008 | 0.232±0.004 |
| 192 | 0.296±0.002 | **0.268±0.004** | 0.273±0.002 | 0.272±0.000 | 0.397±0.005 | 0.392±0.014 | 0.314±0.007 | **0.290±0.001** | 0.297±0.000 | **0.284±0.000** | 0.297±0.002 | 0.287±0.009 |
| 336 | 0.349±0.007 | **0.332±0.001** | 0.337±0.002 | 0.338±0.001 | 0.420±0.003 | 0.419±0.001 | 0.376±0.005 | **0.353±0.003** | 0.344±0.000 | **0.338±0.000** | 0.350±0.002 | 0.356±0.001 |
| 720 | 0.468±0.011 | **0.428±0.003** | 0.435±0.001 | 0.439±0.002 | 0.527±0.005 | 0.527±0.003 | 0.476±0.005 | **0.456±0.005** | 0.433±0.000 | 0.433±0.000 | 0.438±0.003 | 0.446±0.002 |
| Exchange 96 | **0.084±0.003** | 0.084±0.001 | 0.089±0.001 | 0.089±0.000 | 0.096±0.001 | 0.103±0.001 | 0.151±0.008 | **0.094±0.003** | 0.114±0.000 | 0.139±0.000 | 0.163±0.005 | 0.121±0.024 |
| 192 | 0.196±0.001 | 0.191±0.004 | **0.179±0.000** | 0.179±0.003 | **0.185±0.001** | 0.199±0.003 | 0.230±0.006 | 0.187±0.000 | 0.230±0.001 | 0.235±0.000 | 0.223±0.003 | **0.190±0.003** |
| 336 | 0.352±0.010 | 0.353±0.012 | **0.321±0.002** | 0.322±0.002 | **0.308±0.001** | 0.316±0.007 | 0.382±0.005 | 0.333±0.001 | 0.374±0.000 | 0.383±0.000 | 0.376±0.000 | **0.337±0.002** |
| 720 | 0.897±0.014 | 0.877±0.003 | **0.844±0.003** | 0.847±0.005 | **0.810±0.004** | 0.825±0.010 | 0.929±0.009 | 0.869±0.004 | 0.931±0.001 | 0.931±0.000 | 0.913±0.003 | **0.882±0.005** |
| Electricity 96 | 0.484±0.065 | **0.269±0.003** | 0.310±0.008 | 0.289±0.002 | 0.438±0.021 | **0.283±0.001** | 0.336±0.004 | 0.313±0.009 | 0.687±0.001 | 0.846±0.000 | 0.384±0.015 | **0.279±0.016** |
| 192 | 0.525±0.019 | **0.267±0.004** | 0.300±0.004 | 0.285±0.001 | 0.492±0.026 | **0.288±0.002** | 0.324±0.003 | 0.306±0.004 | 0.730±0.001 | 0.849±0.000 | **0.347±0.018** | 0.349±0.005 |
| 336 | 0.444±0.040 | **0.280±0.002** | 0.322±0.002 | 0.304±0.001 | 0.476±0.012 | **0.301±0.002** | 0.343±0.002 | 0.319±0.003 | 0.764±0.001 | 0.861±0.000 | 0.369±0.014 | **0.327±0.004** |
| 720 | 0.543±0.074 | **0.316±0.004** | 0.353±0.004 | 0.337±0.003 | 0.660±0.022 | **0.348±0.004** | 0.374±0.003 | 0.342±0.003 | 0.892±0.000 | 0.892±0.000 | 0.422±0.021 | **0.330±0.002** |
| Traffic 96 | 1.098±0.072 | **0.880±0.004** | 1.087±0.028 | 0.961±0.005 | 0.963±0.004 | **0.950±0.000** | 1.378±0.031 | 1.097±0.017 | 1.251±0.001 | 1.410±0.000 | 1.149±0.000 | **0.869±0.031** |
| 192 | 1.142±0.041 | **0.822±0.012** | 0.998±0.011 | 0.889±0.005 | 0.929±0.006 | **0.880±0.004** | 1.283±0.018 | 1.014±0.008 | 1.293±0.001 | 1.413±0.000 | 1.007±0.022 | **0.991±0.011** |
| 336 | 1.046±0.070 | **0.828±0.011** | 1.016±0.015 | 0.898±0.003 | 0.957±0.015 | **0.895±0.006** | 1.232±0.011 | 1.009±0.008 | 1.328±0.001 | 1.429±0.000 | 0.994±0.013 | **0.948±0.018** |
| 720 | 1.135±0.089 | **0.850±0.005** | 1.040±0.013 | 0.927±0.009 | 0.996±0.010 | **0.926±0.000** | 1.241±0.012 | 0.999±0.009 | 1.451±0.000 | 1.451±0.000 | 0.992±0.017 | **0.916±0.010** |
| Weather 96 | 0.258±0.010 | **0.242±0.004** | 0.252±0.011 | 0.257±0.009 | 0.263±0.002 | 0.259±0.001 | **0.254±0.002** | 0.291±0.005 | **0.197±0.000** | 0.216±0.000 | 0.260±0.004 | 0.292±0.012 |
| 192 | 0.300±0.007 | **0.280±0.011** | 0.292±0.017 | 0.304±0.009 | 0.314±0.004 | 0.311±0.010 | **0.303±0.001** | 0.357±0.009 | **0.255±0.000** | 0.264±0.000 | 0.283±0.005 | 0.324±0.008 |
| 336 | 0.369±0.008 | **0.328±0.005** | 0.342±0.010 | 0.353±0.009 | **0.346±0.001** | 0.346±0.001 | 0.358±0.001 | 0.399±0.008 | **0.303±0.000** | 0.312±0.000 | 0.329±0.003 | 0.361±0.004 |
| 720 | 0.439±0.017 | **0.392±0.007** | 0.403±0.003 | 0.418±0.008 | 0.432±0.002 | 0.432±0.001 | 0.432±0.003 | 0.451±0.006 | **0.381±0.000** | 0.381±0.000 | 0.392±0.003 | 0.413±0.006 |

| Model Data Type dataset horizon | UniTime R | R+S(M) | S(M) | S | TimeMixer R | R+S(M) | S(M) | S | GPT4TS R | R+S(M) | S(M) | S |
|---|---|---|---|---|---|---|---|---|---|---|---|---|
| ETTh1 96 | 0.599±0.011 | 0.464±0.017 | 0.472±0.008 | **0.439±0.009** | 0.681±0.048 | 0.504±0.014 | 0.527±0.001 | **0.473±0.006** | 0.548±0.015 | 0.536±0.019 | 0.580±0.010 | **0.406±0.003** |
| 192 | 0.733±0.081 | 0.545±0.021 | 0.534±0.008 | **0.499±0.007** | 0.709±0.079 | 0.579±0.023 | 0.585±0.010 | **0.524±0.005** | 0.649±0.019 | 0.609±0.006 | 0.637±0.004 | **0.456±0.004** |
| 336 | 0.827±0.111 | 0.609±0.001 | 0.570±0.006 | **0.541±0.002** | 0.771±0.118 | 0.589±0.020 | 0.632±0.003 | **0.571±0.004** | 0.628±0.006 | 0.610±0.017 | 0.670±0.006 | **0.501±0.008** |
| 720 | 1.641±0.410 | 0.676±0.148 | 0.572±0.015 | **0.540±0.016** | 0.890±0.060 | 0.620±0.018 | 0.618±0.020 | **0.552±0.013** | 0.635±0.011 | 0.614±0.014 | 0.652±0.004 | **0.484±0.011** |
| ETTh2 96 | 0.493±0.013 | 0.435±0.009 | **0.314±0.001** | 0.330±0.006 | 0.562±0.018 | 0.501±0.010 | **0.324±0.003** | 0.326±0.003 | 0.449±0.004 | 0.409±0.026 | 0.344±0.004 | **0.307±0.001** |
| 192 | 0.600±0.015 | 0.528±0.004 | **0.410±0.002** | 0.424±0.008 | 0.648±0.022 | 0.608±0.033 | 0.425±0.002 | **0.422±0.005** | 0.544±0.002 | 0.518±0.013 | 0.443±0.003 | **0.396±0.001** |
| 336 | 0.563±0.009 | 0.555±0.018 | **0.451±0.001** | 0.471±0.004 | 0.614±0.008 | 0.572±0.008 | 0.482±0.002 | **0.480±0.013** | 0.531±0.009 | 0.500±0.004 | 0.489±0.002 | **0.445±0.002** |
| 720 | 0.594±0.006 | 0.588±0.027 | **0.454±0.006** | 0.464±0.003 | 0.670±0.011 | 0.600±0.004 | 0.481±0.005 | **0.469±0.001** | 0.575±0.005 | 0.524±0.007 | 0.485±0.002 | **0.440±0.001** |
| ETTm1 96 | 1.074±0.030 | 1.022±0.046 | 0.540±0.106 | **0.389±0.005** | 1.361±0.055 | 1.017±0.082 | 0.595±0.043 | **0.422±0.003** | 1.257±0.029 | 0.824±0.059 | 0.708±0.061 | **0.354±0.001** |
| 192 | 1.174±0.069 | 1.159±0.040 | 0.610±0.106 | **0.438±0.003** | 1.391±0.062 | 1.141±0.085 | 0.663±0.038 | **0.483±0.006** | 1.237±0.026 | 0.819±0.040 | 0.772±0.082 | **0.400±0.001** |
| 336 | 1.136±0.076 | 1.118±0.083 | 0.594±0.052 | **0.472±0.011** | 1.235±0.030 | 1.038±0.058 | 0.666±0.022 | **0.521±0.009** | 1.091±0.038 | 0.719±0.040 | 0.719±0.044 | **0.439±0.007** |
| 720 | 1.149±0.017 | 1.158±0.125 | 0.659±0.059 | **0.539±0.016** | 1.353±0.068 | 1.079±0.025 | 0.712±0.028 | **0.595±0.013** | 1.169±0.012 | 0.799±0.035 | 0.760±0.038 | **0.509±0.008** |
| ETTm2 96 | 0.313±0.005 | 0.283±0.011 | 0.217±0.008 | **0.204±0.002** | 0.336±0.002 | 0.315±0.015 | 0.242±0.003 | **0.211±0.001** | 0.304±0.007 | 0.260±0.013 | 0.248±0.004 | **0.200±0.002** |
| 192 | 0.384±0.006 | 0.345±0.002 | 0.282±0.006 | **0.269±0.000** | 0.395±0.008 | 0.386±0.010 | 0.303±0.006 | **0.282±0.002** | 0.372±0.004 | 0.334±0.011 | 0.305±0.003 | **0.262±0.001** |
| 336 | 0.410±0.005 | 0.399±0.008 | 0.333±0.003 | **0.330±0.002** | 0.417±0.005 | 0.401±0.010 | 0.361±0.003 | **0.351±0.002** | 0.394±0.003 | 0.362±0.006 | 0.360±0.003 | **0.323±0.003** |
| 720 | 0.520±0.003 | 0.500±0.006 | 0.436±0.004 | **0.430±0.002** | 0.521±0.008 | 0.501±0.010 | 0.457±0.005 | **0.453±0.004** | 0.503±0.001 | 0.466±0.006 | 0.456±0.002 | **0.423±0.003** |
| Exchange 96 | 0.099±0.001 | 0.094±0.001 | 0.144±0.010 | **0.092±0.000** | 0.096±0.000 | 0.095±0.002 | 0.152±0.005 | **0.089±0.002** | 0.104±0.004 | 0.109±0.003 | 0.159±0.005 | **0.094±0.001** |
| 192 | 0.204±0.008 | 0.189±0.001 | 0.229±0.012 | **0.180±0.001** | 0.205±0.008 | 0.208±0.001 | 0.226±0.004 | **0.179±0.004** | 0.230±0.003 | 0.231±0.001 | 0.233±0.005 | **0.186±0.001** |
| 336 | 0.334±0.002 | 0.316±0.004 | 0.374±0.008 | 0.324±0.004 | 0.335±0.007 | 0.332±0.004 | 0.382±0.005 | 0.324±0.003 | 0.354±0.004 | 0.363±0.002 | 0.379±0.005 | **0.329±0.001** |
| 720 | **0.829±0.003** | 0.837±0.013 | 0.918±0.012 | 0.842±0.015 | **0.833±0.004** | 0.847±0.005 | 0.926±0.011 | 0.853±0.007 | 0.882±0.006 | 0.899±0.004 | 0.922±0.008 | 0.858±0.002 |
| Electricity 96 | 0.412±0.011 | **0.273±0.001** | 0.298±0.004 | 0.291±0.007 | 0.359±0.015 | **0.284±0.001** | 0.313±0.006 | 0.294±0.002 | 0.411±0.013 | 0.306±0.004 | 0.356±0.006 | **0.278±0.001** |
| 192 | 0.460±0.007 | **0.277±0.005** | 0.292±0.006 | 0.286±0.006 | 0.383±0.021 | **0.280±0.004** | 0.305±0.003 | 0.289±0.002 | 0.451±0.002 | 0.306±0.003 | 0.352±0.002 | **0.271±0.000** |
| 336 | 0.577±0.063 | **0.295±0.003** | 0.309±0.005 | 0.298±0.001 | 0.382±0.016 | **0.288±0.010** | 0.331±0.004 | 0.308±0.005 | 0.432±0.004 | 0.307±0.001 | 0.362±0.003 | **0.288±0.001** |
| 720 | 0.844±0.160 | 0.345±0.004 | 0.343±0.004 | **0.337±0.004** | 0.461±0.036 | **0.329±0.003** | 0.367±0.011 | 0.341±0.005 | 0.476±0.012 | 0.346±0.003 | 0.392±0.005 | **0.323±0.001** |
| Traffic 96 | 0.976±0.008 | 0.921±0.023 | 1.065±0.018 | **0.909±0.013** | **0.903±0.016** | 0.929±0.019 | 1.223±0.011 | 0.986±0.011 | 0.994±0.003 | 0.975±0.027 | 1.315±0.023 | **0.863±0.006** |
| 192 | 0.957±0.022 | 0.872±0.005 | 0.997±0.029 | **0.844±0.010** | 0.882±0.021 | 0.844±0.024 | 1.138±0.020 | 0.912±0.005 | 1.010±0.040 | 0.915±0.008 | 1.253±0.017 | **0.792±0.005** |
| 336 | 0.979±0.010 | 0.890±0.003 | 0.959±0.013 | **0.849±0.008** | 0.866±0.019 | 0.854±0.031 | 1.104±0.006 | 0.915±0.012 | 0.954±0.016 | 0.881±0.012 | 1.190±0.017 | **0.806±0.006** |
| 720 | 1.029±0.013 | 0.935±0.009 | 0.990±0.019 | **0.886±0.021** | 0.938±0.017 | 0.910±0.018 | 1.148±0.044 | 0.936±0.017 | 0.985±0.010 | 0.911±0.014 | 1.203±0.017 | **0.834±0.004** |
| Weather 96 | 0.219±0.005 | **0.209±0.002** | 0.223±0.004 | 0.263±0.009 | 0.217±0.002 | **0.213±0.002** | 0.244±0.001 | 0.295±0.023 | 0.221±0.002 | **0.220±0.002** | 0.242±0.006 | 0.262±0.006 |
| 192 | 0.274±0.003 | **0.263±0.006** | 0.270±0.003 | 0.302±0.009 | **0.263±0.003** | 0.268±0.005 | 0.288±0.003 | 0.344±0.028 | 0.289±0.005 | 0.291±0.001 | **0.290±0.007** | 0.308±0.007 |
| 336 | 0.323±0.005 | 0.317±0.006 | **0.316±0.003** | 0.349±0.004 | 0.313±0.029 | **0.312±0.004** | 0.341±0.003 | 0.391±0.020 | 0.328±0.001 | **0.325±0.005** | 0.339±0.004 | 0.352±0.009 |
| 720 | 0.399±0.005 | 0.396±0.002 | **0.392±0.005** | 0.413±0.004 | 0.388±0.001 | **0.385±0.004** | 0.414±0.005 | 0.445±0.021 | 0.406±0.002 | **0.400±0.002** | 0.410±0.004 | 0.412±0.012 |

| Model Data Type dataset horizon | PatchTST R | R+S(M) | S(M) | S | DLinear R | R+S(M) | S(M) | S |
|---|---|---|---|---|---|---|---|---|
| ETTh1 96 | 0.728±0.013 | 0.567±0.018 | 0.480±0.006 | **0.433±0.001** | 0.592±0.007 | 0.502±0.007 | 0.661±0.019 | **0.470±0.002** |
| 192 | 1.154±0.030 | 0.803±0.095 | 0.537±0.006 | **0.487±0.005** | 0.630±0.008 | 0.543±0.007 | 0.732±0.015 | **0.522±0.004** |
| 336 | 2.845±0.238 | 1.275±0.361 | 0.577±0.006 | **0.533±0.012** | 0.628±0.006 | 0.570±0.002 | 0.735±0.012 | **0.564±0.003** |
| 720 | 3.103±0.465 | 1.830±0.268 | 0.556±0.013 | **0.517±0.009** | 0.635±0.006 | 0.582±0.007 | 0.689±0.078 | **0.561±0.004** |
| ETTh2 96 | 0.537±0.005 | 0.521±0.018 | 0.320±0.005 | **0.312±0.003** | 0.949±0.033 | 0.479±0.046 | 0.370±0.005 | **0.329±0.002** |
| 192 | 0.653±0.010 | 0.631±0.025 | 0.419±0.001 | **0.405±0.004** | 1.066±0.065 | 0.583±0.031 | 0.463±0.005 | **0.422±0.004** |
| 336 | 0.610±0.007 | 0.576±0.050 | 0.466±0.002 | **0.459±0.006** | 0.933±0.050 | 0.602±0.016 | 0.514±0.007 | **0.498±0.004** |
| 720 | 0.667±0.015 | 0.615±0.006 | 0.461±0.002 | **0.452±0.004** | 1.092±0.045 | 0.774±0.028 | 0.625±0.006 | **0.578±0.032** |
| ETTm1 96 | 1.142±0.008 | 1.092±0.110 | 0.521±0.073 | **0.373±0.005** | 0.998±0.005 | 0.788±0.018 | 0.616±0.007 | **0.433±0.003** |
| 192 | 1.209±0.032 | 1.143±0.091 | 0.565±0.059 | **0.424±0.005** | 1.017±0.018 | 0.828±0.004 | 0.709±0.028 | **0.497±0.014** |
| 336 | 1.270±0.065 | 1.098±0.208 | 0.573±0.044 | **0.457±0.005** | 0.860±0.016 | 0.737±0.009 | 0.688±0.012 | **0.540±0.012** |
| 720 | 1.351±0.161 | 1.123±0.067 | 0.628±0.035 | **0.527±0.009** | 0.898±0.007 | 0.774±0.009 | 0.715±0.021 | **0.601±0.010** |
| ETTm2 96 | 0.357±0.007 | 0.358±0.008 | 0.220±0.003 | **0.193±0.001** | 0.908±0.034 | 0.411±0.049 | 0.254±0.003 | **0.216±0.001** |
| 192 | 0.434±0.007 | 0.424±0.011 | 0.281±0.003 | **0.260±0.001** | 1.007±0.068 | 0.495±0.031 | 0.322±0.004 | **0.292±0.003** |
| 336 | 0.440±0.006 | 0.433±0.012 | 0.341±0.001 | **0.323±0.002** | 0.855±0.053 | 0.501±0.027 | 0.385±0.017 | **0.379±0.004** |
| 720 | 0.563±0.008 | 0.557±0.007 | 0.439±0.003 | **0.426±0.005** | 1.087±0.046 | 0.751±0.028 | 0.580±0.009 | **0.553±0.031** |
| Exchange 96 | 0.094±0.000 | 0.098±0.002 | 0.142±0.005 | **0.092±0.002** | 0.742±0.037 | 0.251±0.049 | 0.127±0.003 | **0.078±0.001** |
| 192 | 0.189±0.005 | 0.189±0.007 | 0.221±0.005 | **0.183±0.001** | 0.811±0.069 | 0.323±0.026 | 0.194±0.007 | **0.152±0.002** |
| 336 | 0.319±0.002 | 0.312±0.003 | 0.371±0.005 | 0.323±0.003 | 0.704±0.051 | 0.364±0.010 | 0.290±0.003 | **0.247±0.001** |
| 720 | 0.848±0.004 | **0.838±0.008** | 0.918±0.014 | 0.847±0.028 | 0.886±0.038 | 0.563±0.013 | 0.495±0.006 | **0.468±0.013** |
| Electricity 96 | 0.421±0.032 | **0.280±0.002** | 0.311±0.003 | 0.287±0.003 | 0.396±0.004 | 0.321±0.011 | 0.381±0.029 | **0.301±0.001** |
| 192 | 0.577±0.055 | **0.284±0.001** | 0.305±0.002 | 0.285±0.004 | 0.419±0.006 | 0.341±0.008 | 0.391±0.023 | **0.294±0.002** |
| 336 | 0.678±0.036 | **0.297±0.007** | 0.322±0.003 | 0.305±0.001 | 0.407±0.002 | 0.332±0.003 | 0.385±0.017 | **0.309±0.002** |
| 720 | 0.919±0.038 | 0.349±0.012 | 0.355±0.001 | **0.337±0.003** | 0.431±0.003 | 0.364±0.009 | 0.395±0.025 | **0.332±0.004** |
| Traffic 96 | 0.930±0.008 | 0.917±0.008 | 1.116±0.044 | **0.901±0.007** | 1.005±0.006 | 1.059±0.021 | 1.640±0.056 | **0.981±0.005** |
| 192 | 0.912±0.008 | 0.851±0.014 | 1.066±0.037 | **0.845±0.011** | 0.988±0.011 | 1.014±0.010 | 1.655±0.047 | **0.907±0.008** |
| 336 | 0.939±0.016 | 0.856±0.037 | 1.040±0.024 | **0.857±0.003** | 0.957±0.006 | 0.969±0.002 | 1.498±0.036 | **0.913±0.005** |
| 720 | 1.034±0.047 | 0.923±0.017 | 1.067±0.016 | **0.878±0.012** | 0.977±0.002 | 0.975±0.009 | 1.325±0.250 | **0.917±0.006** |
| Weather 96 | **0.227±0.001** | 0.235±0.007 | 0.246±0.003 | 0.222±0.006 | 0.282±0.005 | **0.222±0.006** | 0.253±0.005 | 0.394±0.023 |
| 192 | **0.285±0.003** | 0.300±0.003 | 0.272±0.002 | 0.292±0.005 | 0.332±0.010 | **0.270±0.004** | 0.299±0.007 | 0.433±0.022 |
| 336 | **0.330±0.007** | 0.343±0.008 | 0.324±0.002 | 0.340±0.006 | 0.349±0.008 | **0.308±0.000** | 0.336±0.003 | 0.464±0.032 |
| 720 | 0.411±0.004 | 0.420±0.004 | **0.396±0.004** | 0.405±0.008 | 0.408±0.005 | **0.374±0.004** | 0.388±0.016 | 0.483±0.021 |

