# OpenReview forum: "Synth-FAR: A Synthetic Frequency-Autoregressive Driven Framework for Time Series Forecasting"
_TMLR — Accepted by TMLR_

### Review · Reviewer_JA2B · 2026-03-23

**Summary Of Contributions:**

This paper focuses on time series forecasting (TSF) for low-data regimes. It provides both an analytical perspective (frequency analysis) and a practical tool, termed Synth-FAR, which is a synthetic data generation method. Specifically speaking, this work introduces theoretical concepts—‌frequency adaptation‌ and ‌frequency generalization‌—to analyze the challenges models face when training data contains irrelevant frequencies or lacks target-domain frequencies. It demonstrates, via Fourier analysis, that poor frequency generalization and adaptation are key factors limiting the performance of both foundational and non-foundational time series forecasting models, especially in data-scarce or zero-shot scenarios. The paper proposes a new synthetic data generation framework designed specifically to address the identified frequency-related shortcomings. It generates data by creating a pool of sine waves (based on a fundamental frequency and its harmonics) mixed with autoregressive (AR) processes, providing two key configurations: one that utilizes a target dataset's sampling rate and one that does not (a mixed-frequency approach). Through experiments, the paper shows that Synth-FAR outperforms other state-of-the-art synthetic data generation methods like TimesFM, Kernel-Synth, and ForecastPFN in zero-shot forecasting benchmarks.

**Audience:**

Yes

**Audience Explanation:**

This paper could interest part of TMLR's audience that research on time series forcasting.

**Claims And Evidence:**

No

**Claims Explanation:**

The performance of Synth-FAR is heavily dependent on identifying an accurate fundamental frequency for the target domain. The paper acknowledges that in datasets with a "wider range of dominant frequencies" or highly random signals, the single fundamental frequency heuristic may fail (e.g., poor performance on the Exchange dataset is noted). The method struggles when the target signal lacks a clear, dominant periodic component, as its core assumption—that a single fundamental frequency dominates—breaks down.

The comparison in Section 5.3 focuses on existing pre-trained foundation models (FM) that may have their own architectural and training limitations. While Synth-FAR-based models outperform them, a more direct comparison could involve using Synth-FAR as a data augmentation technique during the pre-training of a new foundation model, to see if it helps overcome the "frequency overfit" problem the paper identifies in FMs. The current experiment shows that ‌using Synth-FAR with a non-FM can be more effective than a pre-trained FM alone‌, which is strong, but a further question is whether it could improve FMs.

The framework heavily relies on sinusoidal waves, which may lead to synthetic data that is too structured for datasets with complex, non-stationary patterns or strong non-linearities not easily represented as a mixture of harmonics. The added AR(p) component helps with this, but its role as a "conflicting frequency mitigator" (as described in the ablation) is not deeply analyzed theoretically. Its integration, while empirically beneficial, is somewhat heuristic.

**Requested Changes:**

See comments above.

---

> ### Author Response · Authors · 2026-04-28
>
> We thank the reviewer for the thoughtful and constructive feedback. Below, we address each concern in detail.
>
> ### 1. Dependence on a Single Fundamental Frequency ###
>
>
> > Synth-FAR relies on identifying a dominant fundamental frequency, which may fail for datasets with complex or diffuse frequency spectra (e.g., Exchange).
>
> Response:
> We agree that relying on a single dominant frequency is an assumption and may not hold for all datasets. However, the paper explicitly acknowledges and mitigates this limitation in several ways:
>
> * Mixed-frequency setting (no prior knowledge):
> Synth-FAR includes a frequency-agnostic configuration where multiple common real-world frequencies are used (e.g., hourly, daily, weekly), removing the dependency on accurate frequency estimation.
> This setting is specifically designed for cases where the dominant frequency is unclear or absent. see Table 1 with Known Sampling Rate vs Unknown Sampling Rate (mix).
>
> * Harmonics and multi-frequency coverage:
> Even in the “single-frequency” setting, Synth-FAR does not generate a single sinusoid but rather a distribution over harmonics, This allows the generated data to span a local frequency neighborhood, partially addressing spectral spread.
>
>
> * AR(p) compensation mechanism:
> When frequency structure is weak, the model shifts reliance toward temporal locality: “AR(p) supports cases with complex, low-frequency, or mixed frequency patterns”. This is empiricaly validated in the appendix B.1, where the where increasing the number AR(p) proccess contibutes to the performance of **weather** and **ETTh2**, which are datasets with mixed seasonalities, and a low to modearete level of seasonlity socre (0.415) as shown in Figure 7.
>
>
> * Empirical validation and acknowledgement of the limitation:
> The paper explicitly reports degraded performance on datasets such as Exchange and attributes this to complex / weak periodic structure. This is also discussed in the Appedinx (C.4 Synth-FAR limitations).
>
> ### 2. Lack of Integration with Foundation Model Pre-training ###
>
> > The paper does not test Synth-FAR as a data augmentation strategy during foundation model (FM) pre-training.
>
> Response:
> We appreciate this point. Super-Linear does not explicitly follow a full foundation-model pretraining regime, primarily due to practical constraints: large-scale pretraining demands substantial compute, many training pipelines and datasets are not publicly accessible, and controlled comparisons across diverse foundation models are difficult. Instead, we evaluate a broad set of representative architectures—including CNNs, linear models, patch-based methods, and Transformer-based models (e.g., GPT-style and Timer-like)—which capture the core components of most foundation models (patch embeddings, attention, and linear projections). We also train several widely used time-series models (e.g., TTM, Timer, MoMENT) from scratch to strengthen this connection.
>
> Our experimental design intentionally isolates the effect of synthetic data by training non-foundation models purely on synthetic data and comparing them directly against pre-trained foundation models. This allows us to demonstrate a key result: task-specific synthetic data can enable relatively simple models to outperform large pre-trained models. Furthermore, our results show that synthetic data improves performance when combined with real data and highlight frequency misalignment as a key limitation of foundation models, suggesting that Synth-FAR can serve as a targeted regularization mechanism within foundation model pipelines.
>
>
> ### 3. Overly Sinusoidal / Structured Data Assumption
>
>
> > Synth-FAR may be too structured (sinusoidal)
>
> Our goal is too rather emphasize the importance of such stuctures for generalization. This work  isolates the sinunsidal waves and shows the potential in this componenets in terms of generalization. According to our analysis, instead of generation rich and sphosticated data generation regimes such as Kernel-Synth, it can be shown that with a selective set of simple sine waves much can be already achieved.

---

### Review · Reviewer_C5ua · 2026-04-06

**Summary Of Contributions:**

The paper proposes a concise and fast approach to generate synthetic time series data which can augment real data for training time series forecasting models. Approach is dubbed Synth-FAR, and assumes that target forecasting problems have small number of underlying main frequencies. Extensive empirical evaluations were conducted, and the observations were discussed and interpreted, mainly from viewpoint of underlying frequencies. Main conclusion is that addition of Synth FAR synthetic data helps improve zero shot prediction performance across forecasting methods and datasets, compared to other synthetic data generation approaches like TimesFM, ForecastPFN, and Kernel-Synth.

**Audience:**

Yes

**Audience Explanation:**

Improving zero shot time forecasting through synthetic data augmentation is fairly relevant topic, and even beyond time series domain general data augmentation approaches for foundational models are of increasing importance.

**Claims And Evidence:**

No

**Claims Explanation:**

Strengths:
- Problem is well formulated and motivated and is fairly relevant
- Extensive empirical evaluation with many aspects investigated
- Clear flow and very detailed description of conducted work

Weaknesses:
- Assumption on "known sampling frequency" is strong - and the proposed framework can utilize such "unfair" advantage of having that information fed to it directly
- Frequency adaptation and generalization concepts are not quite novel - and are more like task-specific re-labelings of heterogeneous training distributions and out of domain generalization concepts - need more connecting to existing literature
- Fairness of experimentation setup: datasets seems biased toward ones with "natural periodicity", context windows of 96 are not most suitable for pretrained models, Synth-FAR setups used 50 percent of training data - maybe using more data for alternatives led to overfit

**Requested Changes:**

- For more fair comparison - can other approaches also use fundamental frequency for generating synthetic data?
- Can you discuss potential ways to empirically estimate the fundamental frequencies from the data?
- Can the model training be controlled for overfitting for more apple to apple comparison?
- Can you comment on why choosing AR coefficients to be uniform and sum to exactly 1?
- Can you evaluate pretrained models for their intended (larger) context windows?
- While synthetic data generation is faster, can you comment on its overall effect on the end to end process, since main time expensive stage is actually model training?

---

> ### Author Response · Authors · 2026-04-28
>
> Thank you for your detailed and constructive review, we appreciate the time and effort you put into carefully evaluating our work. Your comments raise important points regarding assumptions, experimental fairness, and the positioning of our contributions within existing literature. Below, we address your concerns and clarify our design choices, empirical setup, and the scope of our claim.
>
> > Assumption on "known sampling frequency" is strong - and the proposed framework can utilize such "unfair" advantage of having that information fed to it directly
>
> Thanks for this remark. We understand that is poses an advantage however, in his work we propose 2 settings, 1) "known sampling rate" with access to the fundemtal frequency, and 2) unknown sampling rate which consists of a several mixed frequencies. Both are utilized and compared in the main experiments: Table 1 and Table 2.
>
> > Frequency adaptation and generalization concepts are not quite novel - and are more like task-specific re-labelings of heterogeneous training distributions and out of domain generalization concepts - need more connecting to existing literature
>
> Thanks for this remark. it is correct, the term Frequency Adaptation and Frequency Generalization were adapted from the terms Domain generalization and domain adaptation, as described in [1]
> We now emphasized in the in revision (section 4.1) where we make direct assosication to domain adapatation and generalization. Additionaly, we added more references to works that cover these topics
> most importantly before their definition we added:
>
>  "... and the following two new
> frequency-based concepts **that were adapted from the transfer learning literature**"
>
> [1] Generalizing to Unseen Domains: A Survey on Domain Generalization
>
>
>
> ---
> Requested Changes:
>
>
> > For more fair comparison - can other approaches also use fundamental frequency for generating synthetic data?
>
> We would like to refer to Table 1, where we compare two settings, with known and unknown sampling rates. For the known sampling rate, we also utilize the same fundamental frequency for the compared methods: Kernel-Synth, TimesFM, and PFN. whereas the right part of the table, all approaches utilize mixed data, including ours (Synth-FAR)
>
> > Can you discuss potential ways to empirically estimate the fundamental frequencies from the data?
>
> This is discussed in C.1 Fundamental Frequency Estimation. We also improved the explanations in the revised version.
>
> > Can the model training be controlled for overfitting for more apple to apple comparison?
>
> All methods were already evaluated under a unified training protocol (same architectures, lookback, horizons, splits, seeds, optimizer settings where possible). Regarding the fundamental frequency, please see our answer above. Additionally, in Table 1, the synthetic data across all methods include approximately the same amount of train data timesteps.
>
> > Can you comment on why choosing AR coefficients to be uniform and sum to exactly 1?
>
> This choice was intentional as a simple, stable prior rather than an optimized AR estimator. Setting $\gamma_j = 1/p$ and $\sum_j \gamma_j=1$ creates a smooth averaging process spanning recent history: for small $p$ it resembles **naive** behavior, while larger $p$ approximates moving-average smoothing. This keeps the generator lightweight,  and easy to control, while complementing the sinusoidal components. We agree richer learned/non-uniform coefficients are an interesting extension.
>
> the motivation is that some time-series forecast signals are better forecasted by the last seen value (naive $p=1$) or the average signal's lookback  (mean $p=lookback$). This may be favourable if the seasonal components are not strong, and in our case, support generalization. This is also discussed in 4.3 Synth-FAR: Synthetic time series based on fundamental frequencies and AR(p).
>
> Furthermore, we provide an ablation for different $p$ parameters in B.3 Ablation: AR p size.

---

> > ### Author Response · Authors · 2026-04-28
> >
> > > Can you evaluate pretrained models for their intended (larger) context windows?
> >
> > While pre-trained models commonly operate with look-back windows of 512 or more, our objective in this section is essentially the opposite. We deliberately explore settings where foundation models are known to be less reliable, specifically, frequencies outside their pre-training frequency range and short look-back windows. These configurations expose scenarios in which TSFMs tend to struggle, allowing us to highlight the strengths and potential of Synth-FAR in addressing these gaps.
> >
> > Furthermore, to reinforce the focus on short lookbacks, the datasets used in Section 5.3 are inherently short. Several datasets, such as those from the Time-MMD collection (Agriculture, Climate, Economy, Security), contain fewer than 500 time steps and therefore cannot support a look-back window of 512. For this reason, these datasets were intentionally selected to evaluate Synth-FAR under realistic short-context conditions.
> >
> > it is important for us to emphasize: We do not aim to compete directly with large foundation models, but rather to identify their limitations, particularly in pre-training, and propose simple, principled ways to address them. Accordingly, some of our evaluations deliberately move away from standard benchmarks (e.g., 512-context lookback) to better expose potential pre-training failures.
> >
> > > While synthetic data generation is faster, can you comment on its overall effect on the end to end process, since main time expensive stage is actually model training?
> >
> > Correct that model training is usually the dominant computational cost. Our point is that synthetic generation should be evaluated as part of the full pipeline, not only as an isolated preprocessing step. Since Synth-FAR generation is extremely lightweight, it adds negligible overhead relative to training time. In contrast, heavier synthetic pipelines can introduce additional preprocessing cost, especially when generating large synthetic corpora.
> >
> > Even though the overall runtime remains dominated by training, the ease of SynthFAR enables users to generate or refresh synthetic datasets rapidly, explore multiple configurations efficiently, and adapt the data to new target domains with only minimal additional computational cost.

---

### Review · Reviewer_Zt78 · 2026-04-15

**Summary Of Contributions:**

The paper analyzes the importance of frequencies in time series models, especially in the context of generalization. It introduces the concepts frequency adaptation and frequency generalization as an analysis lens to facilitate the identification of potential challenges in Zero Shot (ZS) forecasting. The paper proposes a simple, easy-to-code, multi purpose and efficient time series synthetic generator (Synth-FAR) that is effective for different tasks, and through several tests, it demonstrates that Synth-FAR achieves better (ZS) results than other synthetic datasets and complements foundation and non-foundation models in areas where they fall short.

The paper is well written and presents several experiments to demonstrate the value and merit of the proposed synthetic generator, which not only leads to moderate improvements in the predictive performance of the trained models, but is also computationally lightweight compared to state-of-the-art time series generation approaches.

There are still some issues that need further clarification or improvement, which I explain in the next fields.

Overall, I find this work quite promising and useful for the community.

**Additional Comments:**

While I worked several years ago on the problem of time series forecasting/prediction, during the last years I haven't followed the developments in the field. Therefore, I am not fully confident on my feedback and may have missed recent works that have proposed similar ideas and results.

**Audience:**

Yes

**Audience Explanation:**

Yes, the paper could be appealing to a sizeable part of TMLR's audience who are interested in time-series forecasting. The proposed approach is straightforward and easy to embed in follow-up work and therefore could also have a notable impact on the domain.

**Broader Impact Concerns:**

Not applicable.

**Claims And Evidence:**

Yes

**Claims Explanation:**

The paper’s first main empirical observation is that models struggle when the training set contains more irrelevant frequencies, and that having access to the target frequency during training is important. The experiments in Figure 1 give evidence in favour of this claim since test MSE rises as additional frequencies are added, and performance is much better when the target frequency is included in training than when it is absent. The transfer experiment also supports the claim that frequency-space alignment can matter for transfer. So the paper does provide evidence that “frequency adaptation” and “frequency generalization” are useful concepts for analysing model behavior.

The practical claim that Synth-FAR often beats other synthetic-data generators is also supported reasonably well in the paper’s own benchmark. In Table 1, Synth-FAR has the best average MSE in both the known-sampling-rate setting and the unknown-sampling-rate mixed setting. The generation-time comparison is also strong in favour of Synth-FAR.

There are, however, some claims that are not, in my view, fully supported by the evidence. For instance, the paper argues that poor results of real-data pretraining are due to poor frequency generalization or poor frequency adaptation, and that Synth-FAR fixes this by matching target-domain frequency structure. This is a plausible hypothesis, but I am not sure that the experiments have fully isolated frequency as a factor in order to test it. The comparisons bundle together several things at once: synthetic versus real data, narrower versus broader frequency support, AR components, different training distributions, and in some cases use of target sampling-rate information. The results are indeed consistent with the proposed mechanism, but it is not entirely clear that they conclusively establish that the gains come primarily from the Fourier-based reasoning rather than from other factors.

Another issue is the reliance on sampling rate as a proxy for fundamental frequency. The paper notes this heuristic can fail when the true dominant frequency is unnatural, and that the method struggles when the spectral content is spread out. That is important because many real-world series are not clearly bound to a single fundamental period. Since the known-sampling-rate setting is where the largest gains appear, this might compromise the generality of the strongest claims.

In terms of methodology, the paper reports averaging over three random seeds, which is good, but it does not present standard deviations, confidence intervals, or significance tests in the main quantitative results. Given that many improvements are moderate, especially in some rows of Tables 1 and 2, it is hard to judge robustness.

Last, I think that in terms of related work, there is likely much more relevant work where frequency analysis is employed to improve time series prediction, and therefore the phrasing of claims about the novelty of the proposed study should be a bit toned down and better contextualized (potentially also some additional related methods, e.g. frequency-domain deep models, spectral forecasting, etc. should be discussed).

**Requested Changes:**

I would recommend the following:
- provide uncertainty estimates and/or significance tests together with the results,
- conduct more tightly controlled ablations isolating frequency alignment from other factors,
- carry out stronger analysis of cases where the sampling-rate-to-frequency heuristic fails.
- bring some of the material from the appendix into the main text. For instance, I found useful the short description of the competing methods and datasets.

Additionally, some minor presentation issues:
- use "training set" instead of "train set"
- check the alignment of labels to the bars of Figure 3. They seem out of place to me.
- conduct a careful proofreading of the paper.

---

> ### Author Response · Authors · 2026-04-28
>
> Thank you for the thorough and thoughtful review—we sincerely appreciate the time and care you invested in evaluating our work. We are glad that you found the paper promising and relevant to the community, and that the main ideas and empirical findings were clear. Your comments regarding the need for stronger evidence, clearer isolation of factors, and improved statistical reporting are particularly valuable. We also appreciate your suggestions on presentation and related work, which will help us strengthen the paper further. Below, we address each of your points in detail and clarify the key aspects of our methodology, experiments, and claims.
>
>
> > There are, however, some claims that are not, in my view, fully supported by the evidence...., but it is not entirely clear that they conclusively establish that the gains come primarily from the Fourier-based reasoning rather than from other factors.
>
> We agree that isolating the exact source of performance gains is important, and we made a deliberate effort to control for confounding factors throughout our study.
>
> **Controls and isolation of factors:**
> - The order $ p $ of the $AR(p) $ processes across all experiments is fixed, ensuring that improvements are not driven by changes in autoregressive capacity.
> - In Section B.4 (**Varying Lookbacks**), we explicitly isolate the effect of lookback length to evaluate SynthFAR under different temporal contexts.
> - In the same section, we also control for model capacity, allowing us to disentangle gains due to architectural scaling from those induced by the data generation procedure.
>
> **Experimental design and robustness in different scenarios:**
> - We evaluate across multiple settings: synthetic vs. real data, narrow vs. broad frequency support, and with/without AR components.
> - The purpose of this breadth is not to conflate factors, but to verify that improvements persist across diverse regimes rather than being tied to a single configuration.
>
> **Simplicity of the proposed method:**
> - Despite the wide evaluation, the method itself is intentionally minimal:
>   - Relies only on sinusoidal components.
>   - Uses simple autoregressive processes.
> - This contrasts with more complex alternatives (e.g., kernel-based synthetic generators), where multiple interacting components (e.g., varying kernels, series transformations) make it harder to attribute performance gains to a specific mechanism.
>
>
> > Another issue is the reliance on sampling rate as a proxy for fundamental frequency. The paper notes this heuristic can fail when the true dominant frequency is unnatural, and that the method struggles when the spectral content is spread out. That is important because many real-world series are not clearly bound to a single fundamental period. Since the known-sampling-rate setting is where the largest gains appear, this might compromise the generality of the strongest claims.
>
> Thanks for noting this, however, we understand this limitation, but in our work we also emphasize the the mixed setting, where different -natural- frequencies are bundled to create a mixed version of Synth-FAR. This is indeed not better than the setting wi the known fundamental frequency, but in:
> -  Table 1. this appaorch is shown to be most performent compared to other (mixed) synthetic data methods (which some also rely on a set of natural frequencies)
> - Table 2. The **S(M)** (Synth Mixed) on average in all models performs better then $R$ which is arbitrary real datasets for training.

---

> > ### Author Response · Authors · 2026-04-28
> >
> > > In terms of methodology, the paper reports averaging over three random seeds, which is good, but it does not present standard deviations, confidence intervals, or significance tests in the main quantitative results. Given that many improvements are moderate, especially in some rows of Tables 1 and 2, it is hard to judge robustness.
> >
> > Thank you for pointing this out. We have updated and expanded Section **E.2 Full Experimental Results** (formerly section F.2) in the Appendix to include standard deviations for all main experiments, namely, Table 1, Table 2, and Figure 3. Additionally, we have included a new table in that section that provides the results from Table 1 with standard deviations. These changes are reflected in the revised version of the manuscript.
> >
> > > there is likely much more relevant work where frequency analysis is employed to improve time series prediction, and therefore the phrasing of claims about the novelty of the proposed study should be a bit toned down and better contextualized (potentially also some additional related methods, e.g. frequency-domain deep models, spectral forecasting, etc. should be discussed).
> >
> > Thank you for your comment. We note that the paper does not claim novelty in Fourier analysis itself, but rather uses it as a framework to study the impact of synthetic data frequency characteristics on zero-shot time series forecasting.
> >
> > Following this remark, we expanded the **Fourier analysis in time series applications** paragraph in the related work to include additional works on frequency-domain deep models, spectral bias, and spectral forecasting, better positioning our contribution within existing literature.
> >
> > ---
> > Requested Changes:
> >
> > > provide uncertainty estimates and/or significance tests together with the results,
> >
> > Added in Section E.2, see above.
> >
> > > conduct more tightly controlled ablations isolating frequency alignment from other factors,
> >
> > Please our answer above.
> >
> > > carry out stronger analysis of cases where the sampling-rate-to-frequency heuristic fails.
> >
> > We revised Appendix Section **C.4: Synth-FAR Limitations** to explicitly address the requested failure cases and clarify scenarios where Synth-FAR is less effective. In particular, we added both quantitative and qualitative analyses through a new table and figure.
> >
> > Low Seasonality Mismatch: We show that in datasets such as Agriculture and Exchange, spectral energy is diffuse and no clear dominant periodicity exists. In these cases, persistence-based baselines (e.g., naive forecasting) can outperform Synth-FAR.
> >
> >
> > Multivariate Mismatch: We further analyze the Weather dataset, where different variates exhibit substantially different spectral profiles. This makes a single dataset-level estimated frequency unsuitable across channels, leading to weaker Synth-FAR performance.
> >
> >
> > New Figure and Table: We added Figure 9, which visualizes the periodograms of Agriculture, Exchange, and Weather, together with the estimated fundamental frequency (red dashed line), highlighting the mismatch types. We also added Table 7, summarizing representative benchmark results for these failure scenarios.
> >
> >
> > Overall, this revised section now provides a clearer discussion of Synth-FAR limitations and demonstrates that the method is most effective when the target data exhibits coherent seasonal structure.
> >
> > > bring some of the material from the appendix into the main text. For instance, I found useful the short description of the competing methods and datasets.
> >
> > In the revision, we moved the \textit{Datasets and Models} section to the main paper to improve clarity and make the manuscript more self-contained.
> >
> > > Additionally, some minor presentation issues...
> >
> > All the the mentioned representation issues were fixed.

---

> > > ### Comment · Reviewer_Zt78 · 2026-04-28
> > > **Improved but some remaining issues**
> > >
> > > The revision substantially improved the related work and corrected the most problematic claim. However, I find the positioning of the contribution is still somewhat overstated. The paper would benefit from a clearer and more precise articulation of how its contributions differ from prior frequency-based methods, especially those addressing spectral representations and generalization. The paper still implies that frequency-based understanding of learning is largely missing from prior work, while the several related works discussed already offer such understanding. Perhaps a statement along the lines of "Prior work uses frequency representations, but does not study generalization under frequency shift." could help address this concern.
> > >
> > > Moreover, existing lines of work are still not sufficiently discussed. These for instance include explicit frequency-domain learning models (not just transformers with frequency components), works on spectral bias/frequency learning theory, and broader signal processing forecasting literature.
> > >
> > > The paper still includes strong claims, e.g. "Fourier analysis is the natural framework for assessing effectiveness of synthetic data". While this might be a good framework, calling it a natural framework implies that it is superior to alternatives even though this is not empirically validated. I would suggest to tone down the phrasing accordingly.

---

> ### Author Response · Authors · 2026-05-09
>
> # Phrasing
>
> Thanks for pointing it out, we altered various phrases across the papers in line with the argument that Fourier analysis is uniquely suited for TSF, but rather to show that spectral alignment provides a useful lens for understanding synthetic-data-driven generalization, including:
>
>
>
> **We advocate that Fourier analysis is the natural framework for assessing the effectiveness of synthetic data in TSF**
>
> to
>
> **We employ Fourier analysis as a principled framework for studying the effectiveness of synthetic data in TSF.**
>
> ---
>
>
>
> **Unfortunately, the factors that govern effective learning from synthetic data using non-foundation models remain unclear, and our work aims to advance the general understanding of this challenge.**
>
> to
>
> **To better understand how TSF models leverage synthetic data, this work investigates the impact of frequency coverage and shift on zero-shot forecasting. Through Fourier analysis, we introduce the notions of frequency adaptation and frequency generalization.**
>
> ---
> In section 4.2
>
> **..., the above analysis reinforces the emergence of Fourier analysis as a key tool for determining the factors that affect effective learning**
>
> to
>
> **..., the above analysis reinforces the emergence of Fourier analysis as a useful perspective for studying factors that affect effective learning.**
>
>
>
> ---
>
> We altered the related work section to ensure better positioning of our work, hence, we added:
>
>
> **In contrast to prior works that primarily incorporate spectral representations into model architectures or study spectral properties of neural optimization, our work focuses on how frequency alignment, frequency coverage, and spectral mismatch influence transfer and zero-shot forecasting performance across datasets and synthetic training distributions. In particular, we study generalization under frequency shift and analyze how synthetic data can improve robustness to unseen or underrepresented frequencies.**
>
>
> # Related Work Restructure
> We restructured the **Fourier analysis in time series applications** to:
>
> - **Frequency-domain learning models**, which presents different works that utilize the frequency domain, not limited to transformers.
>
> - **Spectral learning and analysis**, which presents works that seek to understand frequencies in the process of learning and how it can be utilized
>
> we added works such as:
>
> [1] Rahaman, Nasim, et al. "On the spectral bias of neural networks." International conference on machine learning.
>
> [2] Jacot, Arthur, Franck Gabriel, and Clément Hongler. "Neural tangent kernel: Convergence and generalization in neural networks." Advances in neural information processing systems 31 (2018).
>
> [3] Basri, Ronen, et al. "Frequency bias in neural networks for input of non-uniform density." International conference on machine learning. PMLR, 2020.
>
>
> and more....
>
> We hope the revisions above improve the clarity, positioning, and overall presentation of this work. We would like to sincerely thank you again for your time, effort, and thoughtful feedback in reviewing our manuscript and highlighting these aspects, which have contributed to the strengthening of out work.

---

### Author Response · Authors · 2026-04-28

We sincerely thank the reviewers and editors for their careful reading and constructive feedback, which helped improve the quality and clarity of our paper. In the revised version, we implemented the following changes:


* Updated and expanded Section **E.2 Full Experimental Results** (formerly section F.2) in the Appendix to include standard deviations for all main experiments, namely, Table 1, Table 2, and Figure 3. (viewer )

* Expanded the **Fourier analysis in time series applications** paragraph in the related work to include additional works on frequency-domain deep models, spectral bias, and spectral forecasting

* Revised Appendix Section **C.4: Synth-FAR Limitations** to explicitly address the requested failure cases and clarify scenarios where Synth-FAR is less effective.

* Moved the **Datasets and Models** section to the main paper to improve clarity and make the manuscript more self-contained.


* Emphasized connection to existing literature w.r.t frequency generalization/adaptation in section 4.1.

* Improved discussion in C.1 Fundamental Frequency Estimation.

Sections, subsections or paragraphs that underwent changes were marked with a red heading for better tracking.

---

### Decision · Action_Editor_LAbR · 2026-06-09

**Recommendation:** Accept with minor revision

**Additional Comments:**

While many of the reviewers' concerns were addressed during the revision process, some questions remain regarding the fairness of the experimental comparisons and the degree of conceptual novelty. The authors are encouraged to provide additional clarification and discussion on these points in future revisions.

**Audience:**

Yes

**Audience Explanation:**

All three reviewers explicitly agreed that the findings would appeal to TMLR's audience.

Reviewer Zt78 commented that the method is relatively straightforward and could be readily integrated into follow-up studies. Reviewer C5ua also considered the use of synthetic data augmentation for zero-shot time series forecasting to be a meaningful problem with potential relevance beyond the immediate application setting.

**Claims And Evidence:**

Yes

**Claims Explanation:**

The paper received one Accept, one Leaning Accept, and one Leaning Reject recommendation. After the revision round, most of the concerns raised by the reviewers were either addressed directly or clarified with additional experiments and discussion, and the overall evidence was considered sufficient to support the main claims of the paper.

Reviewer Zt78 found the toy experiments in Figure 1 to be convincing, noting that they clearly show test MSE increasing as irrelevant frequencies are added, which supports the paper’s concepts of frequency adaptation and frequency generalization. The reviewer also considered the claim that Synth-FAR outperforms existing synthetic data generators to be reasonably supported, based on both its strongest average MSE results in Table 1 and its significantly lower data generation cost.

One concern from Reviewer Zt78 was the lack of standard deviations and uncertainty estimates in the original submission. The authors addressed this by expanding Appendix E.2 and reporting standard deviations for all major tables and figures. Following these revisions, the reviewer indicated that the concern had been resolved.

Reviewer C5ua raised questions about the assumption that the sampling frequency is known in advance and about the method’s dependence on accurately estimating the fundamental frequency. In response, the authors added further analysis in Table 2 to better characterize the robustness of the method under these conditions.

Reviewer JA2B pointed out that the method appears to be less effective when the target data do not exhibit strong periodic patterns, as seen on the Exchange dataset. The authors addressed this by revising Appendix C.4 to explicitly discuss such limitations, including low-seasonality settings and multivariate distribution mismatch scenarios.

Overall, the reviewers identified several weaknesses related to assumptions, robustness, and performance on less periodic datasets. However, the authors provided additional experiments, analyses, and discussion that substantially strengthened the paper and addressed the key concerns. Based on the revised manuscript and the reviewer feedback, the Action Editor concluded that the paper’s main claims are adequately supported by the presented evidence.

---

> ### Author Response · Authors · 2026-07-19
>
> Dear Action Editor,
>
> Thank you for your consideration. We have now uploaded the final revised version of our manuscript.
>
> We sincerely thank you and the reviewers for your time, careful evaluation, and valuable feedback, which helped us improve the paper.